# Fibrolytic vaccination against ADAM12 reduces desmoplasia in preclinical pancreatic adenocarcinomas

Jing Chen [1], Michal Sobecki[1], Ewelina Krzywinska[1], Kevin Thierry [2], Mélissa Masmoudi [2], Shunmugam Nagarajan[1], Zheng Fan[1], Jingyi He [1], Irina Ferapontova[1], Eric Nelius[1], Frauke Seehusen[3], Dagmar Gotthardt [4], Norihiko Takeda[5], Lukas Sommer [1], Veronika Sexl[6], Christian Münz [7], David DeNardo[8], Ana Hennino [2] & Christian Stockmann [1,9✉]

## Abstract

A hallmark feature of pancreatic ductal adenocarcinoma (PDAC) is massive intratumoral fibrosis, designated as desmoplasia. Desmoplasia is characterized by the expansion of cancer-associated fibroblasts (CAFs) and a massive increase in extracellular matrix (ECM). During fibrogenesis, distinct genes become reactivated specifically in fibroblasts, e.g., the disintegrin metalloprotease, *ADAM12*. Previous studies have shown that immunotherapeutic ablation of ADAM12+ cells reduces fibrosis in various organs. In preclinical mouse models of PDAC, we observe ADAM12 expression in CAFs as well as in tumor cells but not in healthy mouse pancreas. Therefore, we tested prophylactic and therapeutic vaccination against ADAM12 in murine PDAC and observed delayed tumor growth along with a reduction in CAFs and tumor desmoplasia. This is furthermore associated with vascular normalization and alleviated tumor hypoxia. The ADAM12 vaccine induces a redistribution of CD8+ T cells within the tumor and cytotoxic responses against ADAM12+ cells. In summary, vaccination against the endogenous fibroblast target ADAM12 effectively depletes CAFs, reduces desmoplasia and delays the growth of murine PDACs. These results provide proof-of-principle for the development of vaccination-based immunotherapies to treat tumor desmoplasia.

**Keywords** Pancreatic Adenocarcinoma; Vaccination; Cancer-Associated Fibroblasts; Immunotherapy
**Subject Categories** Cancer; Digestive System; Immunology

## Introduction

We have recently developed a novel vaccination-based approach to induce T cell responses that selectively ablate fibroblasts in fibrotic organs without harming organ homeostasis or physiological tissue regeneration (Sobecki et al, 2022). In brief, we demonstrated that immunization against ADAM12, endogenous proteins expressed in fibrogenic cells but highly restricted in quiescent fibroblasts, is efficient and safe in eliciting an antigen-specific cytotoxic T cell response to reduce fibroblasts in mouse models of organ fibrosis. These results provided proof-of-principle for vaccination-based immunotherapies to treat fibrosis.

Massive intratumoral fibrosis and the presence of cancer-associated fibroblasts (CAFs), also designated as desmoplasia, are hallmark features of pancreatic ductal adenocarcinoma (PDAC) (Thomas and Radhakrishnan, 2019). Pancreatic cancer is the 12th most common cancer and one of the most lethal tumor malignancies worldwide (Siegel et al, 2016; Lafaro and Melstrom, 2019; Thomas and Radhakrishnan, 2019). Despite increasing knowledge in tumor biology, PDAC is notoriously difficult to treat with limited success rates of traditional therapies such as surgery, chemotherapy, and radiation, with a 5-year overall survival rate of less than 10% (Siegel et al, 2016), largely owing to the widespread desmoplasia.

Tumor desmoplasia is driven by the expansion of CAFs and a massive deposition of extracellular matrix (ECM) components. As a consequence, up to 85% of the PDAC mass consists of stromal tissue and activated CAFs (Lafaro and Melstrom, 2019). Upon activation, CAFs increase expression of α-smooth muscle actin (α-SMA), fibroblast activation protein (FAP), fibroblast-specific protein 1 (FSP1), vimentin, and platelet-derived growth factor receptor alpha and/or beta (PDGFRα/ß) (Cazet et al, 2018; Pereira et al, 2019; Vennin et al, 2019). Activated CAFs also produce large amounts of ECM components, such as collagens and fibronectin

[1]University of Zurich, Institute of Anatomy, Winterthurerstrasse 190, CH - 8057 Zurich, Switzerland. [2]Cancer Research Center of Lyon, UMR INSERM 1052, CNRS, 5286 Lyon, France. [3]Laboratory for Animal Model Pathology (LAMP), Institute of Veterinary Pathology, Vetsuisse Faculty, University of Zurich, 8057 Zurich, Switzerland. [4]Institute of Pharmacology and Toxicology, University of Veterinary Medicine, 1210 Vienna, Austria. [5]Department of Cardiovascular Medicine, Graduate School of Medicine, The University of Tokyo, Tokyo, Japan. [6]University of Innsbruck, Innrain 52, 6020 Innsbruck, Austria. [7]Viral Immunobiology, Institute of Experimental Immunology, University of Zürich, 8057 Zürich, Switzerland. [8]Department of Pathology and Immunology, Washington University School of Medicine, St. Louis, MO, USA. [9]Comprehensive Cancer Center Zurich, 8091 Zurich, Switzerland. ✉E-mail: christian.stockmann@anatomy.uzh.ch

(Pereira et al, 2019). The stromal matrix is poorly vascularized and prevents infiltration of cytotoxic lymphocytes and drug delivery into the tumor, resulting in relentless tumor progression, facilitated metastasis, and resistance to therapy (Thomas and Radhakrishnan, 2019). Therefore, selectively targeting activated CAFs alone or in combination with tumor cells represents a promising therapeutic approach for PDAC and desmoplastic tumors in general. ADAM12 expression is correlated with reduced survival of patients with PDAC (Veenstra et al, 2018; Zhang et al, 2020) and genetic ablation of ADAM12+ mesenchymal cells has been shown to promote antitumor immunity (Di Carlo et al, 2023). Therefore, we set out to test the impact of prophylactic and therapeutic vaccination against ADAM12 on tumor desmoplasia in preclinical models of pancreatic ductal adenocarcinoma.

# Results and discussion

## Prophylactic fibrolytic vaccination against ADAM12 limits the growth of pancreatic carcinomas

We observe stromal ADAM12 expression in the "KPPC" (p48-Cre; $Kras^{LSL-G12D}$; $Trp53^{fl/fl}$) transgenic mouse model of pancreas cancer (Fig. EV1A), which is widely used because of its fidelity to human PDAC (Hegde et al, 2020). This suggests ADAM12 as a potential target to ablate CAFs in pancreatic tumors. In analogy to our previously reported successful ablation of fibroblasts in liver and lung fibrosis, we performed vaccination with GFP-expressing lentiviral vectors encoding the ADAM12 full-length protein or "empty" only GFP-expressing lentiviral vectors (v-A12 and v-CTRL, vector constructs as in (Sobecki et al, 2022)) with adjuvants (incomplete Freunds' Adjuvant (IFA) plus the TLR9 agonist CpG oligodeoxynucleotides (ODN)). First, we sought to test the outcome of immunization against ADAM12 (v-A12) prior to subcutaneous inoculation of "KPPC"-derived KP2 PDAC cells (Hegde et al, 2020) (on day -9 ("prime") and day -2 ("boost") before tumor cell injection, see scheme Fig. 1A), which form highly desmoplastic tumors (Fig. EV1B) with an accumulation of ADAM12+ cells (Fig. EV1C). ADAM12-expressing cells in subcutaneous PDACs comprised α-SMA- and PDGFR-β-expressing CAFs (42% and 14% of ADAM12+ cells, respectively) as well as a fraction of cytokeratin 19+ (CK19, 39% of ADAM12+ cells) tumor cells (Fig. EV1D). As shown in Fig. 1B, v-A12 vaccination resulted in delayed PDAC outgrowth as well as a reduction of the tumor volume by 50% at the endpoint (Fig. 1C). Tumors from v-A12 vaccinated mice were characterized by a decrease in the area covered by ADAM12-expressing cells (Fig. 1D). More precisely, v-A12 vaccination resulted in a reduction in absolute numbers of ADAM12-expressing α-SMA+ CAFs (mean 45 versus 19.6 ADAM12+/α-SMA+ cells per high power field (HPF) in v-CTRL and v-A12, respectively, Fig. EV1F) and PDGFR-β + CAFs (mean 17.1 versus 5.7 ADAM12+/PDGFR-β+ cells per high power field in v-CTRL and v-A12 mice, respectively, Fig. EV1F) but not in CK19+ tumor cells (mean 42 versus 46.5 ADAM12+/CK19+ cells per high power field in v-CTRL and v-A12 mice, respectively, Fig. EV1F). This suggests that the vaccination dominantly targets ADAM12+ CAFs and that the ADAM12+ tumor cells can evade immune cell-mediated destruction in vivo. The reduction of intratumoral ADAM12+/α-SMA+ CAFs (Fig. EV1F) upon immunization was associated with a

lower tumor collagen content (Fig. 1E) as well as a more tubular morphology of the tumors, and hence more reminiscent of benign pancreatic tissue (Fig. 1F). Noteworthy, this was associated with a reduced number of proliferating (KI67+) ADAM12+ cells as well as CK19+ tumor cells (Appendix Fig. S1A,B).

ADAM12 vaccination reduced the number of intratumoral neutrophils and macrophages (Fig. EV2A) but did not affect total CD4+ and CD8+ T cell counts (Fig. 2A) within the tumor. Next, we analyzed the intratumoral localization of CD4+ and CD8+ T cells by means of immunofluorescence. In tumors from v-CTRL mice, about 80% of CD8+ T cells were located at the tumor edge, whereas only 20% of the cells infiltrated the tumor center (Fig. 2B,C). Of note, v-A12 treatment enhanced CD8+ T cell infiltration into the tumor center (about 55% of CD8+ T cells), suggesting a massive redistribution of cytotoxic T cells from the tumor edge towards the intratumoral compartment. This is consistent with the notion that desmoplasia-associated massive ECM can prevent T cell infiltration of tumors. Moreover, among splenic cells from tumor-bearing mice, we detected an increase in CD62L- effector T cells, along with a decrease in naïve T cells within the CD4+ (Fig. 2D) and CD8+ (Fig. 2E) T cell populations after ADAM12 vaccination, suggesting a specific T cell response. This was further substantiated by the release of IFN-γ by splenic CD8+ but not CD4+ T cells from tumor-bearing mice upon exposure to ADAM12-expressing dendritic cells (Fig. 2F). Next, we performed killing assays with splenic CD8+ T cells from immunized tumor-bearing mice and different target cells, MHC-I-deficient YAC-1 lymphoma cells with minimal ADAM12 expression as well as ADAM12-expressing KP2 PDAC cells and NIH 3T3 A12 fibroblasts (Fig. 2G). Whereas spontaneous killing of YAC-1 cells was similar between v-CTRL and v-A12 CD8+ T cells (4% and 4.8% of specific lysis, respectively, Fig. 2H), CD8+ T cells from v-A12 mice showed increased specific lysis and IFN-γ release when exposed to KP2 tumor cells or NIH 3T3 A12 cells (Fig. 2I,J).

Certain studies raised the concern that targeting CAFs in PDAC could deplete pericytes from the tumor vasculature, increase tumor hypoxia, and, hence, promote tumor cell dissemination across blood vessels and metastasis (Özdemir et al, 2014). To address this, we assessed vascular density, pericyte coverage, and the degree of tumor hypoxia in PDACs from vaccinated mice. Consistent with previous reports on targeting desmoplasia in PDAC (Olive et al, 2009), we observe an increase in vascular density and pericyte coverage in PDACs from v-A12 mice (Fig. EV2B,C). Moreover, v-A12-mediated reduction of CAFs was associated with a decrease in tumor hypoxia as assessed with the expression of the surrogate marker glucose transporter 1 (GLUT1) (Airley et al, 2003) (Fig. EV2D). A short-coming of the subcutaneous KP2 PDAC model is the absence of spontaneous micro– and macrometastasis. However, in order to estimate the level of tumor cell intravasation and circulating tumor cells, we analyzed the expression of the pancreatic epithelial cell markers CK19 and OLFM4 in the peripheral blood (Finisguerra et al, 2015; Krzywinska et al, 2017) from v-CTRL and v-A12 PDAC-bearing mice at day 21 after tumor inoculation, when tumor volumes are still similar between treatment groups and before the onset of tumor growth divergence. The expression of CK19 and OLFM4 as a surrogate marker for circulating tumor cells were found to be similar in v-CTRL and v-A12 PDAC-bearing mice (Fig. EV2E), suggesting that vaccination with v-A12 does not lead to pericyte depletion and increased metastasis. In summary, we demonstrate that our

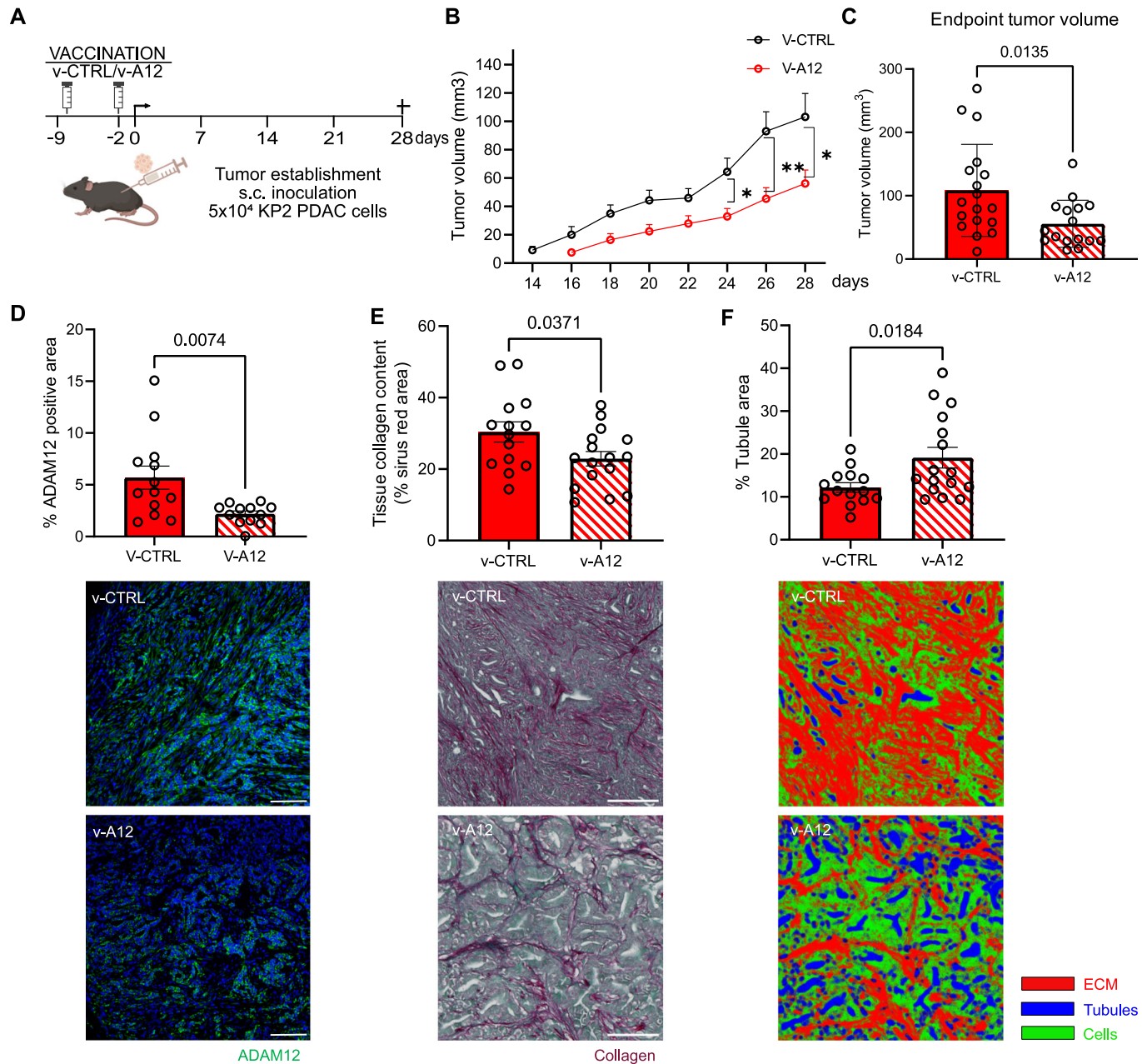

**Figure 1. Prophylactic ADAM12 vaccination hampered KP2 PDAC tumor growth.**

(A) Experimental scheme of prophylactic ADAM12 vaccination (v-A12) on subcutaneous KP2 PDAC tumor-bearing mice. (B) Time course of tumor volume measurement in $mm^3$ after tumors were measurable. ($n = 18$ mice in v-CTRL, $n = 16$ mice in v-A12; * at day 24 indicates $p = 0.0116$, ** at day 26 indicates $p = 0.0062$, * at day 28 indicates $p = 0.0229$. Data were presented as mean ± SEM. Statistical test: unpaired two-tailed Student's $t$-test.). (C) Tumor volume (C) at the endpoint. ($n = 18$ mice in v-CTRL, $n = 16$ mice in v-A12. Data were presented as mean ± SEM. Statistical test: unpaired two-tailed Student's $t$-test.). (D) Quantitative analysis of ADAM12 expression via immunofluorescence staining (ADAM12 antibody, Invitrogen, PA5-50594) and corresponding representative images (bottom). ($n = 13$ mice in v-CTRL, $n = 12$ mice in v-A12. Data were presented as mean ± SEM. Statistical test: unpaired two-tailed Student's $t$-test. Scale bar 100 μm). (E) Quantitative analysis of tissue collagen deposition (upper) by Sirius Red/Fast Green staining and corresponding representative images (bottom). ($n = 14$ mice in v-CTRL, $n = 16$ mice in v-A12. Data were presented as mean ± SEM. Statistical test: unpaired two-tailed Student's $t$-test. Scale bar 50 μm). (F) Tubule area quantification result based on Sirius red staining pictures with Image segmentation (Ilastik). ($n = 14$ mice in v-CTRL, $n = 16$ mice in v-A12. Data were presented as mean ± SEM. Statistical test: unpaired two-tailed Student's $t$-test.). Source data are available online for this figure.

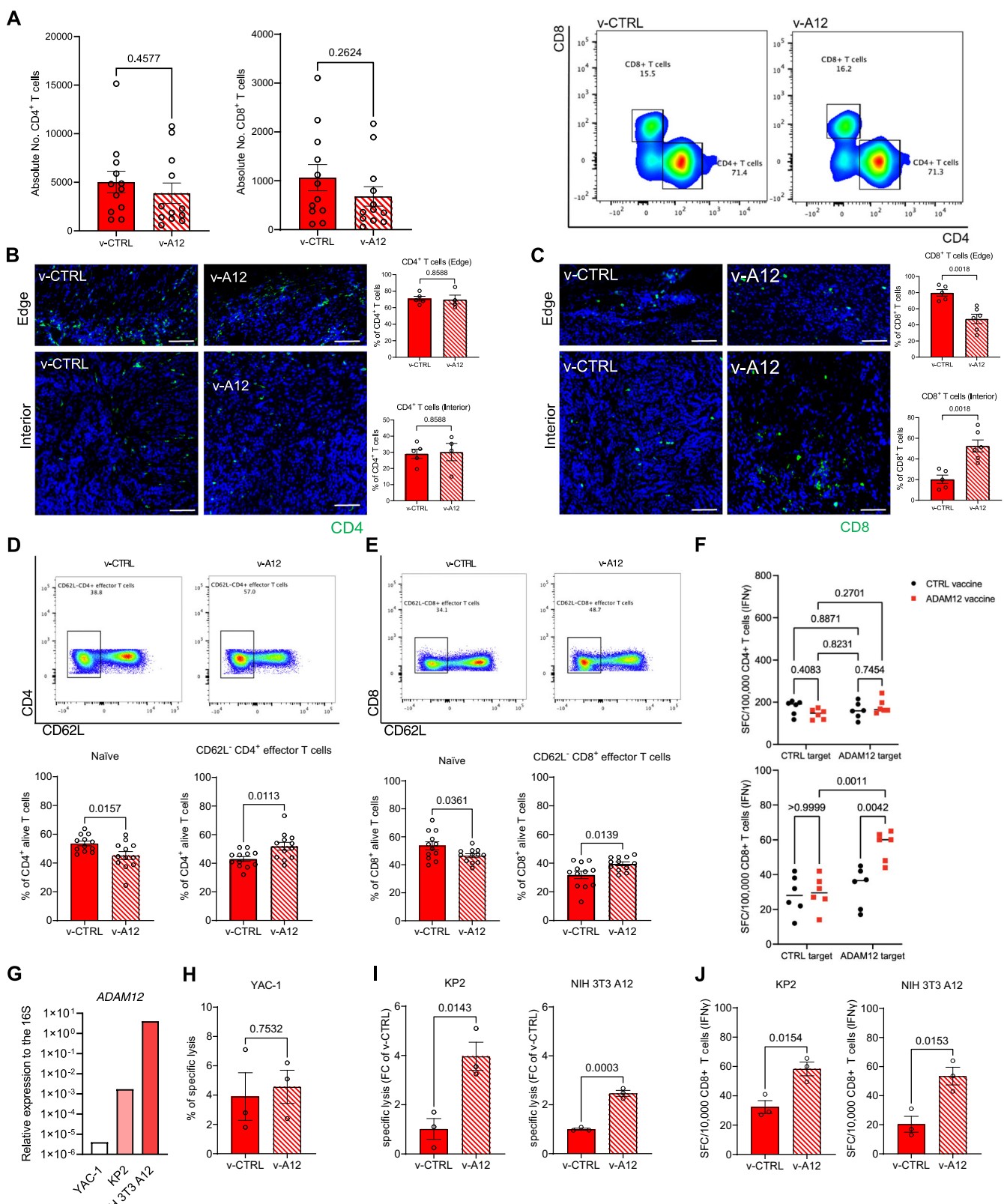

**Figure 2.** Prophylactic ADAM12 vaccination stimulates specific T cell response in PDAC tumor-bearing mice.

(A) Quantitative analysis by flow cytometry of absolute count of CD4$^+$ T cells (left) and CD8$^+$ T cells (middle) in tumor samples from subcutaneous PDAC model treated with control vaccine (v-CTRL) and ADAM12 vaccine (v-A12), along with corresponding gating strategy (right). ($n = 12$ mice in v-CTRL and v-A12 respectively. Data were presented as mean ± SEM. Statistical test: unpaired two-tailed Student's $t$-test.). (B) Quantitative analysis of CD4$^+$ T cell population at the edge of PDAC tumor tissue (left, upper) and interior of PDAC tissue (left, bottom), along with corresponding representative images. ($n = 5$ mice in v-CTRL and $n = 4$ mice in v-A12. Data were presented as mean ± SEM. Statistical test: unpaired two-tailed Student's $t$-test. Scale bar 50 μm). (C) Quantitative analysis of CD8$^+$ T cell population at the edge of PDAC tumor tissue (left, upper) and interior of PDAC tissue (left, bottom), along with corresponding representative images. ($n = 5$ mice in v-CTRL and $n = 6$ mice in v-A12. Data were presented as mean ± SEM. Statistical test: unpaired two-tailed Student's $t$-test. Scale bar 50 μm). (D) Representative flow cytometry gating (upper) of CD62L$^-$ CD4$^+$ effector T cells between control-vaccinated (v-CTRL) mice and ADAM12-vaccinated (v-A12) mice; corresponding quantitative analysis of naïve CD4$^+$ T cells (bottom, left) and CD62L$^-$ CD4$^+$ effector T cells (bottom, right) within splenic CD4$^+$ T cells from PDAC tumor-bearing mice treated with control vaccine (v-CTRL) and ADAM12 vaccine (v-A12). ($n = 12$ mice in v-CTRL and v-A12 respectively. Data were presented as mean ± SEM. Statistical test: unpaired two-tailed Student's $t$-test). (E) Representative flow cytometry gating of CD62L$^-$ CD8$^+$ effector T cells (upper) between control-vaccinated mice and ADAM12-vaccinated mice; corresponding quantitative result of naïve CD8$^+$ T cells (bottom, left) and CD62L$^-$ CD8$^+$ effector T cells (bottom, right) within splenic CD8$^+$ T cells from PDAC tumor-bearing mice treated with control vaccine (v-CTRL) and ADAM12 vaccine (v-A12). ($n = 12$ mice in v-CTRL and v-A12 respectively. Data were presented as mean ± SEM. Statistical test: unpaired two-tailed Student's $t$-test.). (F) Spot-forming cells (SFC) in IFN-γ ELISpot of splenic CD4$^+$ T cell and CD8$^+$ T cell populations purified from tumor-bearing animals vaccinated with v-CTRL or v-A12 and restimulated with MutuDC1940 cells expressing ADAM12-derived epitopes (ADAM12 target) or with corresponding CTRL target (TurboGFP alone). ($n = 6$ independent samples in v-CTRL and v-A12 respectively. Data were presented as mean ± SEM. Statistical test: two-way ANOVA.). (G) Gene expression analysis of Adam12 in YAC-1, "KPPC"-derived KP2 PDAC cells (KP2) and ADAM12-expressing NIH 3T3 cells (NIH 3T3 A12). (H) Spontaneous baseline killing of YAC-1 cells by purified splenic CD8$^+$ T cells from PDAC tumor-bearing mice vaccinated with v-CTRL or v-A12. ($n = 3$ independent samples in v-CTRL and v-A12, respectively. Data were presented as mean ± SEM. Statistical test: unpaired two-tailed Student's $t$-test.). (I) Cytotoxic assay of purified splenic CD8$^+$ T cells from PDAC tumor-bearing mice vaccinated with v-CTRL or v-A12 with target cells (KP2, NIH 3T3 A12). Data represented as fold change to v-CTRL. ($n = 3$ samples in v-CTRL and v-A12, respectively. Data were presented as mean ± SEM. Statistical test: unpaired two-tailed Student's $t$-test.). (J) Spot-forming cells (SFC) in IFN-γ ELISpot of purified splenic CD8$^+$ T cells from PDAC tumor-bearing mice vaccinated with control vaccine (v-CTRL) or ADAM12 vaccine (v-A12) co-cultured with target cells (KP2, NIH 3T3 A12). ($n = 3$ samples in v-CTRL and v-A12, respectively. Data were presented as mean ± SEM. Statistical test: unpaired two-tailed Student's $t$-test). Source data are available online for this figure.

vaccination approach can effectively reduce intratumoral fibrosis and delay tumor growth in a murine model of PDAC in a prophylactic setting.

## Therapeutic fibrolytic vaccination against ADAM12 limits the growth of pancreatic carcinomas

Next, we sought to test the outcome of immunization against ADAM12 (v-A12) in a therapeutic setting, hence after the subcutaneous inoculation of "KPPC"-derived KP2 PDAC cells (Hegde et al, 2020) (on day 21 ("prime") and day 28 ("boost") after tumor cell injection (see scheme Fig. 3A), when solid tumors were already established.

As shown in Fig. 3B,C, v-A12 vaccination resulted in a significant reduction of PDAC growth and tumor volume at the endpoint. Tumors from v-A12 vaccinated animals were characterized by a decrease in the area covered by ADAM12-expressing cells (Fig. 3D). More precisely, v-A12 vaccination resulted in a reduction in absolute numbers of ADAM12-expressing α-SMA$^+$ CAFs (Fig. EV3A) and PDGFR-β$^+$ CAFs (Fig. EV3A) but not CK19$^+$ tumor cells (Fig. EV3A), suggesting that the vaccination dominantly targets ADAM12$^+$ CAFs. Of note, the reduction of ADAM12-expressing α-SMA$^+$ CAFs (Fig. EV3A) was also associated with a lower tumor collagen content (Fig. 3E) and increased tubular appearance of PDACs (Fig. 3F). Noteworthy, this was associated with a reduced number of proliferating (KI67$^+$) ADAM12$^+$ cells as well as CK19$^+$ tumor cells (Fig. EV3B,C).

ADAM12 vaccination did neither affect the number of intratumoral neutrophils and macrophages subsets (Appendix Fig. S2A), nor total CD4$^+$ and CD8$^+$ T cell subsets (Appendix Fig. S2B) within the tumor or the spleen (Appendix Fig. S2E,F). However, when we analyzed the localization of CD4$^+$ and CD8$^+$ T cells within tumors, we observed again a redistribution of CD8$^+$ T cells from the edge towards the center of the tumor (Fig. 3G,H; Appendix Fig. S2C,D).

Consistent with previous reports on targeting desmoplasia in PDAC (Olive et al, 2009), we observe an increase in vascular density and pericyte coverage in PDACs from v-A12 mice (Fig. EV4A,B). Next, we wanted to assess whether the v-A12-related vascular changes translate into altered tumor perfusion by means of FITC-Lectin injection and simultaneous immunodetection of CD31 on tumor sections. As shown in Fig. EV4B, v-A12 increased the Lectin/CD31 double-positive area, indicating enhanced tumor perfusion. Moreover, v-A12-mediated reduction of CAFs was associated with a decrease in tumor hypoxia as assessed by Hypoxyprobe staining as well as the expression of the surrogate marker glucose transporter 1 (GLUT1) (Airley et al, 2003) (Fig. EV4C). The poorly vascularized stromal matrix is and prevents drug delivery into the tumor, and contributes to resistance to therapy (Thomas and Radhakrishnan, 2019). When we treated subcutaneous PDACs on day 29 after the second therapeutic v-A12 vaccination (day 28 after tumor cell injection) with the cytotoxic agent gemcitabine (10 mg/kg, Fig. EV3D), the chemotherapy-induced tumor cell apoptosis was significantly enhanced in the v-A12 cohort Fig. EV3E. This suggests that, in combination with chemotherapy, v-A12 is likely to enhance drug delivery and improve the responsiveness of the tumor to cytotoxic agents. The expression of CK19 and OLFM4 as a surrogate marker for circulating tumor cells were found to be similar in v-CTRL and v-A12 PDAC-bearing mice (Fig. EV4D), suggesting that vaccination with v-A12 does not lead to increased metastasis.

In summary, we demonstrate that our vaccination approach can effectively reduce intratumoral fibrosis and tumor growth in a murine model of PDAC in a therapeutic setting.

## Vaccination against ADAM12 limits the growth of orthotopic pancreatic carcinomas

Next, we sought to test the outcome of therapeutic immunization against ADAM12 in an orthotopic model, with inoculation of

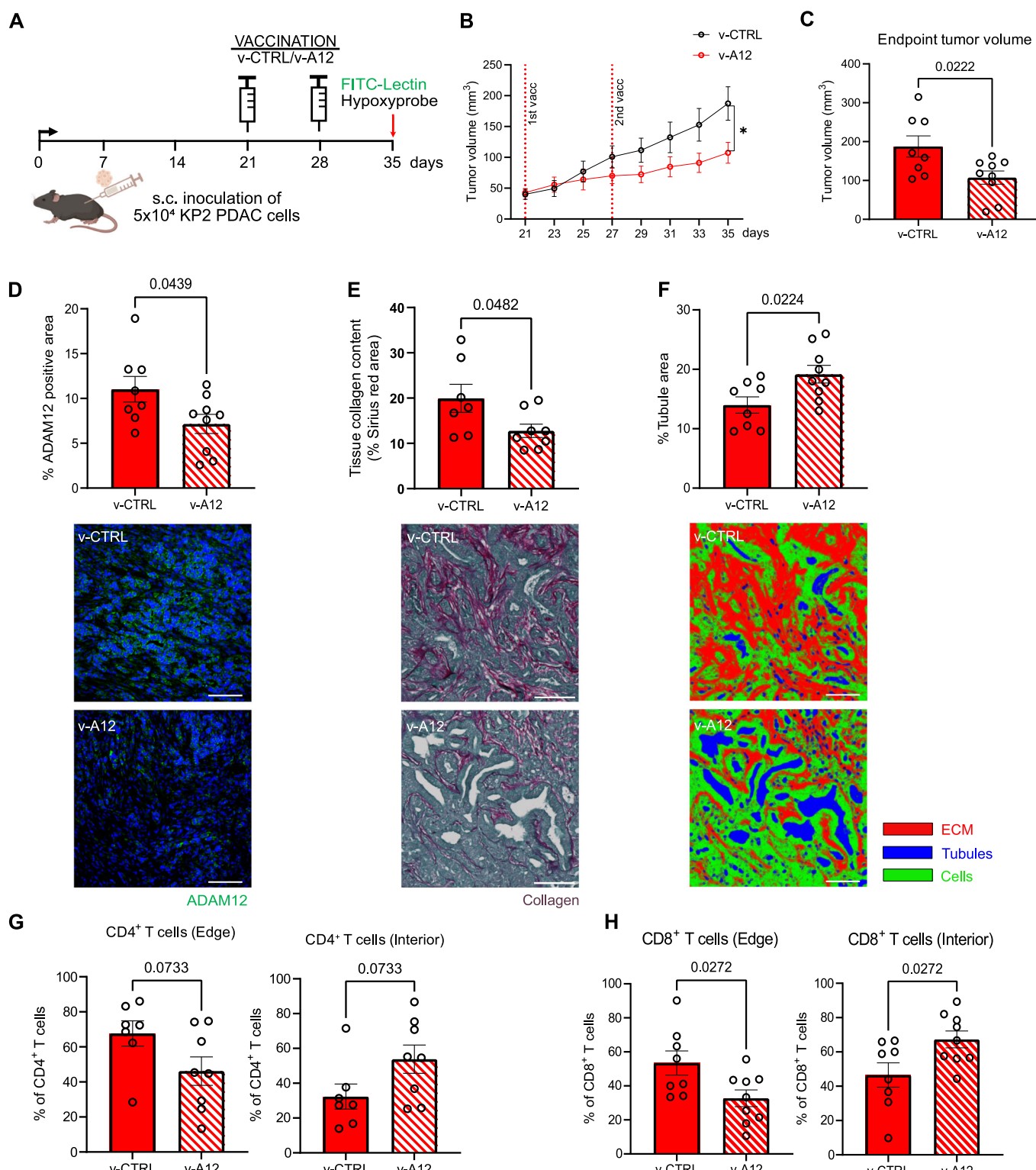

“KPC”-derived KPC PDAC cells (Hegde et al, 2020). Vaccination with v-A12 and v-CTRL was performed (on day 8 (“prime”) and day 15 (“boost”) after tumor cell injection (see scheme Fig. 4A). As shown in Fig. 4B, v-A12 vaccination resulted in a reduction of the tumor volume by 29.5% at the endpoint (Fig. 4B). Tumors from

v-A12 vaccinated mice were characterized by a decrease in the area covered by ADAM12-expressing cells (Fig. 4C). More precisely, v-A12 vaccination resulted in a reduction in absolute numbers of ADAM12-expressing α-SMA⁺ CAFs (mean 30 versus 15 ADAM12⁺/α-SMA⁺ cells per high power field (HPF) in v-CTRL

**Figure 3.  Therapeutic ADAM12 vaccination limits the growth of subcutaneous pancreatic carcinoma.**

(A) Scheme of therapeutic vaccination on subcutaneous KP2 PDAC tumor-bearing mice. (B) Time course of tumor volume (in mm³) of KP2 PDAC allografts upon the first prime vaccination (day 21). ($n = 8$ mice in v-CTRL, $n = 9$ mice in v-A12, * indicates $p = 0.0222$, Statistical test: unpaired two-tailed Student's $t$-test.). (C) KP2 PDAC tumor volume at the endpoint day 35. ($n = 8$ mice in v-CTRL, $n = 9$ mice in v-A12. Data were presented as mean ± SEM. Statistical test: unpaired two-tailed Student's $t$-test.). (D, E) Quantitative analysis of ADAM12-positive area (D, upper) (ADAM12 antibody, Invitrogen, PA5-50594) and collagen deposition (E, upper) on tumor sections of KP2 PDAC allografts treated with v-CTRL and v-A12 and representative images of ADAM12 immunostaining (D, bottom) and Sirius Red/Fast Green staining (E, bottom). ($n = 7$–8 mice in v-CTRL, $n = 9$ mice in v-A12. Data were presented as mean ± SEM. Statistical test: unpaired two-tailed Student's $t$-test. Scale bar 100 μm). (F) Quantitative analysis of tubule area on PDAC tumors treated with v-CTRL and v-A12 (upper), and representative images of image segmentation strategy (bottom). ($n = 8$ mice in v-CTRL, $n = 9$ mice in v-A12. Data were presented as mean ± SEM. Statistical test: unpaired two-tailed Student's $t$-test. Scale bar 50 μm). (G) Quantitative analysis of CD4⁺ T cell population at the edge of PADC tumor tissue (left) and interior of PDAC tissue (right). ($n = 7$ mice in v-CTRL, $n = 8$ mice in v-A12. Data were presented as mean ± SEM. Statistical test: unpaired two-tailed Student's $t$-test. Scale bar 100 μm). (H) Quantitative analysis of CD8⁺ T cell population at the edge of PADC tumor tissue (left) and interior of PDAC tissue (right). ($n = 8$ mice in v-CTRL, $n = 9$ mice in v-A12. Data were presented as mean ± SEM. Statistical test: unpaired two-tailed Student's $t$-test. Scale bar 100 μm). Source data are available online for this figure.

and v-A12, respectively, EV5A) and PDGFR-β ⁺ CAFs (mean 6.8 versus 3.4 ADAM12⁺/PDGFR-β⁺ cells per high power field (HPF) in v-CTRL and v-A12 mice, respectively, Fig. EV5A) but not in CK19⁺ tumor cells (mean 26.6 versus 19 ADAM12⁺/CK19⁺ cells per high power field (HPF) in v-CTRL and v-A12 mice, respectively, Fig. EV5A). The reduction of intratumoral ADAM12⁺ cells (Fig. 4C) upon immunization was associated with a lower tumor collagen content (Fig. 4D).

ADAM12 vaccination increased the total number of splenic T cells, including CD8⁺ T cells and the proportion of CD62L⁻ CD8⁺ effector T cells but not CD4⁺ T cells (Fig. EV5B). Of note, the number of intratumoral total T cells and particularly CD8⁺ T cells was increased after ADAM12 vaccination (Fig. 4E), while the number of intratumoral neutrophils was reduced and macrophages remained unchanged (Fig. EV5C). Next, we analyzed the intratumoral localization of CD4⁺ and CD8⁺ T cells by means of immunofluorescence. In tumors from v-CTRL mice, about 67.7% of CD8⁺ and 53% of CD4⁺ T cells were located at the tumor edge, whereas only 32.3% and 47%, respectively, of the cells infiltrated the tumor center (Fig. EV5D). Of note, v-A12 treatment enhanced CD4⁺ and CD8⁺ T cell infiltration into the tumor center (about 66% of CD4⁺ T cells and 64.9% of CD8⁺ T cells), suggesting a significant redistribution of cytotoxic T cells from the tumor edge towards the intratumoral compartment (Fig. EV5D).

Consistent with our results from the subcutaneous PDAC model and previous reports on targeting desmoplasia in PDAC (Olive et al, 2009), we observe an increase in vascular density and pericyte coverage in PDACs from v-A12 mice (Fig. 4F). Moreover, v-A12-mediated reduction of desmoplasia was associated with a decrease in tumor hypoxia as assessed by GLUT1 expression (Fig. 4G). The ability of murine syngeneic PDAC tumor cell lines to form liver metastases has been demonstrated in orthotopic models. Consistently, we observe hepatic micrometastasis at the endpoint (Fig. EV5E). Noteworthy, reduced desmoplasia, along with the vascular normalization upon ADAM12 vaccination, was associated with reduced hepatic metastasis of orthotopic PDAC (Fig. 4H).

In summary, our results provide proof-of-concept for the feasibility to reduce tumor desmoplasia in murine PDAC with vaccine-based immunotherapy to target fibroblast-specific transcripts. Of note, we do not target the function of ADAM12 but exploit it as a "tag", that allows immunotherapeutic depletion of CAFs.

We pursued a lentiviral vector-based immunization to surmount certain limitations of peptide vaccines, such as MHC class I

restriction (Facciponte et al, 2014; Hollingsworth and Jansen, 2019). Moreover, lentiviral vaccines are known to induce a robust multi-epitope-based cytotoxic T lymphocyte response along with prolonged antigen presentation and adequate co-stimulation by dendritic cells to T cells (Dullaers and Thielemans, 2006; Stripecke, 2009; Pincha et al, 2010; Uhlig et al, 2015). Despite successful fibrolytic vaccination, desmoplasia was only partially reduced upon ADAM12 vaccination, and fibroblast populations across organs and even within a fibrotic tissue showed remarkable heterogeneity (Lynch and Watt, 2018). Therefore, vaccination targets within tumors are likely to be diverse. Moreover, the outcome of T cell-based immunotherapies can fail due to an immunosuppressive environment and the expression of checkpoint inhibitor molecules such as PD-1 in the target tissue (Devaud et al, 2013), which need to be further characterized and considered. However, upon ADAM12 vaccination and reduction of desmoplasia, we observe a relocalization of tumor-infiltrating lymphocytes, away from fibrotic tumor regions that segregate lymphocytes and tumor cells towards a more homogenous distribution across the tumor. This is in line with the notion that desmoplasia hampers infiltration of cytotoxic lymphocytes and blunts immunotherapeutic approaches. Interestingly, ADAM12 vaccination leads to increased T cell-mediated in vitro killing of KP2 PDAC cells, whereas in vivo, the number of ADAM12⁺/CK19⁺ tumor cells was not reduced upon vaccination. This is consistent with the notion that the tumor microenvironment confers protection against immune cell-mediated destruction of tumor cells (Hanahan and Weinberg, 2011). Therefore, it will be tempting to test this approach in combination with conventional tumor epitope-targeting cancer vaccines or checkpoint inhibitor therapy.

The impact of CAFs on tumor progression is controversial as genetic ablation of distinct CAF subsets can promote or abrogate tumor progression (Kalluri, 2016), likely owing to phenotypical and functional CAF heterogeneity. However, by targeting ADAM12⁺ CAFs via immunization, we reduce intratumoral fibrosis and slow down tumor growth without facilitating tumor cell intravasation. This is in line with a recent report that improves antitumor immunity and growth control by means of genetic ablation of ADAM12⁺/PDGFRα⁺ mesenchymal cells in a transgenic pancreatic cancer mouse model (Di Carlo et al, 2023). Interestingly, although the ADAM12 vaccine does apparently not induce direct killing of ADAM12⁺/CK19⁺ tumor cells in vivo, we observe a reduction of CK19⁺ proliferating tumor cells. This is consistent with the concept that CAFs support tumor cell growth, and it is, hence, conceivable

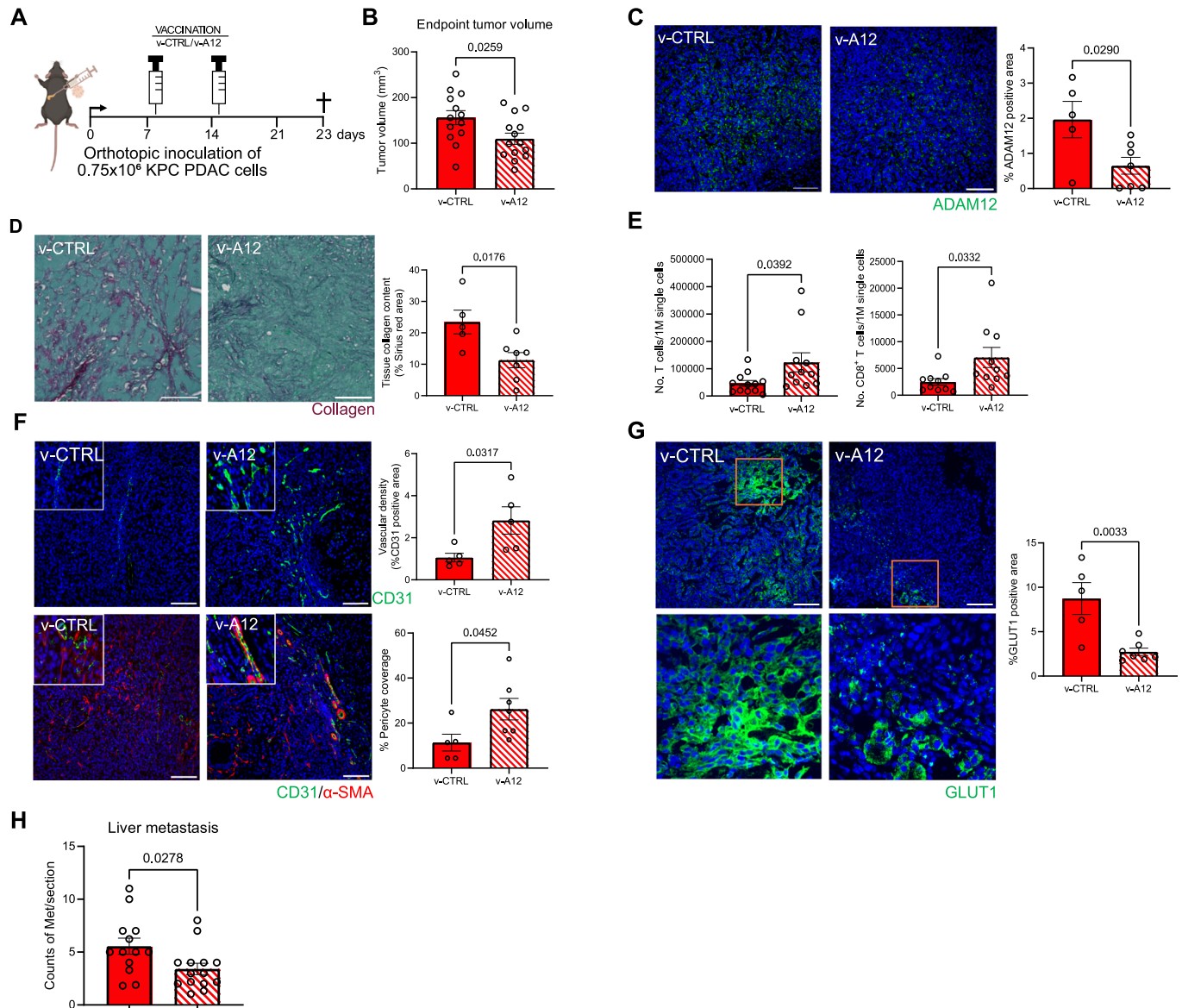

**Figure 4.  Therapeutic vaccination against ADAM12 limits the growth of orthotopic pancreatic carcinomas.**

(A) Experimental scheme of therapeutic ADAM12 vaccination (v-A12) on orthotopic KPC PDAC tumor-bearing mice. (B) Tumor volume at the endpoint, day 23. ($n = 13$ mice in v-CTRL, $n = 14$ mice in v-A12. Data were presented as mean ± SEM. Statistical test: unpaired two-tailed Student's $t$-test.). (C) Representative images of ADAM12 immunostaining (left) (ADAM12 antibody, Invitrogen, PA5-50594) and quantitative analysis of ADAM12-positive area (right) on tumor sections of orthotopic PDAC allografts treated with v-CTRL and v-A12. ($n = 5$ mice in v-CTRL, $n = 7$ mice in v-A12. Data were presented as mean ± SEM. Statistical test: unpaired two-tailed Student's $t$-test. Scale bar 100 μm). (D) Representative images of collagen tissue collagen deposition by Sirius Red/Fast Green staining (left) and quantitative analysis of collagen deposition (right) on tumor sections of orthotopic PDAC allografts treated with v-CTRL and v-A12. ($n = 5$ mice in v-CTRL and $n = 7$ mice in v-A12. Data were presented as mean ± SEM. Statistical test: unpaired two-tailed Student's $t$-test. Scale bar 50 μm). (E) FACS analysis of the total number of T cells (left) and CD8[+] T cells (right) per 1 M single cells analyzed in orthotopic PDAC tumors. ($n = 10$–12 mice in v-CTRL, $n = 10$–11 mice in v-A12. Data were presented as mean ± SEM. Statistical test: unpaired two-tailed Student's $t$-test.). (F) Representative immunofluorescence staining images (left) and quantitative analysis of vascular density as indicated by CD31 (upper, right) and pericyte coverage assessed by α-SMA/CD31 co-localization (bottom, right) on PDAC tumors with either control vaccine (v-CTRL) or ADAM12 vaccine (v-A12). ($n = 5$ mice in v-CTRL and $n = 7$ mice in v-A12. Data were presented as mean ± SEM. Statistical test: unpaired two-tailed Student's $t$-test. Scale bar 100 μm). (G) Representative immunofluorescence staining images (left) and quantitative analysis of tumor hypoxia via the coverage of GLUT1-positive cells (fight) on orthotopic PDAC from v-CTRL mice v-A12 mice. ($n = 5$ mice in v-CTRL and $n = 7$ mice in v-A12. Data were presented as mean ± SEM. Statistical test: unpaired two-tailed Student's $t$-test. Scale bar 100 μm). (H) Liver metastasis quantification. ($n = 13$ mice in v-CTRL, $n = 14$ mice in v-A12. Data were presented as mean ± SEM. Statistical test: unpaired two-tailed Student's $t$-test.). Source data are available online for this figure.

that ADAM12 vaccination reduces tumor cell proliferation via CAF ablation.

Moreover, ADAM12 vaccination results in increased vascular density and alleviated tumor hypoxia. This aligns with a study from Olive et al, showing that a reduction of CAFs and desmoplasia along with transiently increased vascular density impairs PDAC progression and metastasis (Olive et al, 2009).

Under physiological conditions, ADAM12 protein expression in humans is restricted to the reproductive organs such as the testis and placenta and undetectable in other tissues like the gut and skin (https://www.proteinatlas.org/ENSG00000148848-ADAM12/tissue). Similarly, ADAM12 gene expression in humans is very high in the placenta, while other tissues barely show ADAM12 gene expression according to the BioGPS database (ADAM12 (ADAM metallopeptidase domain 12) | Gene Report | BioGPS). In the placenta, ADAM12 expression has been associated with trophoblast invasion during pregnancy (Aghababaei et al, 2013). Therefore, we need to be concerned about the potential risk of infertility, reproductive failure, or pregnancy complications after ADAM12 vaccination. However, patients with PDAC are usually diagnosed at the age of 55-85 years old, and it's uncommon to see this disease in patients under 40 years old (within the reproductive age) (Yang et al, 2021). Moreover, our previous studies demonstrated that fibrosis vaccination does not adversely affect organ homeostasis and wound healing (Sobecki et al, 2022).

Significant advancements in preventing and treating infectious diseases, along with specific cancers, have resulted from vaccination. Therefore, we propose that vaccination could serve as both a preventive and curative immunotherapeutic strategy for addressing tumor desmoplasia.

# Methods

### Reagents and tools table

| Reagent/resource | Reference or source | Identifier or catalog number |
|---|---|---|
| **Experimental models** | | |
| C57BL/6JRj (*M.musculus*) | JANVIER LABS | |
| NIH 3T3 cell line | ATCC | ATCC number: CRL-1658 |
| YAC-1 cell line | Veronika Sexl lab provided | N/A |
| MutuDC1940 cell line | Hans Acha-Orbea lab provided | Cat# ACC635 |
| KP2 cell line | David DeNardo lab provided | N/A |
| KPC cell line | Ana Hennino lab provided | N/A |
| pLV.EF1.TurboGFP.WPRE-CTRL vaccine (v-CTRL) | Flash Therapeutics | N/A |
| pLV.EF1.Adam12.T2A.TurboGFP – ADAM12 vaccine (v-A12) | Flash Therapeutics | N/A |
| **Recombinant DNA** | | |
| pcDNA.EF1.TurboGFP | Flash Therapeutics | N/A |

| Reagent/resource | Reference or source | Identifier or catalog number |
|---|---|---|
| pcDNA.EF1.Adam12.T2A.TurboGFP | Flash Therapeutics | NM_007400.2 |
| **Antibodies** | | |
| Rabbit anti-α-SMA (1:100) | Abcam | Cat# ab5694 |
| Cy3 conjugated mouse anti-α-SMA (1:1000) | Sigma-Aldrich | Cat# C6198 |
| Rabbit anti-ADAM12 (1:100) | Thermo Fisher | Cat# PA5-50594 |
| Goat anti-ADAM12 (1:50) | Novus Biologicals | Cat# NB300-889 |
| Rabbit anti-PDGFR-β (1:100) | Cell Signaling Technology | Cat# 3169 |
| Rat anti-CK19 (10 ng/mL) | DSHB | Cat# TROMA-III |
| Rabbit anti-GLUT1 (1:100) | Abcam | Cat# ab652 |
| Rat anti-CD31 (1:10) | Dianova | Cat# DIA-31 |
| Rabbit anti-CD4 (1:100) | Abcam | Cat# ab183685 |
| Rabbit anti-CD8a (1:100) | Abcam | Cat# ab217344 |
| Sheep anti-Ki67 (1:100) | R&D system | Cat# AF7649 |
| Rabbit anti-Cleaved Caspase 3 (1:200) | Cell Signaling Technology | Cat# 9661 |
| Mouse anti-Pimonidazole (1:50) | HPI | Cat# HP1-XXX |
| Donkey anti-rabbit IgG (H + L) Cy3 (1:300) | Jackson Immunoresearch Labs | Cat# 711-165-152 |
| Donkey anti-goat IgG (H + L) Alexa Fluor 488 (1:500) | Thermo Fisher Scientific | Cat# A11055 |
| Goat anti-rabbit IgG (H + L) Alexa Fluor 488 (1:500) | Thermo Fisher Scientific | Cat# A11034 |
| Donkey anti-sheep IgG Cy3 (1:300) | Jackson Immunoresearch Labs | Cat# 713-165-003 |
| Donkey anti-rat IgG (H + L) Alexa Flour 647 (1:500) | Thermo Fisher Scientific | Cat# A78947 |
| Goat anti-rat IgG (H + L) Alexa Flour 568 (1:500) | Thermo Fisher Scientific | Cat# A11077 |
| Goat anti-mouse antibody Cy3 (1:300) | Jackson Immunoresearch Labs | Cat# 115-165-003 |
| Rat anti-CD8a (53-6.7) (1:100) | Biolegend | Cat# 100730 |
| Rat anti-CD62L (MEL-14) (1:100) | Biolegend | Cat# 104436 |
| Rat anti-CD44 (1M7) (1:100) | Biolegend | Cat# 103049 |
| Rat anti-CD45 (30-F11) (1:100) | Biolegend | Cat# 564225 |
| Rat anti-CD4 (GK1.5) (1:100) | BD Biosciences | Cat# 563790 |
| Rat anti-CD4 (GK1.5) (1:100) | BD Biosciences | Cat# 565974 |
| Hamster anti-CD3 (145-2C11) (1:100) | BD Biosciences | Cat# 562286 |
| Rat anti-F4/80 (BM8) (1:100) | Biolegend | Cat# 123128 |
| Hamster anti-CD11c (N418) (1:100) | Biolegend | Cat# 117310 |
| Rat anti-Ly6G (1A8) (1:100) | BD Biosciences | Cat# 746448 |
| Rat anti-Ly6C (HK1.4) (1:100) | Biolegend | Cat# 128035 |
| Rat anti-CD45 (30-F11) (1:100) | Biolegend | Cat# 103132 |
| Rat anti-CD11b (M1/70) (1:100) | BD Biosciences | Cat# 564443 |

| Reagent/resource | Reference or source | Identifier or catalog number |
|---|---|---|
| Rat anti-Siglec-F (E50-2440) (1:100) | BD Biosciences | Cat# 562757 |
| Rat anti-CD45 (30-F11) (1:100) | Biolegend | Cat# 103149 |
| Hamster anti-CD69 (1:100) | BD Biosciences | Cat# 551113 |
| Rat anti-CD8a (53-6.7) (1:100) | Biolegend | Cat# 100738 |
| Rat anti-CD62L (MEL-14) (1:100) | Biolegend | Cat# 104407 |
| Rat anti-CD44 (IM7) (1:100) | Biolegend | Cat# 103026 |
| Rat anti-CD4 (RM4-5) (1:100) | Biolegend | Cat# 100548 |
| Rat anti-CD3 (17A2) (1:100) | Invitrogen | Cat# 50-0032-80 |
| Rat anti-CD45 (30-F11) (30-F11) | Biolegend | Cat# 103151 |
| Hamster anti-CD11c (N418) (1:100) | Biolegend | Cat# 117302 |
| Rat anti-F4/80 (BM8) (1:100) | Biolegend | Cat# 123114 |
| Rat anti-Ly6G (1A8) (1:100) | Biolegend | Cat# 127645 |
| Rat anti-Ly6C (HK1.4) (1:100) | Biolegend | Cat# 128008 |
| Rat anti-CD11b (M1/70) (1:100) | Biolegend | Cat# 101230 |
| Anti-CD3ε (145-2C11) (1:100) | Biolegend | Cat# 100359 |
| Anti-CD28 (37.51) (1:100) | Biolegend | Cat# 102121 |
| Anti-CD31 (MEC13.3) (1:100) | Biolegend | Cat# 102514 |
| **Oligonucleotides and other sequence-based reagents** | | |
| *16s* forward primer: 5′-AGATGATCGAGCCGCGC-3′ | This paper | Fig. EV2E, EV4D |
| *Olfm4* forward primer: 5′-GCCACTTTCCAATTTCAC-3′ | This paper | Fig. EV2E, EV4D |
| *Krt19* forward primer: 5′-AATGGCGAGCTGGAGGTGAA GA-3′ | This paper | Fig. EV2E, EV4D |
| **Chemicals, enzymes, and other reagents** | | |
| Mouse Tumor Dissociation Kit | Miltenyi Biotech | Cat# 130-096-730 |
| Murine IFN-γ ELISpot assay | Diaclone | Cat# 862.031.010S |
| CD8a⁺ (Ly-2) T Cell Isolation Kit (mouse) | Miltenyi Biotech | Cat# 130-104-075 |
| CD4⁺ (L3T4) T cell isolation Kit (mouse) | Miltenyi Biotech | Cat# 130-117-043 |
| High-Capacity cDNA Reverse Transcription Kit | Applied Biosystem™ | Cat# 4368814 |
| LightCycler® 480 SYBR Green I Master MIX | Roche | Cat# 04887352 |
| TLR9 agonist CpG oligodeoxynucleotides | InvivoGen | Cat# ODN1826 |
| Incomplete Freund's adjuvant | Sigma-Aldrich | Cat# F5506 |
| Bovine Serum Albumin | Sigma-Aldrich | Cat# A9418 |
| HBSS | Thermo Fisher Scientific | Cat# 14025092 |
| **Software** | | |
| FlowJo Software (version 10.4.2) | Three Star | https://www.flowjo.com/ |
| ImageJ (version 1.54 f) | NIH | https://imagej.nih.gov/ij/ |

| Reagent/resource | Reference or source | Identifier or catalog number |
|---|---|---|
| Ilastik (version 1.4.1-b6) | Ilastik | https://www.ilastik.org/ |
| Graphpad Prism (version 10) | GraphPad Software | https://www.graphpad.com/scientific-software/prism/ |
| AID EliSpot Software | AID | N/A |
| Affinity Designer 2 | Affinity | https://affinity.serif.com/ |
| **Other** | | |

## Lentiviral vectors

Recombinant lentiviral vectors were purchased from FLASH THERAPEUTICS (France) at a minimal titer of $2.5 \times 10^9$ TU/mL. Vectors used in the following study:

1. pLV.EF1.TurboGFP.WPRE—CTRL vaccine (v-CTRL)
2. pLV.EF1.Adam12.T2A.TurboGFP – ADAM12 vaccine (v-A12)

For a detailed description of the vectors, see Fig. EV1E.

## Cell lines

The MutuDC1940 (provided by Hans Acha-Orbea lab) and YAC-1 cell line was grown in Iscove's Modified Dulbecco's Medium (IMDM, Gibco, 31980-030) supplemented with 10% FBS, 55 μM 2-Mercaptoethanol at 37 °C in a humidified atmosphere of 5% $CO_2$ in air. The NIH 3T3 and ADAM12-expressing NIH 3T3 cell lines were grown in Dulbecco's Modified Eagle Medium F12 (DMEM/F12, Gibco, 11320033) + GlutaMAX Medium supplemented with 10% FBS and 1% Penicilin-Streptomycin (Penstrep) at 37 °C in a humidified atmosphere of 5% $CO_2$ in air. The KPPC (p48-Cre; Kras$^{LSL-G12D}$; Trp53$^{fl/fl}$)-derived KP2 cell line, a kind gift from Prof. David G. DeNardo, was grown in DMEM/F12 Medium supplemented with 10% FBS and 1% Penstrep in Collagen I-precoated T-75 Flask (Corning, 365485) at 37 °C in a humidified atmosphere of 5% $CO_2$ in the air. The engineered cell lines, the ADAM12-expressing NIH 3T3 cell line and ADAM12-expressing MutuDC1940 cell line were introduced in our previous study (Sobecki et al, 2022). Cell lines were regularly tested negative for mycoplasma contamination.

## Animal models

Wide-type C57BL/6JRj female mice aged 7–8 weeks were purchased from JANVIER LABS (Le Genest-Saint-Isle, France) for PDAC studies. Mice were randomly allocated into different groups based on treatment, and investigators were blinded to the group during all experiments and analyses. All mice were housed at standard house conditions, with an indoor temperature of around 22 °C and, on average, 12 h of the light-dark cycle. The health status of all mice was regularly monitored according to FELASA guidelines. All animal experiments were approved by the local animal ethics committee (Kantonales Veterinärsamt Zürich, licenses ZH170/

2018, ZH219/2017, ZH088/2022, and ZH129/2022) and performed according to local guidelines (TschV, Zürich) and the Swiss animal protection law (TschG, Zürich).

## Mouse immunization

Mice were immunized twice (prime vaccination and a second boost vaccination 1 week later) subcutaneously under the neck with $2 \times 10^7$ TU of either ADAM12 (v-A12) or control (v-CTRL) vector formulated in 200 µL of 50% Incomplete Freund's adjuvant (Sigma-Aldrich, F5506), 50 µg TLR9 agonist CpG oligodeoxynucleotides (InvivoGen, ODN1826) in LAL water.

## Subcutaneous KP2 PDAC mouse model, prophylactic setup

To generate isografts, $5 \times 10^4$ KP2 cells in 100 µL PBS were injected subcutaneously into 8-week-old C57BL/6JRj female mice. Mice were immunized (see Section Mice immunization) on day 9 (prime) and day 2 (boost) before the tumor inoculation. Mice body weight and tumor volume were measured every 2 days when tumors were measurable, and PDAC tumor tissue samples and spleens were collected at the endpoint, day 28.

## Subcutaneous KP2 PDAC mouse model, therapeutic setup

For the therapeutic setting, $5 \times 10^4$ KP2 cells in 100 µL PBS were injected subcutaneously into 8-week-old C57BL/6JRj female mice. Mice were immunized on day 21 (prime), 3 weeks after the tumor inoculation, and followed by a boost vaccination on day 28. Blood was collected for circulating tumor cell analysis one day before the endpoint (day 35). One hour before the sacrifice, mice were intraperitoneally injected with hypoxyprobe (60 mg/kg body weight, HPI, HP1-XXX). For the tumor vascular perfusion study, mice were intravenously injected with 0.05 mg fluorescein-labeled Lycopersicon esculentum (Tomato) lectin (Vector, 1171) and were kept for 10 min before being killed. Body weight and tumor volume were measured every 2 days when tumor size was measurable, and tissue samples and spleens were collected on day 35.

For the combinational therapy with Gemcitabine, 8-week-old C57BL/6JRj female mice underwent subcutaneous injection of $5 \times 10^4$ KP2 cells in 100 µL PBS first, and then immunized with either control vaccine (v-CTRL) or ADAM12 vaccine (v-A12) on day 21 (prime) and day 28 (boost). Subsequently, mice were treated with 10 mg/kg Gemcitabine (Merck, G6423-50MG) (v-CTRL, GEM and v-A12, GEM) for 3 consecutive days. At the endpoint (day 35), mice were sacrificed, and tumor tissue samples were collected for further analysis.

## Orthotopic KPC PDAC mouse model, therapeutic setup

For the orthotopic therapeutic setting, $0.75 \times 10^6$ KPC cells in 30 µL PBS and Matrigel (Corning 354230) (PBS: Matrigel, 1:1) were injected in the pancreas of C57BL/6JRj female mice at the age of 8 to 10 weeks. Mice were vaccinated on day 8 (prime) and day 15 (boost) after the inoculation of KPC cells and were sacrificed one

week after the second vaccination. Mice weight was measured before the KPC cell injection and then monitored weekly.

## Tissue dissociation

For isolation of splenic immune cells, the spleen was mechanically disrupted, and filtered through a 100-µm cell strainer, followed by red blood cell lysis with 3-min incubation of 1x RBC Lysis Buffer (BioLegend, 420301) on ice. Splenic cells were diluted 1:10 with cold PBS and filtered through a 40-µm cell strainer. For tumor tissue dissociation: a piece of tumor tissue was processed with the Tumor Dissociation Kit (Miltenyi Biotec, 130-096-730) according to the manufacturer's instructions. The dissociated cells were further processed and stained as described in the Flow cytometry section.

## Flow cytometry

Single-cell suspension of spleen and tumor tissues were obtained as described in the Tissue dissociation section and stained for FACS analysis. Cell viability was measured using LIVE/DEAD® Fixable Aqua Dead Cell Stain Kit (Thermo Fisher, L34957). For the analysis of T cells in the subcutaneous PDAC tissues and corresponding spleen tissues, the following mAbs from eBioscience or BD Biosciences or BioLegend were used: anti-CD8a (53-6.7; 100730), anti-CD62L (MEL-14; 104436), anti-CD44 (IM7; 103049), anti-CD45 (30-F11; 564225), anti-CD4 (GK1.5; 564667, 563790), anti-CD3 (145-2C11; 562286); for the analysis of myeloid cells in the subcutaneous PDAC tissues, the following mAbs were used: anti-F4/80 (BM8; 123128), anti-CD11c (N418; 117310), anti-Ly-6G (1A8; 746448), anti-Ly-6C (HK1.4; 128035), anti-CD45 (30-F11; 564225), anti-CD11b (M1/70; 741722), anti-Siglec-F (E50-2440; 562757). For the orthotopic PDAC mouse model, the following mAbs from Biolegend or Thermo Fisher Scientific were used for T cell analysis in PDAC tissues and corresponding spleens: anti-CD45 (30-F11, 103149), anti-CD8a (53-6.7, 100738), anti-CD62L (MEL-14, 104407), anti-CD44 (IM7, 103026), anti-CD4 (RM4-5, 100548), anti-CD3 (17A2, 50-0032-80). The following mAbs were used for myeloid cell analysis in the PDAC tissue: anti-CD45 (30-F11, 103151), anti-CD11c (N418, 117337), anti-F4/80 (BM8, 123114), anti-Ly6G (1A8, 127645), anti-Ly6C (HK1.4, 128008), anti-CD11b (M1/70, 101230). Flow cytometry was carried out on a BD FACSymphony™ Flow Cytometer and LSR Fortessa Flow Cytometer. Data were analyzed using FlowJo v10 (Treestar), (see Appendix Fig. S1C; Fig. EV2A for full gating strategies).

## CD8a$^+$ T cell purification

CD8a$^+$ T cells were purified using CD8a$^+$ T Cell Isolation Kit (Miltenyi, 130-104-075), LS Column (Miltenyi, 130-042-401), and a MidiMACS™ Separator (Miltenyi, 130-042-302).

Purified CD8a$^+$ T cells were expanded for 7 days in 10 µg/mL anti-mouse CD3ε and anti-mouse CD28 precoated plates (anti-mouse CD3ε, clone 145-2C11 (Ultra-LEAF™ format, Biolegend, Cat. No. 100359); anti-mouse CD28, clone 37.51 (Ultra-LEAF™ format, Biolegend, Cat. No. 102121)) and cultured in IMDM (Gibco, 31980-030) supplemented with 10% FBS, 55 µM

2-Mercaptoethanol at 37 °C in a humidified atmosphere of 5% $CO_2$ in air.

## ELISpot assays

The frequency of IFN-γ-secreting T cells in response to a specific stimulus was accessed via using the mouse IFN-γ ELISpot assay (Diaclone). Initially, splenic CD8$^+$ T cells were purified using CD8a (Ly-2) MicroBeads (mouse, Miltenyi, 130-117-044), and splenic CD4$^+$ T cells were purified using CD4 (L3T4) MicroBeads (mouse, Miltenyi, 130-117-043) according to the instruction of the manufacturer. Subsequently, $1 \times 10^5$ CD4$^+$ T cells or $1 \times 10^5$ CD8$^+$ T cells were co-cultured with previously ADAM12-transfected $1 \times 10^5$ MutuDC1940 cells. The detection process of IFN-γ was conducted following the instruction of Diaclone. The number of spots was evaluated in a blinded manner using the AID EliSpot/FluoroSpot Reader System and then operated with AID EliSpot Software (Version 7.0). Parameters were adjusted based on the International Harmonization Guidelines for ELISPOT plate evaluation. The frequency of IFN-γ-producing cells per total number of CD4$^+$ or CD8$^+$ T cells was calculated by subtracting the number of spots in the control wells from those in the experimental wells. Additional control wells included reagents alone, non-stimulated effector cells (spontaneous IFN-γ production), CD4$^+$ T cells and CD8$^+$ T cells stimulated with target cells (MutuDC1940 cells) respectively, CD4$^+$ T cells and CD8$^+$ T cells stimulated with phorbol myristate acetate (PMA) at 1 ng/mL and ionomycin at 500 ng/mL as positive assay controls, respectively.

## In vitro cytotoxicity assays

In vitro cytotoxicity assays were conducted by co-culturing expanded CD8a$^+$ T cells with CFSE (Sigma-Aldrich)-labeled target cells: the ADAM12-expressing NIH 3T3 cells, KP2 PDAC cells, and YAC-1 cell lines, respectively at effector-to-target (E:T) ratios of 1:1 for 6 h at 37 °C in an environment with 5% $CO_2$. The culture medium used was RPMI 1640 medium (Invitrogen), supplemented with 10% FBS and 1% PenStrep. As a negative control, CFSE-labeled target cells were cultured alone. Each experimental condition was replicated three times. Subsequently, cells were washed with PBS containing 0.5% BSA (Miltenyi Biotec) and then labeled with a Live/Dead Fixable Zombie Violet (BioLegend). Data acquisition was performed using a BD FACSymphony A5 Flow Cytometer, operated by BD FACSDiva 9.0 Software. Subsequently, data analysis was carried out using FlowJo v10 (Treestar). The spontaneous cell death was defined as the proportion of dead target cells (CFSE$^+$ Live/Dead$^+$) in the negative control setting and was subtracted from the proportion of dead target cells observed in the group co-cultured with effector cells. Post-assay supernatants were utilized for the IFN-γ ELISpot assay.

## Sirius Red/Fast Green staining

Collagen fibers in subcutaneous KP2 PDAC tumor and orthotopic KPC PDAC samples were identified through the following procedure. Firstly, the sample sections were deparaffinated, rinsed with distilled water, and then subjected to a 45-s Hematoxylin stain, followed by another distilled water rinse. Subsequently, the samples were immersed in 0.04% Fast Green for 15 min, followed by

another rinse with distilled water. Next, the samples were soaked in 0.04% Sirius Red/0.1% Fast Green solution for 30 min, and then washed twice with 0.5% glacial acetic acid. Finally, the samples were dehydrated and mounted with DPX mounting media (Sigma-Aldrich, 4197-25-5). After the staining, collagen was visualized as red, while non-collagen proteins appeared in green. To analyze the collagen content of the entire PDAC tumor tissue section, the whole tissue section was scanned with the Zeiss Axio Scan.Z1 microscope at 20× magnification, and ImageJ software was utilized for the collagen quantification.

## Hematoxylin & Eosin staining

To measure liver metastasis in the orthotopic PDAC mouse model, liver tissues were collected and fixed in a 10% neutral-buffered formaldehyde solution overnight at 4 °C. Subsequently, the liver samples underwent ethanol dehydration and paraffin infiltration before being embedded in paraffin blocks. These paraffin blocks were then processed into 10-μm-thick serial sections, and liver sections were collected every five cuts. For the H&E staining, 12–15 liver sections per liver tissue were deparaffinized and rinsed with distilled water twice. Then, liver samples were immersed in alum hematoxylin solution for 30 s to stain nuclei, followed by three times rinse with distilled water. Liver samples were then stained with Eosin for 1 min, followed by serial dehydration (1× 70% ethanol, 2× 90% ethanol, 3× 100% ethanol, and 3X Histoclear) and then mounted with DPX. To analyze liver metastasis, the whole liver tissue section was scanned with the Zeiss Axio Scan.Z1 microscope at 20× magnification, and hepatic micrometastasis was quantified manually based on its histological characteristic (Fig. EV5H).

## Immunofluorescence staining

On the day of sample collection, animals were euthanized using $CO_2$, and PDAC tissue samples were excised post-mortem. These tumor tissues were then fixed in a 4% neutral-buffered formaldehyde solution overnight at 4 °C, followed by ethanol dehydration and paraffin infiltration before being embedded in paraffin blocks (FFPE). These paraffin blocks were then processed into 5 μm-thickness sections. For PDAC tumors treated with gemcitabine, tumors were embedded with OCT. 10 μm PDAC tumor cryosections were fixed in 2% neutral-buffered formaldehyde solution for 5 min, followed by washing with PBS three times for 5 min. For the vascular perfusion study in the therapeutic setting, paraffin-fixed tumor tissues were rinsed with PBS three times for 15 min each time. They were immersed in 15% sucrose and, subsequently, 30% sucrose solution. The tumor tissue was embedded with an OCT embedding matrix (Sakura, Tissue-Tek®, 4583), processed into 10-μm-thickness sections, and kept at −80 °C. The FFPE sections were deparaffinized and subjected to a heat-induced antigen retrieval step using either 1× Citrate buffer (pH 6.0, Sigma, C9999) or 1× Tris-EDTA (pH 7.4), according to the antibody instructions. As for PDAC tumor tissues embedded with OCT, tumor cryosections were replaced at room temperature before use, followed by rinsing with PBS.

For immunofluorescence staining, the sections were permeabilized with 0.5% Triton™ X-100 (Sigma-Aldrich, 9002-93-1) in PBS for 10 min and then were blocked at room temperature for 1 h

using a blocking buffer containing 0.2% bovine serum albumin (BSA, Sigma-Aldrich, A4503-50G) and other components (0.2% Triton™ X-100, 0.2% gelatin type A from porcine skin (Sigma-Aldrich, 9000-70-8), 0.2% casein (Fisher Scientific, 9000-71-9) and 0.001% sodium azide in 1× TBS). Next, primary antibodies (as listed in the table) were applied into antibody incubation buffer (1% donkey serum, 0.05% Triton™ X-100 in 1× PBS) overnight at 4 °C, followed by incubation with corresponding secondary antibodies (also listed in the table) in the same antibody incubation buffer for 1 h at room temperature. For double immunofluorescence staining, the second primary antibodies were diluted into the antibody incubation buffer and incubated for 2 h at room temperature, followed by incubation with the corresponding secondary antibodies listed in the table. DAPI (Thermo Fisher, 19158656) was used as a nuclear counterstain at a 1 μg/mL working concentration in distilled water. To quench endogenous fluorescence, samples were immersed in 0.03% Sudan Black B (Sigma-Aldrich, 4197-25-5) in 70% ethanol, and then mounted with Epredia™ Immu-Mount™ media (Fisher Scientific, 9990402).

## Gene expression by quantitative real-time PCR in blood samples

### General procedure

Blood samples were collected into BD Vacutainer EDTA Blood Collection Tubes. These blood samples were then diluted 20-fold with 1x RBC Lysis Buffer, (Biolegend, 420301) and incubated on ice for 10 min. Then the blood was centrifuged at $500 \times g$ for 10 min, and the cell pellet was lysed using RLT buffer. Total RNA was extracted using Qiagen RNA extraction kits (Qiagen, 74104) following the manufacturer's instructions.

The isolated RNA was reverse transcribed for real-time PCR analysis using a High-Capacity cDNA Reverse Transcription Kit (Applied Biosystems, 4368814). SYBR Green I Master MIX (LightCycler 480 SYBR Green I Master; Roche; 04887352001) was used for PCR reactions. A 10-ng aliquot of cDNA was used as the template to determine the relative amount of mRNA by real-time PCR, conducted on the LightCycler 96 (Roche Detection System). The PCR conditions were as follows: initial denaturation at 95 °C for 10 min, followed by 45 cycles of denaturation at 95 °C for 15 s, and annealing/extension at 60 °C for 1 min. Data obtained were normalized to 16S mRNA levels. The following primers were used: 16s forward primer: 5′-AGATGATCGAGCCGCGC-3′; *Olfm4* forward primer: 5′-GCCACTTTCCAATTTCAC-3′; *Krt19* forward primer: 5′- AATGGCGAGCTGGAGGTGAAGA-3′.

## Quantification and statistical analysis

### Image analysis

Collagen content quantification was carried out by analyzing the Sirius Red signal via Fiji (ImageJ 1.54f) on RGB images as follows: "Image/Color/Color Deconvolution" was applied firstly with user-defined values corresponding to the staining set to determine the percentage of collagen deposition area (Sirius Red positive area) within the whole tumor tissue section (Figs. 1E, 3E, 4D). Then, an equal threshold setting was uniformly applied to a set of samples.

The quantification of positive signals on immunofluorescence images was conducted using ImageJ 1.54f. A threshold was applied to determine the proportion of areas covered by the staining of

**The paper explained**

**Problem**

As one of the most lethal cancers, PDAC is characterized by an excessive accumulation of ECM components produced by cancer-associated fibroblasts (CAFs), termed desmoplasia. Desmoplasia restricts drug delivery and lymphocyte infiltration into the tumor tissue, rendering PDAC difficult to treat.

**Results**

We used subcutaneous and orthotopic PDAC mouse models which develop desmoplasia. Vaccination against ADAM12[+] reduced CAFs, ECM deposition and, hence, desmoplasia in both models. ADAM12 vaccination further promoted infiltration and a favorable redistribution of different T cell subtypes within the tumor tissue. Additionally, ADAM12 vaccination did not promote tumor metastasis but induced vascular normalization, alleviated tumor hypoxia, and delayed PDAC growth.

**Impact**

This study validated the efficacy of a vaccine against ADAM12[+] cells via improved CD8+ cytotoxic T cell response in two different PDAC tumor mouse models. This approach provides proof of principle for vaccination-based immunotherapies to treat tumor desmoplasia by specifically targeting CAFs.

interest within the analyzed images. Similarly, an equal threshold setting was consistently applied to all analyzed images. For immunofluorescence staining, we acquired 5–10 pictures randomly depending on the tumor size (Figs. 1D, 3D, 4C,D,F,G and EV1D,F, EV2B–D, EV3A–C, EV4A–C, EV5A; Appendix Fig. S1A,B). For T cell localization, we scanned the whole PDAC tumor section, and then analyzed 4-8 areas at the edge or four inner areas of tumor tissue, respectively, to quantify the number of T cells (Figs. 2C and EV5D; Appendix Fig. S2C,D). To identify pericytes, α-SMA positive cells within periportal blood vessels of intratumoral blood vessels, ranging from 50 to 150 μm, were focused. The quantification of pericyte coverage was accomplished by calculating the area of α-SMA positive cells per periportal blood vessel. To identify the lumina area, we randomly selected five areas (1000 × 1000 pixels) of each PDAC tumor tissue stained with Sirius Red/Fast Green-stained, and Ilastik was applied to segment tumor tissue compartments, tumor cells (green), tumor extracellular matrix (red), and tubule area (blue). The proportion of the tubule area was determined by calculating the tubule area (tubular length ranges above 40 pixels) within the analyzed images via ImageJ 1.54f.

### Statistical analysis

All statistical analysis was performed using GraphPad Prism v9 or v10. All data were represented as mean values and standard errors of the mean (SEM) of the corresponding number of samples. All experiments in this study were repeated at least three times. All sample numbers ($n$) of corresponding biological samples can be found in the figure legends. Outliers were identified via GraphPad Prism (Results/Column analyses/Identify outliers/ROUT/Q = 0.5%). Other exclusions were explained in the figure legends. Statistical significance was evaluated using the one-sample *t*-test, which compares the mean of the sample with a hypothetical mean, the unpaired two-tailed Student's *t*-test when appropriate, or multiple *t*-test as mentioned in the figure

legends. Before the vaccination, mice were randomly grouped. Tumor size measurement, immunofluorescence staining quantification, collagen quantification, FACS analysis, and liver metastasis were performed blindly. Statistical significance is indicated as $*P < 0.05$, $**P < 0.01$, $***P < 0.001$, and $****P < 0.0001$.

## Data availability

Images can be accessed with the following accession number: S-BIAD1324.

The source data of this paper are collected in the following database record: biostudies:S-SCDT-10_1038-S44321-024-00157-4.

## Peer review information

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

## Acknowledgements

We appreciate the support of the Swiss National Fund (310030_179235), the Swiss Cancer League (KFS-4398-02-2018 and KFS-5402-08-2021) and the Swiss National Centre for Competence in Research "Kidney.CH" (N-403-06-26 HCP project grant) to CS MS was supported by the internal postdoc funding program of the University of Zurich (Forschungskredit UZH Postdoc 2019). This research was also supported by AMED under Grant Number JP23gm6510023. We thank the Laboratory Animal Services Center (LASC) for animal care and the UZH Irchel Flow Cytometry core facility. We acknowledge the support from the SKINTEGRITY.CH collaborative research program.

## Author contributions

**Jing Chen**: Conceptualization; Data curation; Formal analysis; Investigation; Visualization; Methodology; Writing—original draft; Writing—review and editing. **Michal Sobecki**: Conceptualization; Investigation; Visualization; Methodology. **Ewelina Krzywinska**: Conceptualization; Investigation; Visualization; Methodology. **Kevin Thierry**: Formal analysis; Investigation. **Mélissa Masmoudi**: Investigation; Methodology. **Shunmugam Nagarajan**: Formal analysis; Investigation. **Zheng Fan**: Formal analysis; Investigation. **Jingyi He**: Formal analysis; Investigation. **Irina Ferapontova**: Investigation; Visualization; Methodology. **Eric Nelius**: Investigation; Methodology. **Frauke Seehusen**: Conceptualization; Resources. **Dagmar Gotthardt**: Conceptualization; Resources. **Norihiko Takeda**: Conceptualization;

Resources. **Lukas Sommer**: Conceptualization; Resources. **Veronika Sexl**: Conceptualization; Resources. **Christian Münz**: Conceptualization; Resources. **David DeNardo**: Conceptualization; Resources. **Ana Hennino**: Conceptualization; Resources; Formal analysis; Methodology. **Christian Stockmann**: Conceptualization; Resources; Supervision; Funding acquisition; Methodology; Writing—original draft; Project administration; Writing—review and editing.

Source data underlying figure panels in this paper may have individual authorship assigned. Where available, figure panel/source data authorship is listed in the following database record: biostudies:S-SCDT-10_1038-S44321-024-00157-4.

## Disclosure and competing interests statement

A patent application based on this work has been deposited.

# Expanded View Figures

**Figure EV1.  ADAM12 expression and characterization on KP2 PDAC tumor tissue.**

(**A**) Representative image of ADAM12 immunofluorescence staining (ADAM12 antibody, Invitrogen, PA5-50594) on the healthy pancreas and PDAC tumor tissue from KPPC transgenic mice. (Scale bar 100 μm). (**B**) Representative images of Sirius Red/Fast Green staining on subcutaneous KP2 PDAC sections from v-CTRL mice and v-A12 mice. (Scale bare 100 μm). (**C**) Representative image of ADAM12 immunostaining (ADAM12 antibody, Invitrogen, PA5-50594) on PDAC tumor sections from subcutaneous KP2 PDAC grafts. (Scale bar 100 μm; for the zoom-in picture, the scale bar is 50 μm). (**D**) Representative images (left) and quantitative analysis (right) of ADAM12$^+$ cell characterization, ADAM12$^+$ CAFs were indicated as ADAM12$^+$/α-SMA$^+$ (upper, left) and ADAM12$^+$/PDGFR-β$^+$ (middle, left), and ADAM12$^+$ tumor cells were indicated as ADAM12$^+$/CK19$^+$ (bottom, left) co-expression on subcutaneous KP2 PDAC tumors. ($n = 12$ fields of view from four mice. Data were presented as mean ± SEM. Scale bar is 200 μm, and the scale bar of the zoomed-in picture is 50 μm). (**E**) Vector maps of CTRL vaccine (v-CTRL) and ADAM12 vaccine (v-A12). (**F**) Quantitative analysis of ADAM12$^+$/α-SMA$^+$ (left, left), ADAM12$^+$/PDGFR-β$^+$ (left, middle), and ADAM12$^+$/CK19$^+$ (left, right) co-expression on PDAC tumors from control-vaccinated mice (v-CTRL) and ADAM12-vaccinated mice (v-A12). Data were represented as the number of cells per high power field (HPF). Representative double positive cells were indicated by white arrows respectively. ($n = 12$–16 in v-CTRL, $n = 8$–9 in v-A12. Data were presented as mean ± SEM. Statistical test: multiple *t*-test. Scale bar 50 μm).

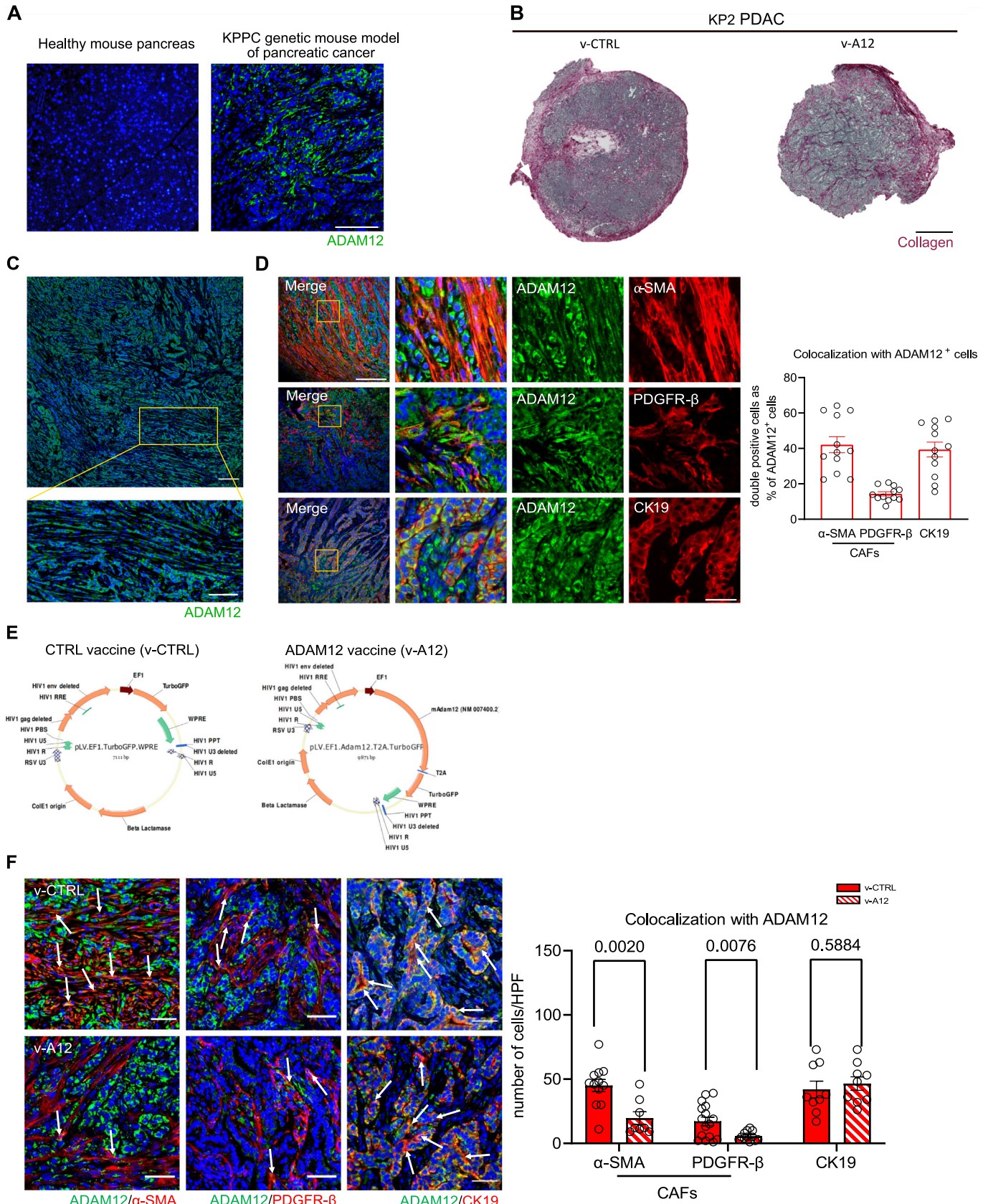

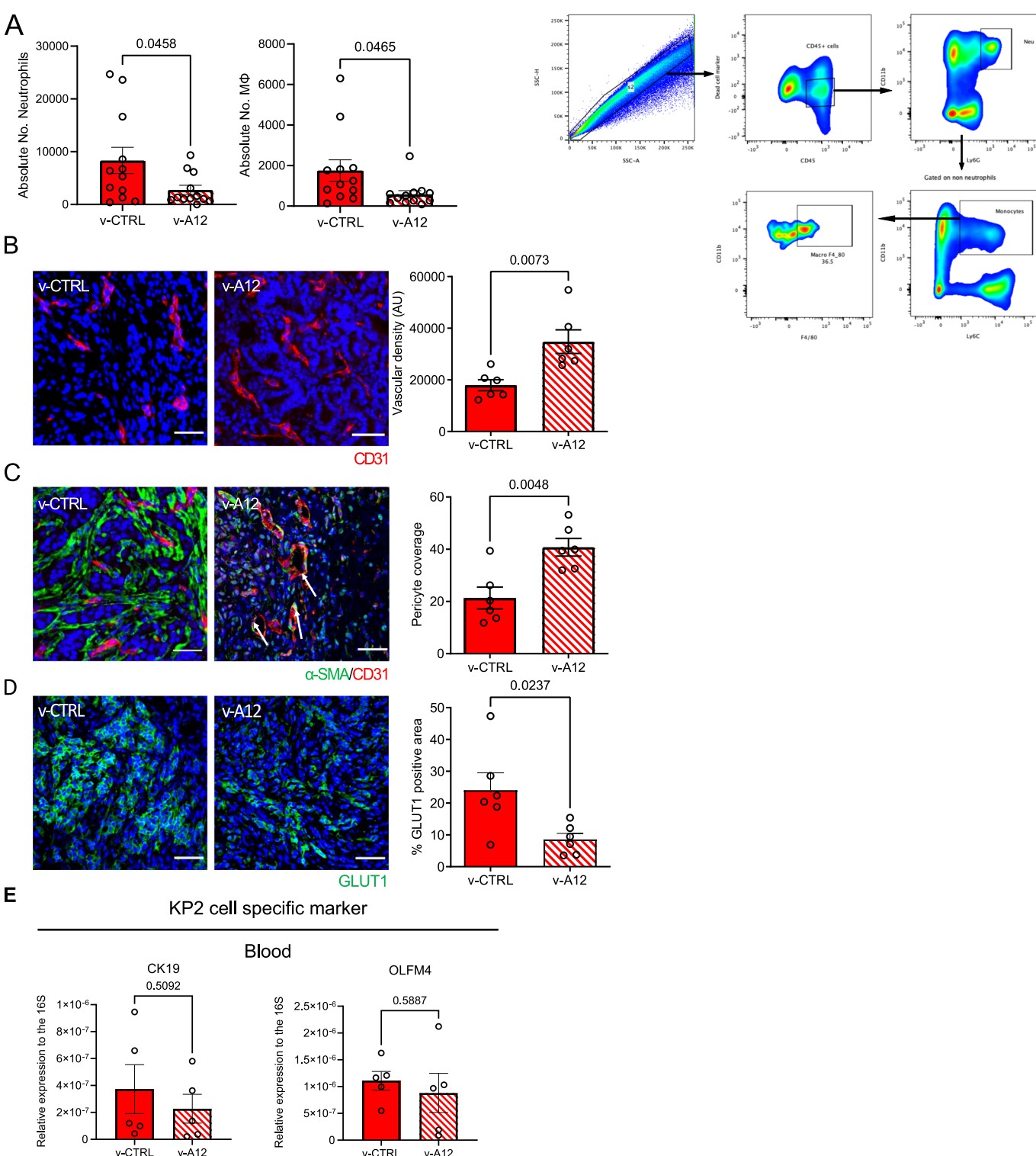

◀ **Figure EV2. Prophylactic ADAM12 vaccination increased tumor vascular density and pericyte coverage.**

(A) Quantitative analysis of the absolute number of neutrophils (left) and macrophages (middle) within PDAC tumor tissues from control-vaccinated mice (v-CTRL) and ADAM12-vaccinated mice (v-A12); corresponding flow cytometry gating strategy of myeloid cells (right): neutrophils were identified as $CD45^+CD11b^+Ly6G^+$ cells; monocytes were gated based on non-neutrophils (negative gating), and then identified as $CD11b^+Ly6C^+$ population. Then, monocyte macrophages were further defined with an $F4/80^+$ signal. ($n = 12$ mice in v-CTRL, $n = 12$ mice in v-A12. Data were presented as mean ± SEM. Statistical test: unpaired two-tailed Student's $t$-test). (B) Representative immunofluorescence staining images (left) and quantitative analysis of CD31-positive endothelial cells (right) on PDAC tumors from mice treated with either control vaccine (v-CTRL) or ADAM12 vaccine (v-A12). ($n = 6$ mice in v-CTRL, $n = 6$ mice in v-A12. Data were presented as mean ± SEM. Statistical test: unpaired two-tailed Student's $t$-test. Scale bar 50 μm). (C) Representative immunofluorescence staining images (left) and quantitative analysis of pericyte coverage assessed by α-SMA/CD31 co-localization on PDAC tumors with either control vaccine (v-CTRL) or ADAM12 vaccine (v-A12) and representative images of α-SMA/CD31 immunostaining. ($n = 6$ mice in v-CTRL, $n = 6$ mice in v-A12. Data were presented as mean ± SEM. Statistical test: unpaired two-tailed Student's $t$-test. Scale bar 50 μm). (D) Representative immunofluorescence staining images (left) and quantitative analysis of tumor hypoxia via GLUT1-positive cells on subcutaneous PDAC from v-CTRL mice and v-A12 mice together with representative images of GLUT1 immunostaining. ($n = 6$ mice in v-CTRL, $n = 6$ mice in v-A12. Data were presented as mean ± SEM. Statistical test: unpaired two-tailed Student's $t$-test. Scale bar 50 μm). (E) Gene expression analysis of the KP2-specific gene CK19 in peripheral blood at endpoint (day 28 post tumor inoculation) from subcutaneous PDAC mouse model with control vaccine (v-CTRL) or ADAM12 vaccine (v-A12). ($n = 5$ mice in v-CTRL, $n = 5$ mice in v-A12. Data were presented as mean ± SEM. Statistical test: unpaired two-tailed Student's $t$-test.)

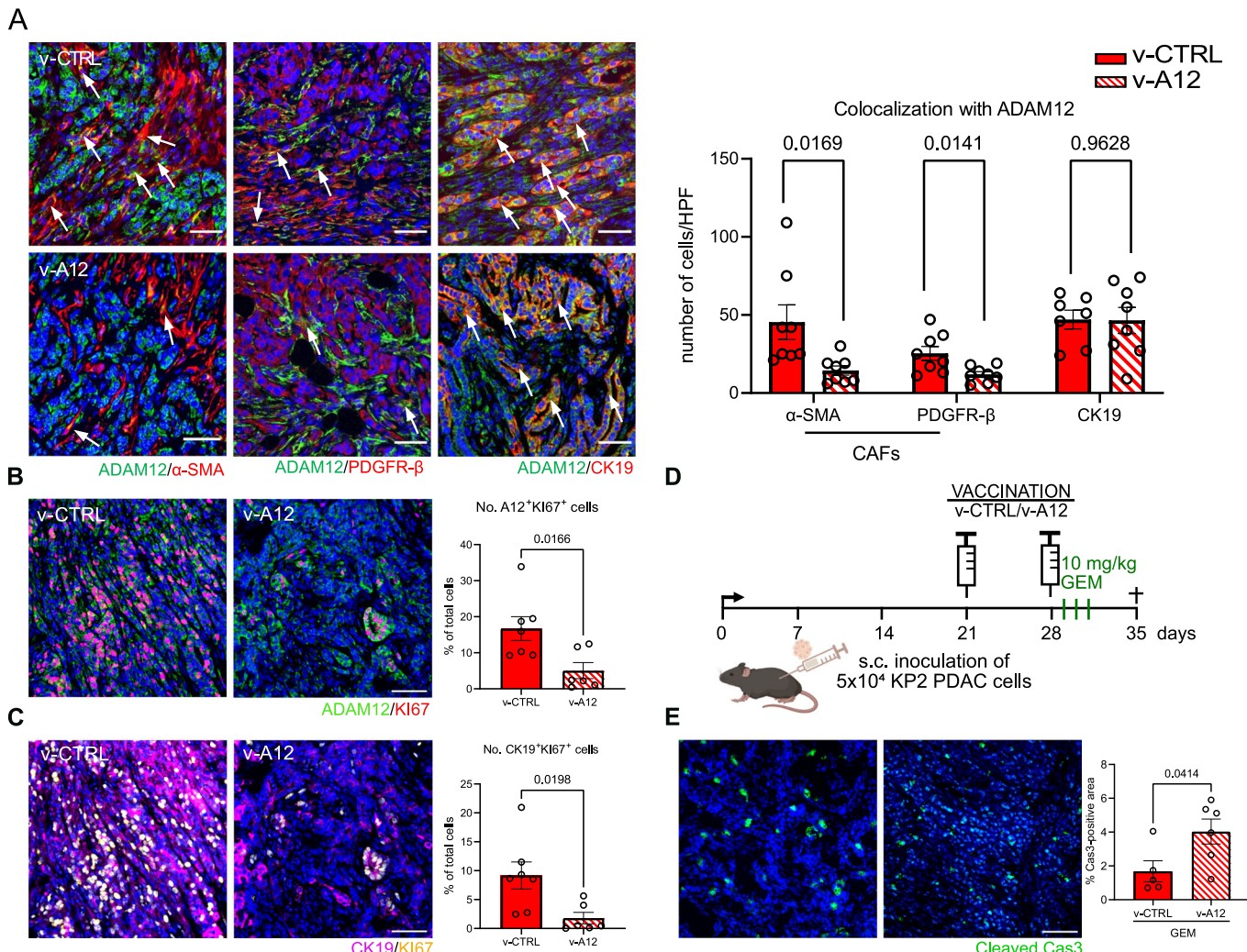

**Figure EV3. Therapeutic ADAM12 vaccination targets ADAM12+ CAFs and inhibits cell proliferation in subcutaneous PDAC tumors.**

(A) Representative immunofluorescence staining images (left) and quantitative analysis (right) of ADAM12+/α-SMA+ (left, left), ADAM12+/PDGFR-β+ (left, middle), and ADAM12+/CK19+ (left, right) co-expression on PDAC tumors from control-vaccinated mice (v-CTRL) and ADAM12-vaccinated mice (v-A12). Data were represented as the number of cells per high power field. Representative double positive cells were indicated by white arrows respectively. ($n = 7–8$ mice in v-CTRL, $n = 8$ mice in v-A12. Data were presented as mean ± SEM. Statistical test: multiple $t$-test. Scale bar 50 μm). (B) Representative immunofluorescence images (left) and quantitative analysis (right) of proliferating (Ki67+) ADAM12+ cells on PDAC tumors from v-CTRL mice and v-A12 mice. ($n = 7$ mice in v-CTRL, $n = 6$ mice in v-A12. Data were presented as mean ± SEM. Statistical test: unpaired two-tailed Student's $t$-test. Scale bar 50 μm). (C) Representative immunofluorescence staining images (left) and quantitative analysis (right) of proliferating (Ki67+) tumor (CK19+) cells on PDAC tumors from v-CTRL mice ($n = 7$) and v-A12 mice ($n = 6$). ($n = 7$ mice in v-CTRL, $n = 6$ mice in v-A12. Data were presented as mean ± SEM. Statistical test: unpaired two-tailed Student's $t$-test. Scale bar 50 μm). (D) Scheme of therapeutic vaccination with gemcitabine treatment on subcutaneous KP2 PDAC tumor-bearing mice. (E) Representative immunofluorescence images (left) and quantitative analysis (right) of apoptotic cells (cleaved caspase 3+) on PDAC tumors from v-CTRL mice and v-A12 mice. ($n = 5$ mice in v-CTRL, $n = 6$ mice in v-A12. Data were presented as mean ± SEM. Statistical test: unpaired two-tailed Student's $t$-test. Scale bar 50 μm).

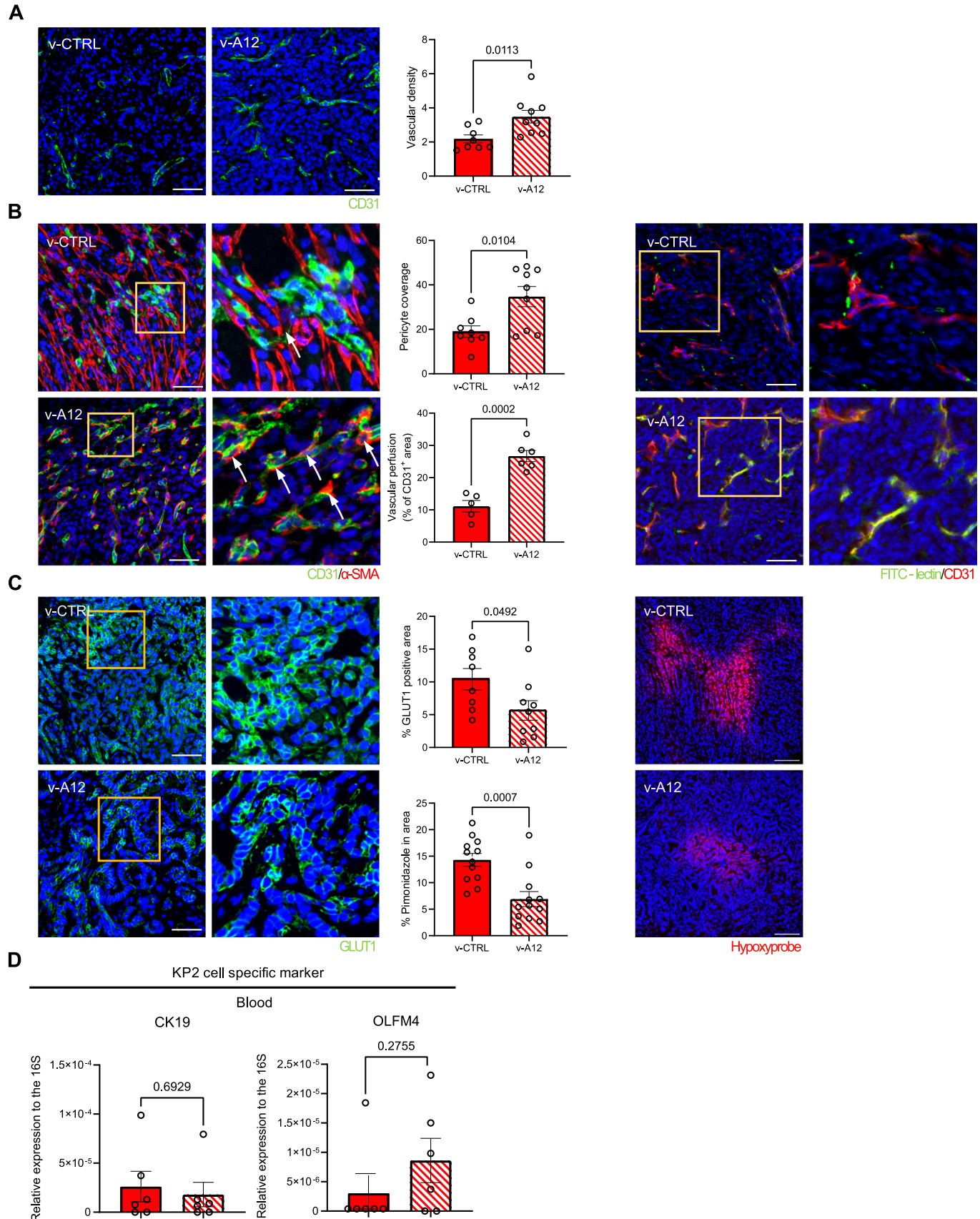

**Figure EV4. Therapeutic ADAM12 vaccination increased tumor vascular density, pericyte coverage, and decreased tumor hypoxia.**

(A) Representative images of immunofluorescence staining (left) and quantitative analysis of CD31-positive endothelial cells (right) on subcutaneous PDAC tumors treated with control vaccine (v-CTRL) and ADAM12 vaccine (v-A12) together with representative images of CD31 immunostaining. ($n = 8$ mice in v-CTRL, $n = 9$ mice in v-A12, Data were presented as mean ± SEM. Statistical test: unpaired two-tailed Student's $t$-test. Scale bar 100 µm). (B) Representative images of immunofluorescence staining (left, right) and quantitative analysis of pericyte coverage (middle, upper) assessed by α-SMA/CD31 co-localization (yellow) as well as vascular perfusion (middle, bottom) presented by FITC-lectin$^+$/CD31$^+$ cells on PDAC tumors treated with control vaccine (v-CTRL) and ADAM12 vaccine (v-A12). Representative pericytes were indicated by white arrows respectively. (Pericyte coverage: $n = 8$ mice in v-CTRL, $n = 9$ mice in v-A12; vascular perfusion: $n = 5$ mice in v-CTRL, $n = 6$ mice in v-A12. Data were presented as mean ± SEM. Statistical test: unpaired two-tailed Student's $t$-test. Scale bar 50 µm). (C) Representative images of immunofluorescence staining (left, right) and quantitative analysis of tumor hypoxia by GLUT1-positive cells (middle, upper) as well as by the coverage of Pimonidazole positive area (middle, bottom) on PDAC tumors from mice treated with control vaccine (v-CTRL, $n = 8$) or ADAM12 vaccine (v-A12, $n = 9$). (GLUT1 staining: $n = 8$ mice in v-CTRL, $n = 9$ mice in v-A12; Pimonidazole staining (hypoxyprobe): $n = 12$ mice in v-CTRL, $n = 12$ in v-A12. Data were presented as mean ± SEM. Statistical test: unpaired two-tailed Student's $t$-test. Scale bar 100 µm). (D) Gene expression analysis of the KP2-specific gene CK19 and OLFM4 in peripheral blood at endpoint (day 28 post tumor inoculation) from subcutaneous PDAC mouse model with control vaccine (v-CTRL) or ADAM12 vaccine (v-A12). ($n = 5$ mice in v-CTRL, $n = 5$ mice in v-A12. Data were presented as mean ± SEM. Statistical test: unpaired two-tailed Student's $t$-test.)

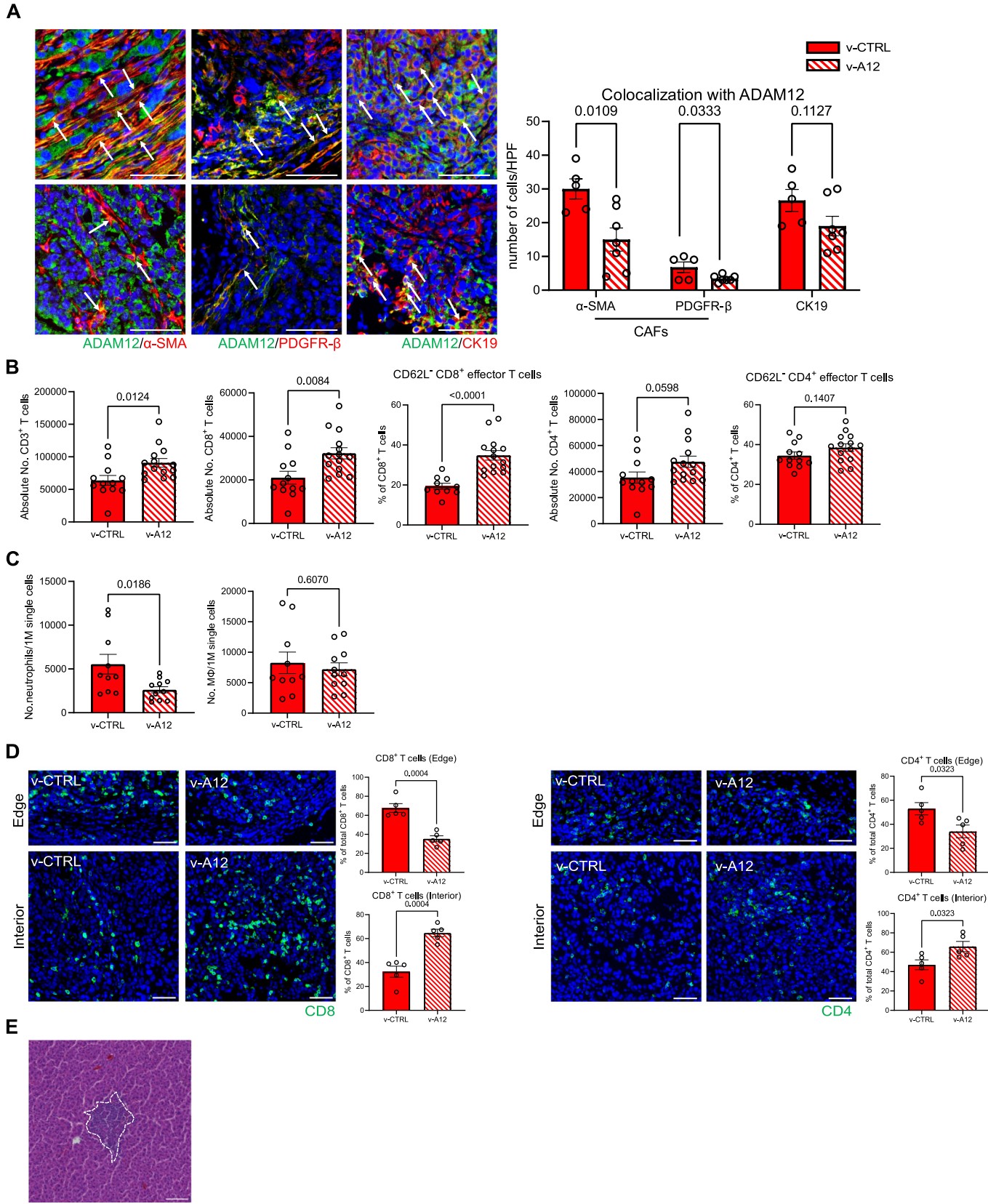

◀   **Figure EV5.   Therapeutic ADAM12 vaccination in mice bearing orthotopic PDAC tumors decreased ADAM12$^+$CAFs, stimulated splenic CD8 T cell response, and promoted T cell relocalization within the PDAC tumor tissue.**

(A) Representative immunofluorescence staining images (left) and quantitative analysis (right) of ADAM12$^+$/α-SMA$^+$ (left, left), ADAM12$^+$/PDGFR-β$^+$ (left, middle) and ADAM12$^+$/CK19$^+$ (left, right) co-expression on PDAC tumors from control-vaccinated mice (v-CTRL, $n = 5$ mice) and ADAM12-vaccinated mice (v-A12, $n = 7$ mice). Data were represented as the number of cells per high power field (HPF). Representative double positive cells were indicated by white arrows respectively. ($n = 5$ in v-CTRL, $n = 7$ in v-A12. Data were presented as mean ± SEM. Statistical test: unpaired two-tailed Student's $t$-test. Scale bar 50 μm). (B) Quantitative FACS analysis of the absolute number of T cells, CD8$^+$ T cells, CD62L$^-$CD8$^+$ effector T cells, CD4$^+$ T cells, and CD62L$^-$CD4$^+$ effector T cells in spleens from control-vaccinated mice (v-CTRL) and ADAM12-vaccinated mice (v-A12). ($n = 12$ mice in v-CTRL, $n = 14$ mice in v-A12. *$n = 10$ mice in v-CTRL in the CD62L$^-$CD8$^+$ effector T cell population because of two outliers. Data were presented as mean ± SEM. Statistical test: unpaired two-tailed Student's $t$-test.). (C) Quantitative analysis of the number of neutrophils (left) and macrophages (right) per 1 M single cells in PDAC tumors from control-vaccinated mice (v-CTRL) and ADAM12-vaccinated mice (v-A12). ($n = 10$ mice in v-CTRL, $n = 11$ mice in v-A12. Data were presented as mean ± SEM. Statistical test: unpaired two-tailed Student's $t$-test.). (D) Representative images of CD8$^+$ T cell (left) and CD4$^+$ T cell (right) localization, and quantitative analysis of CD8$^+$ T cell and CD4$^+$ T cell population at the edge of PADC tumor tissue (right, upper) and the interior of PDAC tumor tissue (right, bottom). ($n = 5$ mice in v-CTRL, $n = 5$ mice in v-A12. Data were presented as mean ± SEM. Statistical test: unpaired two-tailed Student's $t$-test. Scale bar 50 μm). (E) Representative hepatic micrometastasis (area indicated within the white dotted outline) image with H&E staining, scale bar 50 μm.

