## [Peer Review File · EMBO Molecular Medicine]

Fibrolytic vaccination against ADAM12 reduces desmoplasia in preclinical pancreatic adenocarcinomas.

Jing Chen, Michal Sobecki, Ewelina Krzywinska, Kevin Thierry, Mélissa Masmoudi, Shunmugam Nagarajan, Zheng Fan, Jingyi He, Irina Ferapontova, Eric Nelius, Frauke Seehusen, Dagmar Gotthardt, Norihiko Takeda, Lukas Sommer, Veronika Sexl, Christian Münz, David DeNardo, Ana Hennino, and Christian Stockmann

Corresponding author: Christian Stockmann (christian.stockmann@anatomy.uzh.ch)

Review Timeline:

Submission Date:	20th Oct 23
Editorial Decision:	24th Nov 23
Appeal:	26th Nov 23
Editor's Correspondence:	27th Nov 23
Revision Received:	17th May 24
Editorial Decision:	5th Jun 24
Appeal:	14th Jun 24
Editorial Decision:	1st Jul 24
Revision Received:	20th Aug 24
Editorial Decision:	16th Sep 24
Revision Received:	26th Sep 24
Accepted:	9th Oct 24

Editor: Lise Roth

Transaction Report:

23rd Nov 2023

Decision on your manuscript EMM-2023-18859

Dear Prof. Stockmann,

Thank you for the submission of your manuscript to EMBO Molecular Medicine. We have now received the feedback from two of the three referees who had agreed to review your manuscript. We have unfortunately still not received the report from referee #1 but we prefer to make a decision now in order to avoid further delay in the process.

As you will see from the reports below, the referees acknowledge the potential interest of the study, however they also raise major concerns and overall do not find that the conclusions are well supported by the data. In particular, referee #3 mentions the inadequacy of the model (also mentioned by referee #2) and the unclear effects of the vaccination on the tumor cells, tumor progression and metastasis. We feel that these shortcomings limit the translational advance that is key for publication at EMBO Molecular Medicine. Therefore, based on the referees' reports, I am afraid I see little choice but to return the manuscript to you at this point with the decision that we cannot offer to publish it.

I am very sorry to disappoint you in this occasion, and hope that the referees' comments are helpful in your continued work in this area.

Sincerely,

Lise Roth

Lise Roth
Senior Editor
EMBO Molecular Medicine

***** Reviewer's comments *****

Referee #2 (Comments on Novelty/Model System for Author):

Injecting tumor cells in pancreas instead of sc would have strengthen the manuscript but does not reject the manuscript.

Referee #2 (Remarks for Author):

Comments to authors

This manuscript describes testing a lentiviral vector expressing ADAM12 in mice with subcutaneous pancreatic tumors. ADAM12 is expressed by cancer-associated fibroblasts (CAF) and tumor cells. CAF is associated with desmoplasia and a massive deposition of extracellular matrix (ECM) components such collagens and fibronectins resulting in a stromal barrier that prevents infiltration of drugs or immune cells. Increase in CAF activation leads to increase in expression of alpha-smooth muscle actin, fibroblast activation protein, fibroblast-specific protein 1, vimentin, and platelet-derived growth factor receptor alpha and/or beta. Therefore, eliminating ADAM12-pos CAF by vaccination is of high significance. In this study they tested the ADAM12 vaccine preventively and therapeutically. In both cases they showed significant reduction in tumor size in correlation with reduction ADAM12-pos areas in the tumor, as well as collagen and tumor vasculature. Although the total number of CD4 or CD8 T cells did not decrease in the tumors, they found that CD4 and CD8 T cells migrated from the edge to the inner part of the tumor. They also found that CD8 T cells isolated from tumor-bearing mice lysed ADAM12-expressing KP2 cells and ADAM12-expressing NIH3T3 cells. In summary, reduction in the stromal barrier of pancreatic tumors by the ADAM12 vaccination allows T cells and other immune cells to infiltrate tumors more efficiently. This makes the ADAM12 vaccine attractive to combine with other immunotherapies that are hampered by the stromal barrier.

Overall, this is an interesting study with great promise to combine the ADAM12 vaccine with other immunotherapies. The stromal barrier is a great problem in pancreatic tumors and prevents penetration of drugs or infiltration of immune cells. The study is well described. However, some details/controls are lacking as outlined below.

1. Controls of CpG plus IFA only are lacking in the preventive and therapeutic immunizations. Please include also the tumor size of untreated mice.
2. Please describe the genetic background of the KP2 cell line (Kras and p53 mutations?) in Methods.
3. Why were the mice immunized in the neck and not iv or ip?
4. Are NIH 3T3 cells derived from C57BL/6 mice? Please include NIH 3T3 control cells (without ADAM12 expression) in the CTX assays.
5. Similarly are MutuDC1940 derived from C57BL/6 mice?
6. Injecting the KP2 cells in the pancreas instead of sc would have strengthened the manuscript.

Referee #3 (Comments on Novelty/Model System for Author):

Animal models for PDAC-associated desmoplasia include the interaction of tumor cells with pancreatic stellate cells, which trigger the excessive ECM accumulation characteristic of this condition. In this sense, subcutaneous injection of tumor cells does not constitute an adequate tumor model.

Referee #3 (Remarks for Author):

Chen et al. convincingly demonstrate that ADAM12 vaccination can elicit a dramatic reduction of CAFs in a subcutaneous tumor model. This CAF reduction has an impact in tumor volume and vessel density. Although, these results are promising, the tumor model chosen is not the most appropriated for PDAC. Moreover, authors do not demonstrate a benefit of ADAM12 vaccination as a cancer therapy; potential benefits that the reduction of CAFs could produce are not explored (i.e.: increased drug delivery within the tumor, reduced immunosuppression, impact on metastasis). Lastly, data representation should be improved.

Major points:

In general, explanations of how data were acquired, processed and represented are deficient (axis legends are misleading. How many pictures are quantified for each mouse? etc.). This makes difficult to assess whether the data is rigorous or not. In this context, figures 3G and S5C show the same tumor fields with a slightly different zoom (even though scale bars are equal). This is not mentioned in the figure legends or the text. Do authors have other pictures to show?

Subcutaneous injection of tumor cells does not constitute a relevant model for PDAC. Desmoplasia is generated, among other things, by the interaction of tumor cells with pancreatic stellate cells. In literature s.c. models described for PDAC-associated desmoplasia include the co-injection of PSCs together with tumor cells. The best way to explore the effect ADAM12 vaccination, would be to employ GEMMs or, alternatively, orthotopic injection of tumor cells.

In their experiments, authors show a reduction of tumor size in mice vaccinated for ADAM12. Unfortunately, this effect seems to be restricted to a reduction in CAFs. This would indicate that, although tumors are smaller, there is no real impact in tumor cell viability in vivo. Authors should clearly demonstrate that ADAM12 vaccination reduces tumor cell number or their viability alone or in combination with other treatments.

Authors present as relevant for tumor treatment the presence of CD8+ cells in the interior of tumors. The problem is that they also demonstrate that CD8+ T cells from vaccinated mice target preferentially CAFs and do not have an impact on TCs in vivo. It would have been expected that once T cell infiltration is increased, ADMA12-independent cytotoxicity would have increased as well, which would have been a sign of reduced immunosuppression. Does ADAM12+ vaccination reduce immunosuppression in the tumor?

One of the main causes for drug resistance in PDAC is that desmoplasia reduces irrigation within the tumors, making it impossible for the treatment to arrive to the tumor cells. In this sense, a more relevant experiment would have been to check for tumor perfusion in animal models for PDAC mentioned previously. If perfusion is increased, a mouse model for chemotherapy would be very interesting. Presumably, vaccinated mice should have an increased perfusion, leading to increased chemotherapy-mediated tumor cell death.

The method employed to detect circulating tumor cells is not rigorous enough. To study gene expression in blood as a direct indication of circulating tumor cells could be misleading. On top of it, the genes employed are not clear to be relevant in a chemotherapy free model. In addition, to relativize data to 16S is not the most effective way to detect differences in qPCR analyses, because its expression level is much higher than any target gene. Moreover, the concern in the scientific community is whether the absence of CAFs increases metastatic capacity of tumor cells. If this analysis is made when tumors are similar, it is expected that CAFs presence would also be similar. In this sense, this analysis would be more relevant to be carried out in the treatment model (Figure 3). Nonetheless, the most effective way to show circulating tumor cells would be to inject tagged tumor cells into mice (fluorescent or otherwise) and to find those cells in the blood circulation by flow cytometry.

Minor points:

In general, immunofluorescence images would gain clarity if they would include zoomed-out fields of the merged colors, and zoomed-in fields of single colors. For example, panel S1B shows a nice depiction of the whole tumor, it would have been interesting to see a similar image of the vaccinated animals compared to the control littermates. In particular, this is essential when authors speak about edge and interior of tumors.

Authors mention in the material methods the use of two different ADM12 antibodies. It should be indicated in each figure which antibody was employed, given that the one from Thermofisher recognizes the long form of ADAM12, while the one from Novus Biologicals recognizes short and long forms.

For figure S1D, in the text authors speak about % of ADAM12 positivity, which does not correspond with the represented data. It shows the % of SMA+, CK19+ or PDGFRb+ cells that are ADAM12+. It would be more interesting to see, as described in the results, which % of ADAM12+ cells are fibroblasts (SMA+ or PDGFRb+), or tumors (CK19).

To measure the empty space in a tumor is not a convincing parameter to imply tubular morphology. Authors should consult with a pathologist to include a more reliable measurement.

Flow cytometric analyses are not rigorous. For instance, in figure S3A, according to the density plot DCM vs CD45, there must be a problem of over-compensation between fluorophores. In addition, in the gating strategy, it is not clear whether arrows depict selected or excluded populations. Moreover, the positive selection gates often include negatively stained cells (i.e: monocyte, F4/80+ or CD44+/CD62L- gating)

Authors exaggerate in their wording. The word "massively" is used in excess, and to describe differences that are rather moderate.

Glut1 to imply hypoxia is very limited. There are many other methods, much more reliable, to do so.

As a service to authors, EMBO provides authors with the possibility to transfer a manuscript that one journal cannot offer to publish to another EMBO publication. The full manuscript and if applicable, reviewers reports are automatically sent to the receiving journal to allow for fast handling and a prompt decision on your manuscript. For more details of this service, and to transfer your manuscript to another EMBO title please click on Link Not Available

Dear Dr. Roth

Thank you very much for your message and for the rapid conveying of the reviewer's comments to us. We appreciate the comments and feel that addressing the issues raised especially by reviewer 3 with additional experiments would indeed increase the quality of the manuscript. In anticipation of some of the criticism, we have already conducted preliminary experiments in an orthotopic model of PDAC in order to e.g. assess the impact of the ADAM12 vaccine on metastasis. We have also performed experiments where we combine the ADAM12 vaccine with chemotherapy. Although, we don't know whether the pending comments from reviewer 1 are still relevant, we, hence, feel confident that we can experimentally address the concerns of the reviewers during a major revision in a reasonable amount of time, especially given that reviewer sounds rather enthusiastic about the manuscript. We would, therefore, like to ask whether you could imagine to evaluate a revised manuscript at all. If so, we would like to send you a revision plan in the form of a point-by-point response that shows how we plan to address each of the reviewer's concerns in all detail and to discuss with you whether the planned revision would be sufficient.

Sincerely,

Christian Stockmann

Dear Prof. Stockmann,

Thank you for your email and interest in submitting a revised manuscript to EMM. I discussed your study and referees' reports again with my colleagues, and we would be ready to consider a revised version of your manuscript, pending that all concerns raised by the referees are addressed. In particular, addition of an adequate model system (GEMM or orthotopic) will be essential. Please note also that explanations on how data were acquired, processed, and represented should be provided, and attention should be brought to the figure assembly.

As referee #1 still has not gone back to us, we don't expect to receive his/her comments anymore.

A provisional point-by-point rebuttal letter will not be needed, and you may directly resubmit your revised manuscript when it is ready. It will be considered an appeal, and we will aim at securing the same referees. Please note however that we cannot guarantee that the outcome will be favorable at this stage, and will depend on the completeness of your revisions.

Let me know if you have further questions.

With kind regards,

Lise Roth

***** Reviewer's comments *****

Referee #2 (Comments on Novelty/Model System for Author):

Injecting tumor cells in pancreas instead of sc would have strengthen the manuscript but does not reject the manuscript.

OUR REPLY:

We very much appreciate this comment and fully agree with the reviewer. Therefore, we have tested the ADAM12 vaccine in an orthotopic PDAC model after intrapancreatic injection of KP2 cells. The results are presented further below.

Referee #2 (Remarks for Author):

Comments to authors

This manuscript describes testing a lentiviral vector expressing ADAM12 in mice with subcutaneous pancreatic tumors. ADAM12 is expressed by cancer-associated fibroblasts (CAF) and tumor cells. CAF is associated with desmoplasia and a massive deposition of extracellular matrix (ECM) components such collagens and fibronectins resulting in a stromal barrier that prevents infiltration of drugs or immune cells. Increase in CAF activation leads to increase in expression of alpha-smooth muscle actin, fibroblast activation protein, fibroblast-specific protein 1, vimentin, and platelet-derived growth factor receptor alpha and/or beta. Therefore, eliminating ADAM12-pos CAF by vaccination is of high significance. In this study they tested the ADAM12 vaccine preventively and therapeutically. In both cases they showed significant reduction in tumor size in correlation with reduction ADAM12-pos areas in the tumor, as well as collagen and tumor vasculature. Although the total number of CD4 or CD8 T cells did not decrease in the tumors, they found that CD4 and CD8 T cells migrated from the edge to the inner part of the tumor. They also found that CD8 T cells isolated from tumor-bearing mice lysed ADAM12-expressing KP2 cells and ADAM12-expressing NIH3T3 cells. In summary, reduction in the stromal barrier of pancreatic tumors by the ADAM12 vaccination allows T cells and other immune cells to infiltrate tumors more efficiently. This makes the ADAM12 vaccine attractive to combine with other immunotherapies that are hampered by the stromal barrier.

Overall, this is an interesting study with great promise to combine the ADAM12 vaccine with other immunotherapies. The stromal barrier is a great problem in pancreatic tumors and prevents penetration of drugs or infiltration of immune cells. The study is well described. However, some details/controls are lacking as outlined below.

Our reply:

We appreciate the overall positive perception of our work and would like to thank the reviewer for the constructive comments and helpful suggestions.

1. Controls of CpG plus IFA only are lacking in the preventive and therapeutic immunizations. Please include also the tumor size of untreated mice.

Our reply:

We appreciate this comment and have conducted additional experiments by including a “CpG plus IFA only” group, termed v-Adjuvant, in the prophylactic as well as the therapeutic vaccination setting. As shown below, in the prophylactic setting, the tumor size in the v-Adjuvant group was similar to the v-CTRL cohort, whereas v-A12 treatment resulted in a significant reduction of the tumor size.

Prophylactic ADAM12 vaccination

Likewise, in the therapeutic setting, v-A12 treatment resulted in a significant reduction of the tumor size, while in the v-Adjuvant group the tumor size was similar to v-CTRL.

Therapeutic ADAM12 vaccination

In summary, this suggests that CpG plus IFA treatment has no major impact on tumor growth.

We would like to leave it to the reviewer and the editor, whether this data should be included or replace the graphs in Figure 1 and 3, respectively.

2. Please describe the genetic background of the KP2 cell line (Kras and p53 mutations?) in Methods.

Our reply:

We appreciate this comment and would like to provide the following information. The KP2 cell line was derived from female KPPC mice (p48-Cre; $Kras^{LSL-G12D}$; $Trp53^{fl/fl}$), which develop faster pancreatic cancer than KPC mice with key genetic alterations observed in human pancreatic cancer (Hegde et al., 2020a). The Cre recombinase enzyme expression is under the control of the p48 promoter. Kras is a proto-oncogene that can cause uncontrolled tumor cell growth and cancer development when it's mutated and constitutively expressed, as it is the case for the G12D mutation with a substitution of glycine (G) with aspartic acid (D) at position 12 of the Kras protein. This is combined with a deletion of the p53 tumor suppressor protein ($Trp53$), which prevents cancer by regulating apoptosis. As a consequence, KPPC mice (p48-Cre; $Kras^{LSL-G12D}$; $Trp53^{fl/fl}$) develop pancreatic tumors (Hegde et al., 2020a).

3. Why were the mice immunized in the neck and not iv or ip?

Our reply:

We appreciate this question and would like to take the opportunity to explain the rationale for subcutaneous vaccination.

Subcutaneous administration generates a depot and, hence sustained delivery of vaccine and antigen to skin antigen-presenting cells (APC) such as dermal dendritic cells (including CD103⁺cDC1s). Moreover, in previous studies (Sobecki et al. 2022), we have achieved good immunogenicity after injecting the vaccine into the neck.

Intravenous administration allows rapid delivery of vaccines, yet, the concern is that the vaccine is delivered diffusely along the systemic blood circulation, including a first-pass effect with Kupffer cells in the liver. In comparison, antigen release via subcutaneous administration is slower and more restricted to lymphatic vessels. Lastly, i.p injection is usually not considered in vaccine administration and is less effective.

4. Are NIH 3T3 cells derived from C57BL/6 mice? Please include NIH 3T3 control cells (without ADAM12 expression) in the CTX assays.

Our reply:

We would like to thank the reviewer for this question. NIH 3T3 cells are a fibroblast cell line isolated from an NIH/Swiss mouse embryo.

We have performed quantitative PCR for ADAM12 in several types of cells, including YAC-1 (MHC-1-deficient cell line and usually used for NK cell killing assays), KP2 cells, NIH 3T3 (control) cells, and ADAM12-engineered NIH 3T3 fibroblasts (NIH 3T3 A12). Of note, we observed NIH 3T3 cells express higher levels of ADAM12 mRNA than KP2 cells, but still 40 times lower than in ADAM12-engineered NIH 3T3 cells. YAC-1 cells did not show expression of ADAM12 mRNA, suggesting that YAC-1 cells are more suitable as a negative control than NIH 3T3 cells.

This was further corroborated by cell killing assays. Similar to NIH 3T3 A12 cells, although lower in amplitude, there was increased specific lysis of NIH 3T3 CTRL cells when co-cultured with splenic CD8⁺ T cells from ADAM12 vaccinated mice. In contrast, lysis of YAC-1 cells was generally low and similar for splenic CD8⁺ T cells from control vaccinated (v-CTRL) mice and ADAM12-vaccinated (v-A12) mice.

Therefore, we decided to use YAC-1 cells as (ADAM12⁻) negative control for cell killing assay and Elispot assay.

5. Similarly are MutuDC1940 derived from C57BL/6 mice?

Our reply:

The MutuDC1940 dendritic cell line, in its wild-type form, originates from mouse spleen tissues obtained from CD11c:SV40LgT transgenic mice. That is, the cells were isolated from transgenic mice of the C57BL/6 strain carrying the SV40 Large T oncogene.

6. Injecting the KP2 cells in the pancreas instead of sc would have strengthened the manuscript.

Our reply:

We very much appreciate this comment and fully agree with the reviewer. Therefore, we have tested the ADAM12 vaccine in an orthotopic PDAC model after intrapancreatic injection of KP2 cells (see experimental scheme below).

As shown below, v-A12 vaccination resulted in a significant reduction of PDAC tumor volume at the endpoint. Tumors from v-A12 vaccinated animals were characterized by a decrease in the area covered by ADAM12-expressing cells and lower tumor collagen content.

More precisely, v-A12 vaccination resulted in a reduction in absolute numbers of ADAM12-expressing α -SMA⁺ and PDGFR- β ⁺ CAFs but not in CK19⁺ tumor cells.

Next, we performed a quantification of total CK19⁺ cells. As shown below, consistent with the overall reduced tumor size, the area of CK19⁺ cells within orthotopic PDAC was reduced upon therapeutic ADAM12 vaccination.

We would like to mention that the n for histological/immunohistochemical and FACS analysis is not identical and lower than the n for tumor endpoint volume because some tumors were small in size, so we could only perform histological/immunohistochemical or FACS analysis but not both.

ADAM12 vaccination increased the total number of splenic T cells, including CD8⁺ T cells and the proportion of CD62L⁻ CD8⁺ effector T cells but not CD4⁺ T cells.

Likewise, the number of intratumoral total T cells and particularly CD8⁺ T cells was increased after ADAM12 vaccination, while the number of intratumoral neutrophils was reduced and macrophages remained unchanged.

Next, we analyzed the intratumoral localization of CD4⁺ and CD8⁺ T cells by means of immunofluorescence. In tumors from v-CTRL mice, about 67 % of CD8⁺ and 53% of CD4⁺ T cells were located at the tumor edge, whereas only 32% and 47%, respectively of the cells infiltrated the tumor center. Of note, v-A12 treatment enhanced CD4⁺ and CD8⁺ T cell infiltration into the tumor center (about 66% of CD4⁺ T cells and 64.8% of CD8⁺ T cells), suggesting a significant redistribution of cytotoxic T cells from the tumor edge towards the intratumoral compartment.

Consistent with our results from the subcutaneous PDAC model and previous reports on targeting desmoplasia in PDAC (Olive et al., 2009), we observe an increase in vascular density and pericyte coverage in PDACs from v-A12 mice. Moreover, v-A12-mediated reduction of desmoplasia was associated with a decrease in tumor hypoxia as assessed by GLUT1.

Noteworthy, reduced desmoplasia along with the vascular normalization upon ADAM12 vaccination was associated with reduced hepatic metastasis.

The results have been included in Figure 4 of the revised manuscript and are described as follows:

Next, we sought to test the outcome of therapeutic immunization against ADAM12 in an orthotopic model, with inoculation of “KPC”-derived KPC PDAC cells (Hegde et al., 2020a). Vaccination with v-A12 and v-CTRL was performed (on day 8 (“prime”) and day

15 (“boost”) after tumor cell injection (see scheme Fig. 4A). As shown in Fig. 4B, v-A12 vaccination resulted in a reduction of the tumor volume by 29.5% at the endpoint (Fig. 4B). Tumors from v-A12 vaccinated mice were characterized by a decrease in the area covered by ADAM12-expressing cells (Fig. 4C). More precisely, v-A12 vaccination resulted in a reduction in absolute numbers of ADAM12-expressing α -SMA⁺ CAFs (mean 30 versus 15 ADAM12⁺/ α -SMA⁺ cells per high power field in v-CTRL and v-A12, respectively, Fig. S7A) and PDGFR- β ⁺ CAFs (mean 6.8 versus 3.4 ADAM12⁺/PDGFR- β ⁺ cells per high power field in v-CTRL and v-A12 mice, respectively, Fig. S7A) but not in CK19⁺ tumor cells (mean 26.6 versus 19 ADAM12⁺/CK19⁺ cells per high power field in v-CTRL and v-A12 mice, respectively, Fig. S7A). The reduction of intratumoral ADAM12⁺ cells (Fig. 4C) upon immunization was associated with a lower tumor collagen content (Fig. 4D).

ADAM12 vaccination increased the total number of splenic T cells, including CD8⁺ T cells and the proportion of CD62L⁻ CD8⁺ effector T cells but not CD4⁺ T cells (Fig. S7B). Of note, the number of intratumoral total T cells and particularly CD8⁺ T cells was increased after ADAM12 vaccination (Fig. 4E), while the number of intratumoral neutrophils was reduced and macrophages remained unchanged (Fig. S7C). Next, we analyzed the intratumoral localization of CD4⁺ and CD8⁺ T cells by means of immunofluorescence. In tumors from v-CTRL mice, about 67 % of CD8⁺ and 53% of CD4⁺ T cells were located at the tumor edge, whereas only 32% and 47%, respectively of the cells infiltrated the tumor center (Fig. S7D). Of note, v-A12 treatment enhanced CD4⁺ and CD8⁺ T cell infiltration into the tumor center (about 66% of CD4⁺ T cells and 64.8% of CD8⁺ T cells), suggesting a significant redistribution of cytotoxic T cells from the tumor edge towards the intratumoral compartment (Fig. S7D). Consistent with our results from the subcutaneous PDAC model and previous reports on targeting desmoplasia in PDAC (Olive et al., 2009), we observe an increase in vascular density and pericyte coverage in PDACs from v-A12 mice (Fig. 4F). Moreover, v-A12-mediated reduction of desmoplasia was associated with a decrease

in tumor hypoxia as assessed by GLUT1 expression (Fig. 4G). The ability of murine syngeneic PDAC tumor cell lines to form liver metastases has been demonstrated in orthotopic models. Consistently, we observe hepatic micrometastasis at the endpoint (Fig. S7E). Noteworthy, reduced desmoplasia along with the vascular normalization upon ADAM12 vaccination was associated with reduced hepatic metastasis of orthotopic PDAC (Fig. 4H).

In summary, our results provide proof-of-concept for the feasibility of reducing tumor desmoplasia in murine PDAC with vaccine-based immunotherapy to target fibroblast-specific transcripts. Of note, we do not target the function of ADAM12 but exploit it as a “tag”, that allows immunotherapeutic depletion of CAFs.

Referee #3 (Comments on Novelty/Model System for Author):

Animal models for PDAC-associated desmoplasia include the interaction of tumor cells with pancreatic stellate cells, which trigger the excessive ECM accumulation characteristic of this condition. In this sense, subcutaneous injection of tumor cells does not constitute an adequate tumor model.

Our reply:

We very much appreciate this comment and fully agree with the reviewer. Therefore, we have tested the ADAM12 vaccine in an orthotopic PDAC model after intrapancreatic injection of KP2 cells. The results are presented further below.

Referee #3 (Remarks for Author):

Chen et al. convincingly demonstrate that ADAM12 vaccination can elicit a dramatic reduction of CAFs in a subcutaneous tumor model. This CAF reduction has an impact in tumor volume and vessel density. Although, these results are promising, the tumor model chosen is not the most appropriated for PDAC. Moreover, authors do not demonstrate a benefit of ADAM12 vaccination as a cancer therapy; potential benefits that the reduction of CAFs could produce are not explored (i.e.: increased drug delivery within the tumor,

reduced immunosuppression, impact on metastasis). Lastly, data representation should be improved.

Our reply:

We appreciate the very constructive comments and have assessed the impact of the ADAM12 vaccine on primary tumor growth and metastasis in an orthotopic model of PDAC. We have also analyzed the impact of the ADAM12 vaccine on tumor perfusion by means of FITC-lectin injection as well as on drug delivery by means of gemcitabine treatment. The results are described in detail below.

Major points:

In general, explanations of how data were acquired, processed, and represented are deficient (axis legends are misleading. How many pictures are quantified for each mouse? etc.). This makes it difficult to assess whether the data is rigorous or not. In this context, figures 3G and S5C show the same tumor fields with a slightly different zoom (even though scale bars are equal). This is not mentioned in the figure legends or the text. Do authors have other pictures to show?

Our reply:

We apologize for the incomplete description of the data acquisition and would like to provide additional information as follows:

For immunofluorescence stainings, we acquired 5-10 pictures randomly according to the tumor size, in the following figures: Fig 1D, Fig 3D, Fig 4C, D, F and G, Fig S1D and F, Figure S2A and B, Fig S3B-D, Fig S4A-C, Fig S6A-C, and Fig S7A.

For the analysis of the Sirius Red staining, we scanned the whole PDAC tumor tissue, and quantified collagen deposition across the whole PDAC tumor tissue via ImageJ in the following figures: Fig 1E, Fig 3E, and Fig 4D.

For T cell localization, we scanned the whole PDAC tumor section, and then analyzed 8 areas at the edge or 4 inner areas of tumor tissue, respectively, to quantify the number of T cells (see scheme below) in the following figures: Fig 2B and C, Fig S5C and D, Fig S7D

Finally, we would like to apologize for the mistake that the pictures in the figures 3 and S5, which we have corrected in the revised manuscript. The respective images in Figure 3 and S5 are now presented as follows:

Figure 3

Figure S5

Subcutaneous injection of tumor cells does not constitute a relevant model for PDAC. Desmoplasia is generated, among other things, by the interaction of tumor cells with

pancreatic stellate cells. In literature s.c. models described for PDAC-associated desmoplasia include the co-injection of PSCs together with tumor cells. The best way to explore the effect ADAM12 vaccination, would be to employ GEMMs or, alternatively, orthotopic injection of tumor cells.

Our reply:

We very much appreciate this comment and fully agree with the reviewer. Therefore, we have tested the ADAM12 vaccine in an orthotopic PDAC model after intrapancreatic injection of KP2 cells (see experimental scheme below).

As shown below, v-A12 vaccination resulted in a significant reduction of PDAC tumor volume at the endpoint. Tumors from v-A12 vaccinated animals were characterized by a decrease in the area covered by ADAM12-expressing cells and lower tumor collagen content.

More precisely, v-A12 vaccination resulted in a reduction in absolute numbers of ADAM12-expressing α -SMA⁺ and PDGFR- β ⁺ CAFs but not in CK19⁺ tumor cells.

Next, we performed a quantification of total CK19⁺ cells. As shown below, consistent with the overall reduced tumor size, the area of CK19⁺ cells within orthotopic PDAC was reduced upon therapeutic ADAM12 vaccination.

We would like to mention that the n for histological/immunohistochemical and FACS analysis is not identical and lower than the n for tumor endpoint volume because some tumors were small in size, so we could only perform histological/immunohistochemical or FACS analysis but not both.

ADAM12 vaccination increased the total number of splenic T cells, including CD8⁺ T cells and the proportion of CD62L⁻ CD8⁺ effector T cells but not CD4⁺ T cells.

Likewise, the number of intratumoral total T cells and particularly CD8⁺ T cells was increased after ADAM12 vaccination, while the number of intratumoral neutrophils was reduced and macrophages remained unchanged.

Next, we analyzed the intratumoral localization of CD4⁺ and CD8⁺ T cells by means of immunofluorescence. In tumors from v-CTRL mice, about 67 % of CD8⁺ and 53% of CD4⁺ T cells were located at the tumor edge, whereas only 32% and 47%, respectively of the cells infiltrated the tumor center. Of note, v-A12 treatment enhanced CD4⁺ and CD8⁺ T cell infiltration into the tumor center (about 66% of CD4⁺ T cells and 64.8% of CD8⁺ T cells), suggesting a significant redistribution of cytotoxic T cells from the tumor edge towards the intratumoral compartment.

Consistent with our results from the subcutaneous PDAC model and previous reports on targeting desmoplasia in PDAC (Olive et al., 2009), we observe an increase in vascular density and pericyte coverage in PDACs from v-A12 mice. Moreover, v-A12-mediated reduction of desmoplasia was associated with a decrease in tumor hypoxia as assessed by GLUT1.

Noteworthy, reduced desmoplasia along with the vascular normalization upon ADAM12 vaccination was associated with reduced hepatic metastasis.

The results have been included as Figure 4 of the revised manuscript and are described as follows:

Next, we sought to test the outcome of therapeutic immunization against ADAM12 in an orthotopic model, with inoculation of “KPC”-derived KPC PDAC cells (Hegde et al., 2020a). Vaccination with v-A12 and v-CTRL was performed (on day 8 (“prime”) and day 15 (“boost”) after tumor cell injection (see scheme Fig. 4A). As shown in Fig. 4B, v-A12 vaccination resulted in a reduction of the tumor volume by 29.5% at the endpoint (Fig. 4B). Tumors from v-A12 vaccinated mice were characterized by a decrease in the area covered by ADAM12-expressing cells (Fig. 4C). More precisely, v-A12 vaccination resulted in a reduction in absolute numbers of ADAM12-expressing α -SMA⁺ CAFs (mean 30 versus 15 ADAM12⁺/ α -SMA⁺ cells per high power field in v-CTRL and v-A12, respectively, Fig. S7A) and PDGFR- β ⁺ CAFs (mean 6.8 versus 3.4 ADAM12⁺/PDGFR- β ⁺ cells per high power field in v-CTRL and v-A12 mice, respectively, Fig. S7A) but not in CK19⁺ tumor cells (mean 26.6 versus 19 ADAM12⁺/CK19⁺ cells per high power field in v-CTRL and v-A12 mice, respectively, Fig. S7A). The reduction of intratumoral ADAM12⁺ cells (Fig. 4C) upon immunization was associated with a lower tumor collagen content (Fig. 4D).

ADAM12 vaccination increased the total number of splenic T cells, including CD8⁺ T cells and the proportion of CD62L⁻ CD8⁺ effector T cells but not CD4⁺ T cells (Fig. S7B). Of note, the number of intratumoral total T cells and particularly CD8⁺ T cells was increased after ADAM12 vaccination (Fig. 4E), while the number of intratumoral neutrophils was reduced and macrophages remained unchanged (Fig. S7C). Next, we analyzed the intratumoral localization of CD4⁺ and CD8⁺ T cells by means of immunofluorescence. In tumors from v-CTRL mice, about 67 % of CD8⁺ and 53% of CD4⁺ T cells were located at the tumor edge, whereas only 32% and 47%, respectively of the cells infiltrated the tumor center (Fig. S7D). Of note, v-A12 treatment enhanced CD4⁺ and CD8⁺ T cell infiltration into the tumor center (about 66% of CD4⁺ T cells and 64.8% of CD8⁺ T cells), suggesting

a significant redistribution of cytotoxic T cells from the tumor edge towards the intratumoral compartment (Fig. S7D). Consistent with our results from the subcutaneous PDAC model and previous reports on targeting desmoplasia in PDAC (Olive et al., 2009), we observe an increase in vascular density and pericyte coverage in PDACs from v-A12 mice (Fig. 4F). Moreover, v-A12-mediated reduction of desmoplasia was associated with a decrease in tumor hypoxia as assessed by GLUT1 expression (Fig. 4G). The ability of murine syngeneic PDAC tumor cell lines to form liver metastases has been demonstrated in orthotopic models. Consistently, we observe hepatic micrometastasis at the endpoint (Fig. S7E). Noteworthy, reduced desmoplasia along with the vascular normalization upon ADAM12 vaccination was associated with reduced hepatic metastasis of orthotopic PDAC (Fig. 4H).

In summary, our results provide proof-of-concept for the feasibility of reducing tumor desmoplasia in murine PDAC with vaccine-based immunotherapy to target fibroblast-specific transcripts. Of note, we do not target the function of ADAM12 but exploit it as a “tag”, that allows immunotherapeutic depletion of CAFs.

In their experiments, authors show a reduction of tumor size in mice vaccinated for ADAM12. Unfortunately, this effect seems to be restricted to a reduction in CAFs. This would indicate that, although tumors are smaller, there is no real impact in tumor cell viability in vivo. Authors should clearly demonstrate that ADAM12 vaccination reduces tumor cell number or their viability alone or in combination with other treatments. Authors present as relevant for tumor treatment the presence of CD8+ cells in the interior of tumors. The problem is that they also demonstrate that CD8+ T cells from vaccinated mice target preferentially CAFs and do not have an impact on TCs in vivo. It would have been expected that once infiltration is increased, ADMA12-independent cytotoxicity would have increased as well, which would have been a sign of reduced immunosuppression. Does ADAM12+ vaccination reduce immunosuppression in the tumor?

Our reply:

We very much appreciate this comment and would like to clarify that the quantification in Figures S1F and S4A refers to CK19⁺ cells that co-express ADAM12. So, we performed a quantification of total CK19⁺ cells. As shown below, the area of CK19⁺ cells was reduced after v-A12 vaccination in the prophylactic (left) as well as the therapeutic (right) setting, suggesting indeed reduced tumor cell numbers along with a reduction in desmoplasia and tumor size.

Next, we analyzed tumor cell apoptosis by meanings of Caspase-3 immunostaining. Although v-A12 treatment reduces tumor cell proliferation in the prophylactic as well as the therapeutic scheme (see Figures S2B and S4C), v-A12 did not significantly change tumor cell apoptosis.

However, when we treated subcutaneous PDACs on day 29 after the second therapeutic v-A12 vaccination (day 28 after tumor cell injection) with the cytotoxic agent gemcitabine (10 mg/kg), the chemotherapy-induced tumor cell apoptosis was significantly enhanced in the v-A12 cohort.

Combinational therapy

In summary, this data suggests that v-A12 vaccination reduces tumor cell proliferation and total tumor cell numbers. Moreover, in combination with chemotherapy, v-A12 is likely to enhance drug delivery and/or improved responsiveness of the tumor to cytotoxic agents. Both conclusions are compatible with the normalized vasculature and reduced tumor hypoxia that we observe upon v-A12 treatment (figures S3 and S6).

We feel that this data supports the conclusion of the manuscript but would like to leave it to the reviewer and the editor, whether this data should be included in the manuscript.

One of the main causes for drug resistance in PDAC is that desmoplasia reduces irrigation within the tumors, making it impossible for the treatment to arrive to the tumor cells. In this sense, a more relevant experiment would have been to check for tumor perfusion in animal models for PDAC mentioned previously. If perfusion is increased, a mouse model for chemotherapy would be very interesting. Presumably, vaccinated mice should have an increased perfusion, leading to increased chemotherapy-mediated tumor cell death.

Our reply:

We very much appreciate this comment and agree with the reviewer. We assessed tumor perfusion by means of FITC-lectin injection and simultaneous immunodetection of CD31 on tumor sections in the therapeutic setting. As shown below, v-A12 increased the FITC-Lectin/CD31 double-positive area, indicating enhanced tumor perfusion. Moreover, as shown above, when we performed treatment with the cytotoxic agent gemcitabine, the

chemotherapy-induced tumor cell apoptosis was significantly enhanced in the v-A12 cohort. This further suggests enhanced perfusion/drug delivery upon v-A12 treatment.

These results have now been included in the revised manuscript as follows:

Consistent with previous reports on targeting desmoplasia in PDAC (Olive *et al.*, 2009), we observe an increase in vascular density and pericyte coverage in PDACs from v-A12 mice (Fig. S6A and B). Next, we wanted to assess whether the v-A12-related vascular changes translate into altered tumor perfusion by means of FITC-lectin injection and simultaneous immunodetection of CD31 on tumor sections. As shown in Figure S6B, v-A12 increased the Lectin/CD31 double-positive area, indicating enhanced tumor perfusion. Moreover, v-A12-mediated reduction of CAFs was associated with a decrease in tumor hypoxia as assessed by Hypoxyprobe staining as well as the expression of the surrogate marker glucose transporter 1 (GLUT1) (Airley *et al.*, 2003) (Fig. S6C).

The method employed to detect circulating tumor cells is not rigorous enough. To study gene expression in blood as a direct indication of circulating tumor cells could be misleading. On top of it, the genes employed are not clear to be relevant in a chemotherapy free model. In addition, to relativize data to 16S is not the most effective way to detect differences in qPCR analyses, because its expression level is much higher than any target gene. Moreover, the concern in the scientific community is whether the absence of CAFs increases metastatic capacity of tumor cells. If this analysis is made when tumors are similar, it is expected that CAFs presence would also be similar. In this sense, this analysis would be more relevant to be carried out in the treatment model (Figure 3). Nonetheless, the most effective way to show circulating tumor cells would be to inject tagged tumor cells into mice (fluorescent or otherwise) and to find those cells in the blood circulation by flow cytometry.

Our reply:

We would like to thank the reviewer for this comment and would like to point out that we are totally aware of the fact that the method we adopted from Finisguerra et al. (Finisguerra et al., 2015) to assess metastasis in the subcutaneous PDAC is not optimal and at its best a surrogate for circulating tumor cells. As suggested by the reviewer we repeated this experiment in the therapeutic setting and as shown below, the expression of CK19 and OLFM4 as a surrogate marker for circulating tumor cells were found to be similar in v-CTRL and v-A12 PDAC-bearing mice.

These results have now been included in Figure S6 of the revised manuscript as follows: The expression of CK19 and OLFM4 as a surrogate marker for circulating tumor cells were found to be similar in v-CTRL and v-A12 PDAC-bearing mice (Fig. S6D), suggesting that vaccination with v-A12 does not lead to increased metastasis. In summary, we demonstrate that our vaccination approach can effectively reduce intratumoral fibrosis and tumor growth in a murine model of PDAC in a therapeutic setting.

Finally, the reduced metastasis upon v-A12 in the orthotopic PDAC model (shown above) suggest that metastasis is rather prevented than enhanced.

Minor points:

In general, immunofluorescence images would gain clarity if they would include zoomed-out fields of the merged colors, and zoomed-in fields of single colors. For example, panel S1B shows a nice depiction of the whole tumor, it would have been

interesting to see a similar image of the vaccinated animals compared to the control littermates. In particular, this is essential when authors speak about edge and interior of tumors.

Our reply:

We apologize for this oversight and appreciate the reviewer's suggestion. We have improved the presentation of immunofluorescence images throughout the manuscript as requested by the reviewer.

The purpose of Panel S1B was to show that the subcutaneous PDAC is quite desmoplastic/fibrotic with some ductal structures, and hence a limited but still suitable model to study desmoplasia and to some extent also morphology. We agree that it would be interesting to show whole tumor tissue images from vaccinated and unvaccinated mice. Therefore, we have now included pictures from Sirius red/Fast Green-stained PDAC whole tumors from v-CTRL and v-A12 mice in Figure S1B of the revised manuscript.

Authors mention in the material methods the use of two different ADM12 antibodies. It should be indicated in each figure which antibody was employed, given that the one from Thermofisher recognizes the long form of ADAM12, while the one from Novus Biologicals recognizes short and long forms.

Our reply:

We have now indicated in each figure as well as in the "Material and Methods" section, which specific ADAM12 antibody was used in the respective experiment.

For Figure S1D, in the text authors speak about % of ADAM12 positivity, which does not

correspond with the represented data. It shows the % of SMA+, CK19+ or PDGFRb+ cells that are ADAM12+. It would be more interesting to see, as described in the results, which % of ADAM12+ cells are fibroblasts (SMA+ or PDGFRb+), or tumors (CK19). To measure the empty space in a tumor is not a convincing parameter to imply tubular morphology. Authors should consult with a pathologist to include a more reliable measurement.

Our reply:

We would like to apologize for the confusion. The graph in Fig. S1D shows indeed which % of ADAM12+ cells are fibroblasts (α -SMA+ or PDGFR β +), or tumors (CK19). The labeling of the graph has been changed accordingly. Hence the correct description of the data is the following: The ADAM12+ cells within PDACs consisted of CAFs (42% α -SMA+ and 14% PDGFR- β +) as well as tumor cells (39.4% CK19+). The wording has been changed accordingly in the revised manuscript.

Moreover, we would like to point out that we did not just measure “empty space” but the area that is covered by luminal structures as follows:

To identify the lumina area, we randomly selected 5 areas (1000 x 1000 pixels) of each PDAC tumor tissue stained with Sirius Red/Fast Green-stained, and Ilastik was applied to segment tumor tissue compartments: tumor cells (green), tumor extracellular matrix (red), and tubule area (blue). The proportion of the tubule area was determined by calculating

the tubule area (tubular length ranges above 40 pixels) within the analyzed images via ImageJ.

This description has now been included into the "Methods" section of the manuscript.

Flow cytometric analyses are not rigorous. For instance, in figure S3A, according to the density plot DCM vs CD45, there must be a problem of over-compensation between fluorophores. In addition, in the gating strategy, it is not clear whether arrows depict selected or excluded populations. Moreover, the positive selection gates often include negatively stained cells (i.e: monocyte, F4/80+ or CD44+/CD62L- gating)

Our reply:

We would like to thank the reviewer for this comment. We have adjusted the gating accordingly as shown below. The respective graphs have now been included in Fig 2D and E, Fig S2C and S3A of the revised manuscript.

Fig. 2D

We defined effector T cells as CD62L⁻ T cells, as shown in the FACS gating below (Fig. 2D and E), and naïve T cells are identified with CD62L⁺CD44⁻ T cells, as shown in the detailed FACS gating strategy, Fig S2C.

Fig. 2D

Fig. 2E

From the manuscript: we detected an increase in effector and effector memory T cells (defined as CD62L⁻ effector T cells) along with a decrease in naïve T cells (CD62L⁺CD44⁻) within the CD4⁺ (Fig. 2D) and CD8⁺ (Fig. 2E) T cell populations after ADAM12 vaccination.

Gating strategy in Figure S2C

Myeloid cell gating in Figure S3A

Neutrophils are identified with: CD45⁺ CD11b⁺ Ly6G⁺ cells. monocytes were gated based on non-neutrophils (negative gating), as CD11b⁺Ly6C⁺ population. Then, monocyte macrophages were further defined with an F4/80⁺ signal.

Authors exaggerate in their wording. The word "massively" is used in excess, and to describe differences that are rather moderate.

Our reply:

We agree with the reviewer and have removed the term massively from the manuscript.

Glut1 to imply hypoxia is very limited. There are many other methods, much more reliable, to do so.

Our reply:

We agree that GLUT-1 represents a surrogate marker for tumor hypoxia. Therefore, we have assessed the degree by means of the oxygen-sensitive dye Pimonidazole/Hypoxyprobe in the subcutaneous PDAC after therapeutic vaccination. After injection into mice the dye undergoes a conformational change in tissues with a pO₂ of below 10 mmHg and can then be detected with an antibody and by means of immunostaining. As shown below, v-A12 vaccination results in reduced Hypoxyprobe staining, suggesting indeed reduced tumor hypoxia.

The data has now been included in Figure S6C and described as follows:

Moreover, v-A12-mediated reduction of CAFs was associated with a decrease in tumor hypoxia as assessed by Hypoxyprobe staining as well as the expression of the surrogate marker glucose transporter 1 (GLUT1) (Airley et al., 2003) (Fig. S6C).

5th Jun 2024

Decision on your manuscript EMM-2023-18859-V2-Q

Dear Prof. Stockmann,

Thank you for the submission of your revised manuscript to EMBO Molecular Medicine. We have now received the feedback from the two initial referees who re-reviewed your manuscript.

As you will see from the enclosed reports, while referee #2 is satisfied with the revisions, referee #3 still raises major concerns on your work and regrets that the revisions did not satisfactorily address the initial comments. EMBO Press encourages a single round of experimental revisions. Based on referee #3 major concerns, given also that this manuscript had been previously rejected, I am afraid that we cannot offer to consider your manuscript further.

I am very sorry to disappoint you again. Still, I hope that the referees' comments will be helpful in your continued work in this area.

Yours sincerely,

Lise Roth

Lise Roth
Senior Editor
EMBO Molecular Medicine

***** Reviewer's comments *****

Referee #2 (Comments on Novelty/Model System for Author):

The stromal barrier in PDAC is a notorious problem (poor infiltration of drugs and T cells). Vaccination against ADAM12 may dissolve this problem.

Referee #2 (Remarks for Author):

Yes I would recommend to publish this as a full research article

Referee #3 (Comments on Novelty/Model System for Author):

The authors have already published one article showing that this vaccination targets activated fibroblasts in a fibrotic environment. Our previous review tried to increase the novelty of their study, by answering to the controversy in the field on whether CAFs benefit or impair tumor development. We made several suggestions to include important experiments to move in this direction, but they have not performed them, or they include them only in their point-by-point response.

Referee #3 (Remarks for Author):

Authors have significantly improved the manuscript. However, the main concern pointed out in the previous review still remains, authors do not demonstrate a benefit for ADAM12 vaccination as a cancer therapy. Tumor cells are not affected by the vaccine and it has not been assessed the effect on tumor progression. Moreover, although authors show the amount of CK19+ cells as percentage of area (only in their point-by-point response). These data are not relevant, authors should have calculated the percentage of tumor cells. This analysis would give an idea of whether vaccination has an impact on tumor growth. Although authors have included an orthotopic model, the vast majority of the results are restricted to subcutaneous tumors. Moreover, the methodology for orthotopic injection does not seem to be adequate. Orthotopic injection of 7.5×10^5 cells in 30 μ l is exaggerated. Just as an example, in PMID: 38114498 only 1000 cells were injected. Indeed, authors injected less cells subcutaneously, 5×10^4 , which is very odd. Orthotopic PDAC tumors are composed roughly of three main things: tumor cells, CAFs and ECM. Authors claim that all three

components are reduced, but the cell density, according to their DAPI staining, is similar. What is the composition of tumors in vaccinated mice?

Authors show that vaccination improved vessel perfusion (lectin-GFP) only in s.c. tumors. In their point-by-point response, they show that vaccination increased GEM-mediated Caspase 3, but it is not clear whether or not it is cleaved, which is the active form. In addition, they do not mention anything regarding tumor reduction (tumor volume, animal survival, etc). What is the morphology with normal H&E staining? Have the authors performed any other analysis in these samples? This is an experiment that, if made in the orthotopic model, would really add important knowledge to the current state-of-art. Authors should have performed many more analyses in this setting and in orthotopic tumors.

Although authors convincingly demonstrated that vA12 reduces ADAM12+ CAFs also in the orthotopic model, they did not present any data showing a general reduction of CAFs in any of the models. These data could be obtained by the quantification of SMA and PDGFRb positive cells per field, with the addition of a general marker for fibroblasts like desmin. Without these data, authors cannot state that vaccination targets CAFs in a general way.

In the previous review, it was pointed out the difficulty to assess data rigor, but authors have not made enough effort to solve this concern:

- In the previous review, it was mentioned that: "immunofluorescence images would gain clarity if they would include zoomed-out fields of the merged colors, and zoomed-in fields of single colors". Authors now present even more zoomed-in merged pictures, but only one zoomed-out picture has been added. This zoomed out picture is the Sirius Red staining in Figure 1. It shows why it is necessary to show the whole picture, because it does not look like the treated tumors have less collagen. Moreover, in the methods section it is stated that it was analyzed the whole tumor tissue section. This means that also the border of the sample was quantified. When looking at the full picture shown in Fig S1B one can see that in the vaccinated tumor shown the sample is broken and it misses a big part of this cover, with a high collagen content. If this was not considered, it would influence the final result. Assessment of Sirius red should be calculated as average staining area per 10HPF per tumor, this is standard in the field. If this information would have been present in the previous version, this suggestion would have been made before.

- It was mentioned that: "in the gating strategy, it is not clear whether arrows depict selected or excluded populations. Moreover, the positive selection gates often include negatively stained cells". The concern with the gating was the uncertainty of the population selected, because it included also cells negative for the gating (example F4/80 macrophages). It is positive that authors have acknowledged this in Fig S2C and S3A, where the gating has been modified. However, even after changing the gating, the numbers represented in the graphs (Fig. 2A, 2D, 2E, S3A, S5A, B, E&F), remain the same as in the previous version. It is puzzling how changing the gate in flow cytometry would not affect the numbers obtained in the analysis.

- Fig S1: In the previous version these values were 57, 14 and 50% in the text and it is evident, although with a different scale, that the current figure and the previous one do not use the same data. What has been modified in the analysis? Why there has been the need for a new quantification?

- Fig. S1F: This version includes one more sample than in the previous one. However, there is no mention of the reason behind this new sample addition.

- Fig. S4C: the quantification of CK19+/KI67+ cells has been modified, without an explanation or an addition of animals. In previous Fig S4C data were non-significant ($p=0.0528$), in the new Fig. S4C data are significant ($p=0.0198$). What was the need for performing a new quantification, considering that quantifications in Fig. S2A, B and S4B have not been modified?

- Fig 4H, S7E: Given that no specific staining for tumor cells has been performed to characterize the metastasis (i.e. CK19), the analysis is not solid enough. Without any other data, the picture shown in Fig S7E could even be an infection of Helicobacter (see an example in PMID: 25184625). Therefore, Fig. 4H is not relevant.

- Fig. S6D: experiments seem to be performed only on six animals. Why is this n so low? Considering that in the same figure (Fig S6C), presumably obtained from the same mice, they show data collected in more than 12 mice, It is necessary to explain the criteria for the selection of those six mice. Authors evaluated metastasis in the therapeutic setting, but using the same method as in previous version, even though it was pointed out its lack of rigor. In addition, OLFM4 expression is present in 1 out of 6 control mice, while in vaccinated mice it is present in 4 out of 6. It is evident that vaccinated mice have higher presence of these marker. So, if authors consider OLFM4 expression in blood enough to evaluate metastasis, they should have concluded that metastasis is promoted. Have authors checked other distant organs to search for metastatic foci?

Fig S2A S4B: Given that the vaccination reduces the number of ADAM12+ cells, it cannot be claimed that those cells proliferate less using a % of total cells. To see the ratio of proliferation of a cell type (ADAM12+cells), it would be appropriate to quantify the percentage of ADAM1+ cells that are KI67+ and not the % of total cells that are ADAM12+/KI67+.

Fig S2B&S4C: In the image that correspond to the immunized mice there are many cells that are negative for CK19. What are those cells? Do tumor cells lose the CK19 marker? Is there an increase of ADAM12- fibroblasts?

As a service to authors, EMBO provides authors with the possibility to transfer a manuscript that one journal cannot offer to publish to another EMBO publication. The full manuscript and if applicable, reviewers reports are automatically sent to the receiving journal to allow for fast handling and a prompt decision on your manuscript. For more details of this service, and to transfer your manuscript to another EMBO title please click on Link Not Available

5th Jun 2024

Decision on your manuscript EMM-2023-18859-V2-Q

Dear Prof. Stockmann,

Thank you for the submission of your revised manuscript to EMBO Molecular Medicine. We have now received the feedback from the two initial referees who re-reviewed your manuscript.

As you will see from the enclosed reports, while referee #2 is satisfied with the revisions, referee #3 still raises major concerns on your work and regrets that the revisions did not satisfactorily address the initial comments.

EMBO Press encourages a single round of experimental revisions. Based on referee #3 major concerns, given also that this manuscript had been previously rejected, I am afraid that we cannot offer to consider your manuscript further.

I am very sorry to disappoint you again. Still, I hope that the referees' comments will be helpful in your continued work in this area.

Yours sincerely,

Lise Roth
Senior Editor
EMBO Molecular Medicine

Reviewer's

comments

Referee #2 (Comments on Novelty/Model System for Author):

The stroll barrier in PDAC is a motor problem (poor infiltration of drugs and T cells). Vaccination against ADAM12 may dissolve this problem.

Referee #2 (Remarks for Author):

Yes I would recommend to publish this as a full research article.

Referee #3 (Comments on Novelty/Model System for Author):

The authors have already published one article showing that this vaccination targets activated fibroblasts in a fibrotic environment. Our previous review tried to increase the novelty of their study, by answering to the controversy in the field on whether CAFs benefit or impair tumor development. We made several suggestions to include important experiments to move in this direction, but they have not performed them, or they include them only in their point-by-point response.

OUR REPLY:

We appreciate Reviewer 3's concern about the lack of novelty of our approach, however, we believe our work makes a valuable contribution to the area of cancer immunotherapy, by demonstrating the benefit of a targeted vaccination approach against this marker in reducing tumor burden. Furthermore, our work demonstrates the therapeutic benefit of vaccination against a fibroblast marker that is not a known driver of disease, unlike previous approaches targeting FAP or TGF β , and opens the field to exploit the "tagging" function of other potential antigenic targets in the proteome of pro-fibrotic fibroblasts (Ref 1: PMID: 36090474, Ref 2: PMID: 36600556). This builds on our previous work showing the efficacy of ADAM12 vaccination in different fibrotic disease models and makes the non-trivial translation to a more complex desmoplastic tumor model with entirely distinct disease pathogenesis. Additionally, we were happy to take on board the suggestion to explore further potential benefits of CAF reduction such as "increased drug delivery within the tumor". To this end, we have shown that ADAM12 vaccination restores tumor vascular perfusion using FITC-lectin injection and tested ADAM12 vaccination on an orthotopic KPC PDAC model.

Moreover, we would like to emphasize that we are not trying to hide data and, therefore, clearly stated in the point-by-point response that "we would like to leave it to the reviewer and the editor, whether this data should be included in the manuscript."

Reviewer 3: Authors have significantly improved the manuscript. However, the main concern pointed out in the previous review remains, authors do not demonstrate a benefit for ADAM12 vaccination as a cancer therapy. Tumor cells are not affected by the vaccine and it has not been assessed the effect on tumor progression. Moreover, although authors show the amount of CK19+ cells as percentage of area (only in their point-by-point response). These data are not relevant, authors should have calculated the percentage of tumor cells. This analysis would give an idea of whether vaccination has an impact on tumor growth.

OUR REPLY:

We have presented the tumor growth inhibition by reduced volumes in both subcutaneous and orthotopic PDAC mouse models (Fig. 1C, 3C, and 4B). Correspondingly, we have presented a reduction of proliferating CK19⁺ cells in the subcutaneous PDAC model in our manuscript (Fig. S2B and S4C) (PMID: 28283655). We would also like to

mention that we observed a significant reduction in the total population of Ki67⁺ cells, ADAM12⁺Ki67⁺ cells, and CK19⁺Ki67⁺ cells in the subcutaneous PDAC model (Please see below).

Prophylactic ADAM12 vaccination

Therapeutic ADAM12 vaccination

Additionally, we observed fewer PanCK⁺ cells (tumor cells: CD45⁻/CD31⁻/PanCK⁺ cells) in subcutaneous PDAC tumors from therapeutic ADAM12 vaccinated mice (n=6) via FACS analysis, indicating therapeutic ADAM12 vaccination indeed prevented the total population of tumor cells (Please see below). (Though this experiment is only accessible in the subcutaneous PDAC tumor with therapeutic ADAM12 vaccination.)

Gating strategy

(anti-CD45 antibody: Ref: BD564225, anti-CD31 antibody: Ref: BD741892 anti-PanCK antibody: Ref: Novus Bio NBP2-33200PE)

Reviewer 3: Although authors have included an orthotopic model, the vast majority of the results are restricted to subcutaneous tumors. Moreover, the methodology for orthotopic injection does not seem to be adequate. Orthotopic injection of 7.5×10^5 cells in 30 ul is exaggerated. Just as an example, in PMID: 38114498 only 1000 cells were injected. Indeed, authors injected less cells subcutaneously, 5×10^4 , which is very odd.

OUR REPLY:

For the orthotopic PDAC mouse model: The remark concerning the number of KPC cells injected orthotopically is surprising. Unfortunately, we cannot reference our cell line since we have generated our cell line from KPC mice and have not published yet on that. Kevin Thierry's paper using orthotopic injection is in review (not published yet). Several reports show injection of pancreatic cancer cells from 5×10^4 to 2.5×10^5 probably depending on the aggressiveness of the cell line. We have adapted our protocol in order to detect sufficient tumor development in 10 days (100 mm³). Indeed injection of fewer cells leads to the development but on longer time courses (30 days for example to establish the tumor). Showing that the orthotopic grafts present the components expected tumor cells, CAFs, immune cells, and ECM should argue in favor of that model is pertinent and the number of cells injected is a question of fine-tuning of the protocol in order to reduce the whole protocol to 23 days. (From our collaborator: Dr. Ana Hennino)

For the subcutaneous PDAC mouse model: We did a pilot study with different amounts of cells (5×10^4 , 10^5 , and 2×10^5 cells) before applying 5×10^4 cells for the subcutaneous study. The idea subcutaneous PDAC mouse model should have desmoplastic PDAC tumor morphology. Indeed, PDAC tumors grow faster and bigger at the endpoint with more cells injected, while this sacrifices tumor desmoplasia development. We observed that 5×10^4 KP2 cells led to a good desmoplastic PDAC tumor morphology (Fig. S1B).

Moreover, we would like to stress the point that animal experimentation in the EU as well as in Switzerland is strictly regulated and we have to adhere to the experimental protocol that has been approved by the ethics committees and are not allowed to change the number of injected cells or the duration of the experiment.

Reviewer 3: Orthotopic PDAC tumors are composed roughly of three main things: tumor cells, CAFs and ECM. Authors claim that all three components are reduced, but the cell density, according to their DAPI staining, is similar. What is the composition of tumors in vaccinated mice?

OUR REPLY:

We show that the tumor volume was reduced. This reduction in tumor size is not necessarily reflected by a change in cell density (number of cells per high power field), whereas it's likely due to the reduction of different cell types, particularly tumor cells and CAFs, within the tumor. Our findings show that the orthotopic PDACs are indeed composed of tumor cells, CAFs, endothelial cells, pericytes, as well as various immune cell subsets including myeloid cells (e.g., neutrophils and macrophages) and T cells. Their corresponding population in orthotopic PDAC tumors has been shown in Fig 4E ($CD4^+$ T cell and $CD8^+$ T cell population), S7D ($CD4^+$ / $CD8^+$ T cell localization), S7C (neutrophils and macrophages).

Reviewer 3: Authors show that vaccination improved vessel perfusion (lectin-GFP) only in s.c. tumors.

OUR REPLY:

We are aware that the perfusion experiment with FITC-lectin was only performed in the subcutaneous model, as it was not explicitly requested in the previous correspondence to perform the FITC-lectin assay in all models including the orthotopic model. The reviewer has initially requested to test whether the vascular phenotype that we describe in the subcutaneous model translates into changes in tumor perfusion and we feel that we have addressed this question.

Reviewer 3: In their point-by-point response, they show that vaccination increased GEM-mediated Caspase 3, but it is not clear whether or not it is cleaved, which is the active form.

OUR REPLY:

We apologize for this oversight and we agree that a staining for non-cleaved Caspase 3 would not make sense. The apoptosis marker, cleaved-Caspase 3 was stained with the anti-cleaved Caspase 3 antibody (CST, CAS: 9661) in our study. As we mentioned in the previous point-by-point response, we wrote that we observed **increased apoptosis**, which should be as a result of cleaved Caspase 3.

Reviewer 3: In addition, they do not mention anything regarding tumor reduction (tumor volume, animal survival, etc). What is the morphology with normal H&E staining? Have the authors performed any other analysis in these samples? This is an experiment that, if made in the orthotopic model, would really add important knowledge to the current state-of-art. Authors should have performed many more analyses in this setting and in orthotopic tumors.

OUR REPLY:

All mice bearing KP2 cells survived after therapeutic ADAM12 vaccination and GEM treatment. As already mentioned above, due to the local ethics regulations of animal experiments, we are not allowed to increase the endpoint and the total duration of the experiment or even study the "survival" of mice. So we had to end the experiment 4 days after the treatment of GEM which did not allow us to assess for differences in tumor volume. Yet, we were able to demonstrate increased tumor cell death upon chemotherapy as requested by the reviewer.

We would like to point out that reviewer 3`s comments from the 1st round were as follows:

"One of the main causes for drug resistance in PDAC is that desmoplasia reduces irrigation within the tumors, making it impossible for the treatment to arrive to the tumor cells. In this sense, a more relevant experiment would have been to check for tumor perfusion in animal models for PDAC mentioned previously. If perfusion is increased, a mouse model for chemotherapy would be very interesting. Presumably, vaccinated mice should have an increased perfusion, leading to increased chemotherapy-mediated tumor cell death."

Our understanding is we should check tumor perfusion first in the subcutaneous PDAC mouse model. If tumor perfusion is increased, we will observe increased chemotherapy-mediated tumor cell death. We have shown both vascular perfusion (therapeutic ADAM12 vaccination, Fig. S6B) and cleaved-Cas3 staining results (point-by-point response) upon combination chemotherapy.

Likewise for another comments Reviewer 3`s comments from 1st round:

"Addition of an adequate model system (GEMM or orthotopic); authors should clearly demonstrate that ADAM12 vaccination reduces tumor cell number or their viability alone or in combination with other treatment."

Our understanding is: We were supposed to show either GEMM or an orthotopic model, and we showed an orthotopic KPC PDAC model (Fig. 4, and S7). As for the second suggestion, we could show either reduced tumor cell number or reduced viability, or improved combination therapy like GEM. We showed cleaved Cas3 staining results of therapeutic ADAM12 vaccination alone and with GEM treatment (point-by-point response).

Finally, it was not mentioned to perform HE staining to observe the tumor morphology in the previous round, but we are of course more than happy to provide such images in a revised version.

Reviewer 3: Although authors convincingly demonstrated that vA12 reduces ADAM12+ CAFs also in the orthotopic model, they did not present any data showing a general reduction of CAFs in any of the models. These data could be obtained by the quantification of SMA and PDGFRb positive cells per field, with the addition of a general marker for fibroblasts like desmin. Without these data, authors cannot state that vaccination targets CAFs in a general way.

OUR REPLY:

This issue was not mentioned in the first round of reviews. We observe a decrease in α -SMA-positive cells (% of the positive area) in both the prophylactic ($p=0.0224$) and therapeutic vaccination ($p=0.0348$) settings (Please see below). However, we prefer the quantification of tissue collagen content as the more functionally relevant readout not only of CAF abundance but also activity and collagen production.

The following paper used α -SMA representing CAFs (Fig. 1e and h) (Ref: PMID: 37898598)

Reviewer 3: In the previous review, it was pointed out the difficulty to assess data rigor, but authors have not made enough effort to solve this concern:

- In the previous review, it was mentioned that: "immunofluorescence images would gain clarity if they would include zoomed-out fields of the merged colors, and zoomed-in fields of single colors". Authors now present even more zoomed-in merged pictures, but only one zoomed-out picture has been added. This zoomed out picture is the Sirius Red staining in Figure 1. It shows why it is necessary to show the whole picture, because it does not look like the treated tumors have less collagen.

OUR REPLY:

Fig. S1D: We apologize for the misunderstanding, now the representative pictures are shown as follows:

(scale bar in the zoom-out picture: 200 μ m, scale bar in the enlarged picture: 50 μ m)

As for the collagen deposition, please see the color-deconvolution pictures in the next question.

Reviewer 3: Moreover, in the methods section it is stated that it was analyzed the whole tumor tissue section. This means that also the border of the sample was quantified. When looking at the full picture shown in Fig S1B one can see that in the vaccinated tumor shown the sample is broken and it misses a big part of this cover, with a high collagen content. If this was not considered, it would influence the final result. Assessment of Sirius red should be calculated as average staining area per 10HPF per tumor, this is standard in the field. If this information would have been present in the previous version, this suggestion would have been made before.

OUR REPLY:

We described how we performed the collagen quantification in the methods and materials as follows:

Collagen content quantification was carried out by analyzing the Sirius Red signal via Fiji (ImageJ 1.54f) on RGB images as follows: "Image/Color/Color Deconvolution" was applied firstly with user-defined values corresponding to the staining set to determine the percentage of collagen deposition area (Sirius Red positive area) within the whole tumor tissue section (Fig 1E, Fig 3E and Fig 4D). Then, an equal threshold setting was uniformly applied to a set of samples.

To answer reviewer 3's concern: we tried to collect intact tissues as much as possible, and tumors were evenly divided into two parts for FACS analysis and histology analysis. The original pictures representing v-CTRL and v-A12 are shown below, we didn't intentionally remove any big part of the tumor tissue, except fat and loose connective tissue. Then the area within the yellow curve was quantified.

v-CTRL:

562x4073 pixels; RGB; 68MB

2021_03_08_10/LL.czi - 2021_03_08_10/LL.czi #03.jpg - (Colour_1) (16.7%)

4362x4073 pixels; 8-bit; 17MB

v-A12:

We would be happy to reanalyze the tumor sections with 10 HPF per section. However, we feel that quantifying collagen with the slide scanner is eventually more accurate than the chosen HPFs.

Reviewer 3: It was mentioned that: "in the gating strategy, it is not clear whether arrows depict selected or excluded populations. Moreover, the positive selection gates often include negatively stained cells". The concern with the gating was the uncertainty of the population selected because it included also cells negative for the gating (example F4/80 macrophages). It is positive that authors have acknowledged this in Fig S2C and S3A, where the gating has been modified. However, even after changing the gating, the numbers represented in the graphs (Fig. 2A, 2D, 2E, S3A, S5A, B, E&F), remain the same as in the previous version. It is puzzling how changing the gate in flow cytometry would not affect the numbers obtained in the analysis.

OUR REPLY:

We apologize for the confusion. The prophylactic vaccination experiment was performed 3 years ago, and the old gating strategy was performed. In response to reviewer 3's comments, we applied the new gating strategy that was used in the therapeutic setting to the prophylactic setting. Since the general, relative abundances of the different populations and the overall phenotype were not significantly changed with new gating, we did not update the graphs. We apologize for this oversight, we would like to provide the updated data below.

Neutrophil and macrophages population in the prophylactic ADAM12 vaccination setting, based on new gating:

The corresponding results as in Fig S3A:

Appendix Fig S3

A

We also apologize for the confusion concerning the results of spleen effector T cells in both prophylactic and therapeutic settings. In the first version, the results of effector/memory T cells consisted of CD44⁻CD62L⁻ T cell and CD44⁺ CD62L⁻ T cell population. In response to Reviewer 3's suggestion, we re-gated CD44⁻CD62L⁻ T cell and CD44⁺ CD62L⁻ T cell as CD62L⁻ T cells, without changing naïve T cells. Similarly, the results from the new gating are comparable to previous ones (please see below). Therefore, we didn't change the previous results.

Prophylactic ADAM12 vaccination – Spleen FACS analysis of effector CD4⁺/CD8⁺ T cells:

Corresponding results in Fig. 2D and E:

Therapeutic ADAM12 vaccination – Spleen FACS analysis of effector CD4⁺/CD8⁺ T cells:

Corresponding results in Fig. S5E and F:

We apologize that n value is 8 in v-CTRL and 9 in v-A12 in the new CD62L-CD4⁺ effector T cells result. Since there were 2 mice in the control group and 1 mouse in the v-A12 group were unsuccessfully vaccinated (leaking during vaccination) and should be removed.

• Fig S1: In the previous version these, values were 57, 14, and 50% in the text and it is evident, although with a different scale, that the current figure and the previous one do not use the same data. What has been modified in the analysis? Why there has been the need for a new quantification?

OUR REPLY:

In the previous version, the total percentage (57%+14%+50%) was above 100%, therefore we adjusted the threshold.

• Fig. S1F: This version includes one more sample than in the previous one. However, there is no mention of the reason behind this new sample addition.

OUR REPLY:

We apologize that we didn't mention it in the previous point-by-point letter. One sample in A12⁺/CK19⁺ (v-A12) was excluded in the first version. Since in our previous quantification process, this sample was first excluded based on % of ADAM12⁺ cells.

Following more detailed analyses (such as the number of cells per HPF), we decided only to show data according to the number of cells per HPF. Upon this analysis, the sample was not excluded as a significant outlier and we included this sample in the A12⁺/KI67⁺ (v-A12) group.

In the 1st submission:

(Q value was presented multiple t-tests)

In the 2nd submission:

(P value was presented based on multiple t-tests)

- Fig. S4C: the quantification of CK19⁺/KI67⁺ cells has been modified, without an explanation or an addition of animals. In the previous Fig S4C data were non-significant (p= 0.0528), in the new Fig. S4C data are significant (p= 0.0198). What was the need for performing a new quantification, considering that quantifications in Fig. S2A, B and S4B have not been modified?

OUR REPLY:

There was an outlier within one sample in the v-A12 group. Excluding this value resulted in a significant change in the mean value of CK19⁺/KI67⁺ of this sample from 10.7299 to 5.6302, thus the mean value of CK19⁺/KI67⁺ of v-A12 group changed from 2.64448 to 1.79452, which impacts the p-value of the data.

The outlier detection was performed on Prism:

(Prism Analyze data/Column analyses/Identify outliers/Method-ROUT (Q=0.5%)).

If the reviewer feels we should not exclude this outlier within this sample in the v-A12 group, we will be happy to keep the original figure.

- Fig 4H, S7E: Given that no specific staining for tumor cells has been performed to characterize the metastasis (i.e. CK19), the analysis is not solid enough. Without any other data, the picture shown in Fig S7E could even be an infection of Helicobacter (see an example in PMID: 25184625). Therefore, Fig. 4H is not relevant.

OUR REPLY:

We would like to mention that it is not at all unusual to use HE stainings on liver sections to characterize metastasis (Ref: PMID: 32183949).

The idea raised by the reviewer that we eventually quantified a Helicobacter infection is certainly interesting, but we would like to emphasize that Helicobacter is not present in the animal facility and has not been present in the past. However, in order to address this concern, we would be happy to additionally perform CK19 stainings to assess liver metastasis.

- Fig. S6D: experiments seem to be performed only on six animals. Why is this n so low? Considering that in the same figure (Fig S6C), presumably obtained from the same mice, they show data collected in more than 12 mice, It is necessary to explain the criteria for the selection of those six mice. Authors evaluated metastasis in the therapeutic setting, but using the same method as in previous version, even though it was pointed out its lack of rigor. In addition, OLFM4 expression is present in 1 out of 6 control mice, while in vaccinated mice it is present in 4 out of 6. It is evident that vaccinated mice have higher presence of these marker. So, if authors consider OLFM4 expression in blood enough to evaluate metastasis, they should have concluded that metastasis is promoted. Have authors checked other distant organs to search for metastatic foci?

OUR REPLY:

The blood was collected only from the mice that were used for FITC-lectin injection (Fig. S6B) in order to address this in the therapeutic vaccination setting and only 6 mice for each group were enrolled in this experiment.

Once more, we would like to point out that we did not make up this method and would like to refer to the publication by Finisguerra et al (Ref: PMID: 25985180). in which the method has been established. Moreover, we are totally aware of the fact that this method to assess metastasis in the subcutaneous PDAC is not optimal and at its best a surrogate for

circulating tumor cells. As the result in Figure S6D is not statistically significant, we cannot conclude that metastasis is promoted. Finally, if the reviewer and the editor feel that the method is inaccurate, we are happy to remove the PCR analysis from the manuscript.

Fig S2A S4B: Given that the vaccination reduces the number of ADAM12+ cells, it cannot be claimed that those cells proliferate less using a % of total cells. To see the ratio of proliferation of a cell type (ADAM12+ cells), it would be appropriate to quantify the percentage of ADAM1+ cells that are KI67+ and not the % of total cells that are ADAM12+/KI67+.

OUR REPLY:

We acknowledge that it would be reasonable to alternatively use % of ADAM12+ cells instead of % of total cells, although this was not mentioned in the first round of reviews.

We have observed a decrease in the absolute number of Ki67+ cells, ADAM12+Ki67+ cells, and CK19+KI67+ cells in both prophylactic and therapeutic v-A12 groups (see figures below). This shows that vaccination against ADAM12 is effective in reducing the absolute number of ADAM12+ Ki67+ cells in the tumor, which is a more pertinent observation than a change in proliferation expressed as a percentage of ADAM12+ cells, respectively. If the total number of double-positive cells is still not sufficiently convincing, we are happy to quantify the percentage of proliferating ADAM12+ cells.

Prophylactic ADAM12 vaccination

Therapeutic ADAM12 vaccination

Fig S2B&S4C: In the image that corresponds to the immunized mice there are many cells that are negative for CK19. What are those cells? Do tumor cells lose the CK19 marker? Is there an increase of ADAM12⁻ fibroblasts?

OUR REPLY:

Although this question was not mentioned in the first round, it is indeed an important question. CK19 is a widely accepted marker of tumor epithelial-like cells in PDAC, and as the reviewer correctly suggests, not all tumor cells are CK19⁺ cells. We did observe fewer CK19⁺ cells in tumors from the vaccinated group, and we hypothesized that ADAM12 vaccination might selectively affect the CK19⁺ cell population. Besides, ADAM12 vaccination might also change the tumor microenvironment, which may affect the expression of CK19 in tumor cells, which are quite heterogeneous. It should have been tested whether these CK19⁻ cells are tumor cells with other markers like CK7, E-cadherin, or CK20.

Concerning ADAM12⁻ fibroblasts, this was not mentioned in the first round either. As per our previous answer earlier in this rebuttal, we see a decrease in the overall α -SMA⁺ CAF area in the v-A12 group and reduced collagen deposition, both of which demonstrate an overall reduction in fibroblast number and activity. The overall decrease in CAFs, and not the change in ADAM12⁻ CAFs, is the relevant readout in this experiment. Even if there is an increase in the number of ADAM12⁻ CAFs, as may happen due to changes in fibroblast subtype specification upon ADAM12 treatment, the relevant take-home message is that the overall CAF population as well as the target ADAM12⁺ population is reduced, and this, in turn, reduces disease burden. If desired, we are happy to incorporate this in more detail in the discussion section of the manuscript.

1st Jul 2024

Dear Prof. Stockmann,

Thank you for your e-mail asking us to reconsider our decision on your manuscript and for providing a detailed provisional point-by-point letter, and please accept my apologies for the delay in getting back to you as I was waiting to get further advice on your manuscript. I have now received the feedback from an independent expert in the field. As you will see below, he/she is overall supportive of your study:

"I have reviewed the comments by the authors and referees, and I am in quite solid agreement with referee #2 and supportive of the authors and the manuscript. While some clarification of the flow cytometry results across different gating strategies as well as tumor growth kinetic measurements would be helpful from the authors prior to publication, the other comments by referee #3 do not preclude publication in my view. The comment "the most common protocol is to inject 1000 cells" is sometimes true but certainly an oversimplification, as growth kinetics of pancreatic cancer cell lines vary immensely across models. The KP2 model used in this manuscript is on the slower-growing end and, as used by the lab which generated this line, is used typically by injecting 200,000 cells into the pancreas (many examples but one is PMID: 31076405)."

Based on this feedback, and after further discussion within the team, we would welcome the submission of a revised manuscript that would adequately address the remaining comments from the referee as per your provisional rebuttal letter, including clarifications on the flow cytometry results as well as on tumor growth kinetics measurements.

The revised manuscript will be sent to our advisor. EMBO Molecular Medicine encourages a single round of revision only and therefore, acceptance or rejection of the manuscript will depend on the completeness of your responses included in the next, final version of the manuscript.

We are expecting your revised manuscript within three months, if you anticipate any delay, please contact us.

We require:

- 1) A .docx formatted version of the manuscript text (including legends for main figures, EV figures and tables). Please make sure that the changes are highlighted to be clearly visible.
- 2) Individual production quality figure files as .eps, .tif, .jpg (one file per figure). For guidance, download the 'Figure Guide PDF' (<https://www.embopress.org/page/journal/17574684/authorguide#figureformat>).
- 3) At EMBO Press we ask authors to provide source data for the main figures. Our source data coordinator will contact you to discuss which figure panels we would need source data for and will also provide you with helpful tips on how to upload and organize the files.
- 4) A .docx formatted letter INCLUDING the reviewers' reports and your detailed point-by-point responses to their comments. As part of the EMBO Press transparent editorial process, the point-by-point response is part of the Review Process File (RPF), which will be published alongside your paper.
- 5) A complete author checklist, which you can download from our author guidelines (<https://www.embopress.org/page/journal/17574684/authorguide#submissionofrevisions>). Please insert information in the checklist that is also reflected in the manuscript. The completed author checklist will also be part of the RPF.
- 6) All Materials and Methods need to be described in the main text using our 'Structured Methods' format, which is required for all research articles. According to this format, the Methods section includes a Reagents and Tools Table (listing key reagents, experimental models, software and relevant equipment and including their sources and relevant identifiers) followed by a Methods and Protocols section describing the methods using a step-by-step protocol format. The aim is to facilitate adoption of the methodologies across labs. More information on how to adhere to this format as well as a downloadable template (.docx) for the Reagents and Tools Table can be found in our author guidelines: <https://www.embopress.org/page/journal/17574684/authorguide#structuredmethods>

7) Please note that all corresponding authors are required to supply an ORCID ID for their name upon submission of a revised manuscript.

8) It is mandatory to include a 'Data Availability' section after the Materials and Methods. Before submitting your revision, primary datasets produced in this study need to be deposited in an appropriate public database, and the accession numbers and database listed under 'Data Availability'. Please remember to provide a reviewer password if the datasets are not yet public (see <https://www.embopress.org/page/journal/17574684/authorguide#dataavailability>).

9) For data quantification: please specify the name of the statistical test used to generate error bars and P values, the number (n) of independent experiments (specify technical or biological replicates) underlying each data point and the test used to calculate p-values in each figure legend. The figure legends should contain a basic description of n, P and the test applied. Graphs must include a description of the bars and the error bars (s.d., s.e.m.). Please provide exact p values.

10) Our journal encourages inclusion of *data citations in the reference list* to directly cite datasets that were re-used and obtained from public databases. Data citations in the article text are distinct from normal bibliographical citations and should directly link to the database records from which the data can be accessed. In the main text, data citations are formatted as follows: "Data ref: Smith et al, 2001" or "Data ref: NCBI Sequence Read Archive PRJNA342805, 2017". In the Reference list, data citations must be labeled with "[DATASET]". A data reference must provide the database name, accession number/identifiers and a resolvable link to the landing page from which the data can be accessed at the end of the reference. Further instructions are available at .

11) We replaced Supplementary Information with Expanded View (EV) Figures and Tables that are collapsible/expandable online. A maximum of 5 EV Figures can be typeset. EV Figures should be cited as 'Figure EV1, Figure EV2' etc... in the text and their respective legends should be included in the main text after the legends of regular figures.

12) The paper explained: EMBO Molecular Medicine articles are accompanied by a summary of the articles to emphasize the major findings in the paper and their medical implications for the non-specialist reader. Please provide a draft summary of your article highlighting

13) For more information: There is space at the end of each article to list relevant web links for further consultation by our readers. Could you identify some relevant ones and provide such information as well? Some examples are patient associations, relevant databases, OMIM/proteins/genes links, author's websites, etc...

14) Author contributions: CRediT has replaced the traditional author contributions section because it offers a systematic machine readable author contributions format that allows for more effective research assessment. Please remove the Authors Contributions from the manuscript and use the free text boxes beneath each contributing author's name in our system to add specific details on the author's contribution. More information is available in our guide to authors.

15) Disclosure statement and competing interests: We updated our journal's competing interests policy in January 2022 and request authors to consider both actual and perceived competing interests. Please review the policy <https://www.embopress.org/competing-interests> and update your competing interests if necessary.

16) Every published paper now includes a 'Synopsis' to further enhance discoverability. Synopses are displayed on the journal webpage and are freely accessible to all readers. They include a short stand first (maximum of 300 characters, including space) as well as 2-5 one-sentences bullet points that summarizes the paper. Please write the bullet points to summarize the key NEW findings. They should be designed to be complementary to the abstract - i.e. not repeat the same text. We encourage inclusion of key acronyms and quantitative information (maximum of 30 words / bullet point). Please use the passive voice. Please attach these in a separate file or send them by email, we will incorporate them accordingly.

17) As part of the EMBO Publications transparent editorial process initiative (see our Editorial at <http://embomolmed.embopress.org/content/2/9/329>), EMBO Molecular Medicine will publish online a Review Process File (RPF) to accompany accepted manuscripts.

In the event of acceptance, this file will be published in conjunction with your paper and will include the anonymous referee reports, your point-by-point response and all pertinent correspondence relating to the manuscript. Let us know whether you agree with the publication of the RPF and as here, if you want to remove or not any figures from it prior to publication. Please note that the Authors checklist will be published at the end of the RPF.

I look forward to receiving your revised manuscript.

Yours sincerely,

Lise Roth

Dear Dr. Roth,

Once more I would like to address our gratitude for the opportunity to revise our manuscript. We deeply appreciate it.

Independent expert:

"I have reviewed the comments by the authors and referees, and I am in quite solid agreement with referee #2 and supportive of the authors and the manuscript. While some clarification of the flow cytometry results across different gating strategies as well as tumor growth kinetic measurements would be helpful from the authors prior to publication, the other comments by referee #3 do not preclude publication in my view. The comment "the most common protocol is to inject 1000 cells" is sometimes true but certainly an oversimplification, as growth kinetics of pancreatic cancer cell lines vary immensely across models. The KP2 model used in this manuscript is on the slower-growing end and, as used by the lab which generated this line, is used typically by injecting 200,000 cells into the pancreas (many examples but one is PMID: 31076405)."

Our reply:

We appreciate the comments and suggestions of the reviewer as well as the overall positive perception of our revised manuscript.

We have now included the updated FACS results that have been obtained with the gating strategy suggested by reviewer 3 and would like to provide the following clarifications on the gating:

T cell gating strategy:

In the initial point-by-point reply, we described T cell subsets as follows:

We defined effector T cells as CD62L⁻ T cells, as shown in the FACS gating below (Fig. 2D and E), and naïve T cells are identified with CD62L⁺CD44⁻ T cells, as shown in the detailed FACS gating strategy, Fig S2C.

Fig. 2D

Fig. 2E

We have now adjusted the arrows to make it easier to understand the gating strategy:

Figure used before:

New figure layout:

Figure legend in the manuscript (Appendix SF1C): Flow cytometry gating strategy of naïve, CD62L⁻ effector CD4⁺ T cells and CD8⁺ T cells. CD4⁺ T cells were identified as CD45⁺CD3⁺CD4⁺ cells, and CD8⁺ T cells as CD45⁺CD3⁺CD8⁺ cells. For naïve CD4⁺ and CD8⁺ T cells, they were defined as CD62L⁺CD44⁻CD4⁺ T cells or CD62L⁺CD44⁻CD8⁺ T cells. For effector CD4⁺ and CD8⁺ T cells, they were defined as CD62L⁻CD4⁺ T cells and CD62L⁻CD8⁺ T cells, respectively.

Myeloid cell gating strategy:

In the first point-by-point reply, we described myeloid cell as follows:

Neutrophils are identified with: $CD45^+ CD11b^+ Ly6G^+$ cells. monocytes were gated based on non-neutrophils (negative gating), as $CD11b^+ Ly6C^+$ population. Then, monocyte macrophages were further defined with an $F4/80^+$ signal.

We have now adjusted the arrows to make it easier to understand the gating strategy:

Figure used before:

New figure layout:

Figure legend in the manuscript (Fig EV2A): Flow cytometry gating strategy of myeloid cells. Neutrophils were identified as $CD45^+ CD11b^+ Ly6G^+$ cells. monocytes were gated based on non-neutrophils (negative gating), as $CD11b^+ Ly6C^+$ population. Then, monocyte macrophages were further defined with $F4/80^+$ signal.

We are also grateful for the opportunity to provide some clarification on the measurement of the tumor growth kinetics.

Our reply:

For the subcutaneous PDAC mouse model: We did a pilot study with different amounts of cells (5×10^4 , 10^5 , and 2×10^5 cells) before deciding 5×10^4 cells for the subcutaneous study. The focus on our study is on tumor desmoplasia, a hallmark of PDAC and the tumor stroma, namely the CAF populations. We acknowledge that an injection of 2×10^5 cells in a subcutaneous model led to faster tumor growth (measurable between day 8 and 10) and bigger tumor volume at the endpoint, yet at the expense of reduced tumor-stroma co-evolution and tumor desmoplasia within the PDAC tumor. With 5×10^4 KP2 cell injection, we observe proper desmoplasia and that more than half of the tumors were measurable between day 14 and day 16. While it's difficult to say when the tumor growth slow down exactly, as in the pilot study, the mice could only be kept 4 weeks after the cell injection.

Finally, we have now added the results from the treatment of subcutaneous PDACs with the cytotoxic agent gemcitabine along with with the correct CASPase 3 antibody designation to Figure EV3 of the manuscript.

Figure EV3D and E

The results are described as follows in the manuscript:

The poorly vascularized stromal matrix is and prevents drug delivery into the tumor, and contributes to resistance to therapy (Thomas and Radhakrishnan, 2019). When we treated subcutaneous PDACs on day 29 after the second therapeutic v-A12 vaccination (day 28 after tumor cell injection) with the cytotoxic agent gemcitabine (10 mg/kg, Fig. EV3D), the chemotherapy-induced tumor cell apoptosis was significantly enhanced in the v-A12 cohort Fig. EV3E). This suggests that, in combination with chemotherapy, v-A12 is likely to enhance drug delivery and improved responsiveness of the tumor to cytotoxic agents.

16th Sep 2024

Dear Prof. Stockmann,

Thank you for submitting your revised study. We have now received the report from the advisor we consulted on your manuscript (referee #4), who is satisfied with the revisions. I will therefore be able to accept your manuscript once the following editorial issues will be addressed:

1/ Manuscript text:

- Please remove the highlights in the text and only keep in track changes mode any new modification.
- Please provide up to 5 keywords.
- Methods:
 - o Thank you for providing a reagents and tools table, please remove it from the manuscript and upload it as a separate file.
 - o Cells: please indicate whether the cells were authenticated and tested for mycoplasma contamination.
 - o Antibodies: please provide dilutions/concentrations.
 - o Statistics: please provide statements on sample size, blinding, randomization and inclusion/exclusion criteria.
- Data Availability section: please provide more details on the patent (authors, number).
- Acknowledgements: The funding listed in this section should match the information entered in the submission system (currently Swiss National Fund (310030_179235), Forschungskredit UZH Postdoc 2019, Grant Number JP23gm651002 are missing in the submission system).
- Disclosure statement and competing interests: We updated our journal's competing interests policy in January 2022 and request authors to consider both actual and perceived competing interests. Please review the policy <https://www.embopress.org/competing-interests> and update your competing interests if necessary.
- Please remove the text "Supplementary material/Appendix (PDF document)/Expanded view figures (PDF document)"
- References should be listed alphabetically, with 10 authors before et al, and DOIs should be removed.

2/ Figures and Appendix:

- Main figures and EV figures should be uploaded as individual, high resolution figure files.
- EV figure legends should be placed after the main figure legends, under the heading "Expanded View Figure Legends".
- Please make sure that the all figures / figure panels are referenced in the text (There are callouts for a Fig. S1F and for Supplementary Figure S1C and S2A, please correct).
- Please remove the yellow highlights from the Appendix.
- We note a partial overlap in Figure 2C v-CTRL between "Edge" and "Interior", please check and clarify. Please also check the raw data provided for this figure.
- Please address the queries from our copy editors in the figure legends:
 1. Please note that the p value is not represented in the figure 2b, however statistical test related information is provided in the legend of the corresponding figure. This needs to be rectified.
 2. Please note that the exact p values are not provided in the legend of figure EV 5b.
 3. Please indicate the statistical test used for data analysis in the legend of figure 3b.
 4. Although 'n' is provided, please describe the nature of entity for 'n' in the legend of figure EV 1d.
 5. Please note that the error bars are not defined in the legend of figure EV 1d.
 6. Please note that the white arrows are not defined in the legend of figure EV 1f; EV 3a; EV 4b; EV 5a. This needs to be rectified.
 7. Please note that the white dotted outline is not defined in the legend of figure EV 5e.

3/ Thank you for providing Source Data. Please upload them as one zipped file per figure.

4/ Checklist:

- Newly created materials: you indicated restrictions, please clarify.
- Please fill in the section Cell materials/ authentication & mycoplasma contamination.
- Please check the section Experimental animals/animals observed in or captured from the field, as I don't think it applied to your manuscript.
- Please fill in the entire section Experimental study design and statistics.

5/ I introduced minor modifications in your Paper Explained, please let me know if you agree or amend as you see fit. Please add this section to the manuscript file.

Problem

As one of the most lethal cancers, PDAC is characterized by an excessive accumulation of ECM components produced by cancer-associated fibroblasts (CAFs), termed desmoplasia. Desmoplasia restricts drug delivery and lymphocyte infiltration into the tumor tissue, rendering PDAC difficult to treat.

Results

We used subcutaneous and orthotopic PDAC mouse models which develop desmoplasia. Vaccination against ADAM12+ reduced CAFs, ECM deposition and, hence, desmoplasia in both models. ADAM12 vaccination further promoted infiltration and a favorable redistribution of different T cell subtypes within the tumor tissue. Additionally, ADAM12 vaccination did not promote tumor metastasis but induced vascular normalization, alleviated tumor hypoxia, and delayed PDAC growth.

Impact

This study validated the efficacy of a vaccine against ADAM12+ cells via improved CD8+ cytotoxic T cell response in two different PDAC tumor mouse models. This approach provides proof of principle for vaccination-based immunotherapies to treat tumor desmoplasia by specifically targeting CAFs.

6/ Synopsis:

I included minor modifications in your synopsis, please let me know if you agree or amend as you see fit:

"ADAM12, a disintegrin and metalloprotease, is expressed in cancer-associated fibroblasts (CAFs) and tumor cells in pancreatic ductal adenocarcinoma (PDAC). Using both subcutaneous and orthotopic PDAC mouse models, we show that vaccination against ADAM12 depletes CAFs and delays tumor growth.

- ADAM12 vaccination induced a reduction of ADAM12+ CAFs, and decreased deposition of extracellular matrix (ECM).
- ADAM12 vaccination increased cytotoxic CD8+ T cell response and re-localization of T cells within the tumor tissue.
- ADAM12 vaccination induced vascular normalization with decreased tumor hypoxia.
- This study constitutes a proof of principle for the development of vaccination-based immunotherapies to target CAFs and tumor desmoplasia."

Thank you for providing a nice synopsis image. Please resize it to 550 px wide x 300-600 px high and make sure that the text remains legible. Please note that a cropped portion of this image will serve as thumbnail for the table of content on our webpage.

7/ As part of the EMBO Publications transparent editorial process initiative (see our Editorial at <http://embomolmed.embopress.org/content/2/9/329>), EMBO Molecular Medicine will publish online a Review Process File (RPF) to accompany accepted manuscripts.

This file will be published in conjunction with your paper and will include the anonymous referee reports, your point-by-point response and all pertinent correspondence relating to the manuscript. Let us know whether you agree with the publication of the RPF and as here, if you want to remove or not any figures from it prior to publication.

I look forward to receiving your revised manuscript.

With kind regards,

Lise Roth

To submit your manuscript, please follow this link:
<https://embomolmed.msubmit.net/cgi-bin/main.plex>

***** Reviewer's comments *****

Referee #4 (Remarks for Author):

The authors have meaningfully addressed comments from the previous submission.

Editor Comments - 1001

In the methods, please indicate whether the cells were authenticated.

Our reply: In the methods/cell lines, we used the following sentence:

“All cell lines were regularly tested negative for mycoplasma contamination.”

We have provided the ATCC number or Cat# for commercial cell lines (NIH 3T3, MutuDc1940) in the reagent tables. For other cell lines, we didn't authenticate them ourselves.

- Figure 2C: Please see the report attached, which shows the potential overlap in v-CTRL condition between "Edge" and "Interior" pictures. The Source Data provided for this figure do not match the pictures provided in the manuscript. Could you please clarify and provide the pictures that were used for quantification?

Our reply: We apologize for providing the wrong source picture in the previous document, which was the source data for figure Fig 2B, and not Fig 2C.

Regarding Fig 2C, thank you for making us aware that the representative pictures did not match the source data. We apologize for the mistake. After checking all images used for quantification, we noticed that the representative pictures were from the same sample, which was used for the CD8⁺ staining trial to define the appropriate antibody dilution ratio and to identify whether the CD8⁺ antibody is suitable. Thus, this sample was not included in the final quantification. Therefore, I would like to change the representative pictures with pictures I have also deposited on Biostudies. Please see below and the revised Figure 2 which is also attached to this mail:

New images:

Old images:

- Thank you for resizing the synopsis. I have cropped a small portion (115x70px) that will serve as a thumbnail in the table of content on our webpage (attached). Please let us know if you agreed with the thumbnail, or provide another one. Changes at proofing stage are usually not allowed.

Our reply: We agree, thank you very much for taking care of this.

9th Oct 2024

Dear Prof. Stockmann,

Thank you for submitting your revised files. I am pleased to inform you that your manuscript is accepted for publication and is now being sent to our publisher to be included in the next available issue of EMBO Molecular Medicine.

Yours sincerely,

Lise Roth
